# PROVABLE BENEFIT OF ADAPTIVITY IN ADAM

## ABSTRACT

Adaptive Moment Estimation (Adam) has been observed to converge faster than stochastic gradient descent (SGD) in practice. However, such an advantage has not been theoretically characterized – the existing convergence rate of Adam is no better than SGD. We attribute this mismatch between theory and practice to a commonly used assumption: the smoothness is globally upper bounded by some constant $L$ (called *L-smooth condition*). Specifically, compared to SGD, Adam adaptively chooses a learning rate better suited to the *local smoothness*. This advantage becomes prominent when the local smoothness varies drastically across the domain, which however is hided under $L$-smooth condition. In this paper, we analyze the convergence of Adam under a condition called $(L_0, L_1)$-smooth condition, which allows the local smoothness to grow with the gradient norm. This condition has been empirically verified to be more realistic for deep neural networks (Zhang et al., 2019a) than the $L$-smooth condition. Under $(L_0, L_1)$-smooth condition, we establish the convergence for Adam with practical hyperparameters. As such, we argue that Adam can adapt to the local smoothness, justifying Adam's *benefit of adaptivity*. In contrast, SGD can be arbitrarily slow under this condition. Hence, we theoretically characterize the advantage of Adam over SGD.

## 1 INTRODUCTION

Machine learning tasks are often formulated as solving the following finite-sum problem.

$$\min_{\boldsymbol{w} \in \mathbb{R}^d} f(\boldsymbol{w}) = \sum_{i=0}^{n-1} f_i(\boldsymbol{w}), \tag{1}$$

where $\{f_i(\boldsymbol{w})\}_{i=0}^{n-1}$ is lower bounded, $n$ denotes the number of samples or mini-batches, and $\boldsymbol{w}$ denotes the trainable parameters. Among various gradient-based optimizers, Stochastic Gradient Descent (SGD) is a simple and popular method to solve Eq. (1). However, Adaptive gradient methods including Adaptive Moment estimation (Adam) (Kingma & Ba, 2014) is recently observed to outperform SGD in modern deep neural tasks including GANs (Brock et al., 2018), BERTs (Kenton & Toutanova, 2019), GPTs (Brown et al., 2020) and ViTs (Dosovitskiy et al., 2020). For instance, as reported in Figure 1. (a): SGD converges much slower than Adam during the training of Transformers. Similar phenomena are also reported in BERT training (Zhang et al., 2019b).

The empirical success of Adam comes from its special update rules. Firstly, it uses the heavy-ball momentum mechanism controlled by a hyperparameter $\beta_1$. Second, it uses an adaptive learning rate strategy. In particular, the learning rate of Adam contains exponential moving averages of past squared gradients, which is weighted by $\beta_2$. Larger $\beta_1$ and $\beta_2$ will bring more gradient information of historical steps into the update. The update rule of Adam is given in Eq. (2) (presented later in Section 3).

Despite its practical success, the theoretical understanding of Adam is limited. For instance, the existing convergence rates of Adam are no better than that of SGD (Zhang et al., 2022; Shi et al., 2021; Défossez et al., 2020; Zou et al., 2019; De et al., 2018; Guo et al., 2021). As such, there is a mismatch between Adam's superior empirical performance and its theoretical understanding.

To close the gap between theory and practice, we revisit the existing analyses for Adam. Current setups for the analysis of Adam fail to model the real-world applications: all of the convergence analyses of Adam are based on $L$-smooth condition, i.e., the Lipschitz coefficient of the gradient is

globally bounded. However, it has been recently observed that $L$-smooth condition does not hold in many deep learning tasks (Zhang et al., 2019a; Crawshaw et al., 2022). This gap in setting can obscure Adam's superiority: different local Lipschitz coefficients of the gradient (local smoothness) may require different optimal learning rates in terms of convergence. However, the learning rate of SGD is ignorant of the local smoothness along the training trajectory. If the local smoothness does not change sharply along the trajectory, the optimal learning rate does not change much. Then SGD can work well by selecting a learning rate through grid search. However, when the local smoothness varies dramatically, a learning rate fits some points well may fit other points along the trajectory arbitrarily badly, which indicates SGD may converge arbitrarily slow (detailed discussions can be found in [Theorem 4, (Zhang et al., 2019a)] and Section 4.3 in this paper). In contrast, Adam adapts the update according to the local information and does not suffer from such an issue.

Following the above methodology, we analyze the convergence of Adam under $(L_0, L_1)$-smooth condition, which assumes the local smoothness (the spectral norm of the local Hesssian) to be upper bounded by $L_1 \cdot$ (local gradient norm) $+ L_0$ (Assumption 1). As the gradient norm can be unbounded, $(L_0, L_1)$-smooth condition allows the local smoothness to grow fiercely with the gradient norm. Moreover, it has been demonstrated by (Zhang et al., 2019a; 2020a; Crawshaw et al., 2022) that for many practical neural networks, $(L_0, L_1)$-smooth condition more closely characterizes the optimization landscape (as illustrated by Figure 1 (b)) along the optimization trajectories.

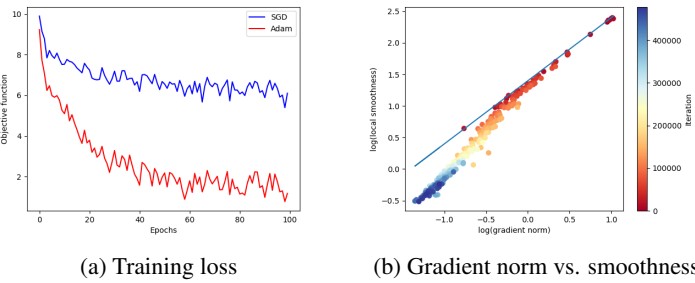

(a) Training loss        (b) Gradient norm vs. smoothness

Figure 1: For Transformer (Vaswani et al., 2017) on the WMT 2014 dataset, we plot (a) the training loss of SGD and Adam and (b) the gradient norm vs. the local smoothness on the training trajectory. The blue line stands for $\log(\text{local smoothness}) = \log(\text{gradient norm}) + 1.4$. It can be observed that all the $(\log(\text{gradient norm}), \log(\text{local smoothness}))$ points lie under this line, and thus the training process obeys $(0, e^{1.4})$-smooth condition.

Under $(L_0, L_1)$-smooth condition, we successfully establish the convergence of Adam. Meanwhile, in contrast, under the same assumption, it is proved that SGD can be arbitrarily slow in (Zhang et al., 2019a) or even diverge (see Section 4.3). Therefore, our theory demonstrates that Adam can provably converge faster than SGD. The main contribution of this paper is summarized as follows.

We derive the first convergence result of Adam under the more realistic $(L_0, L_1)$-smooth condition (Theorem 1). First, our convergence result is established under the mildest assumption so far:

- $(L_0, L_1)$-smooth condition is strictly weaker than $L$-smooth condition. More importantly, $(L_0, L_1)$-smooth condition is observed to hold in practical deep learning tasks. Relaxing the smoothness condition is important to characterize the advantage of Adam over SGD.
- Our result does not require the bounded gradient assumption (i.e. $\|\nabla f(x)\| \leq C$). Removing such a condition is necessary, as otherwise $(L_0, L_1)$-smooth condition will degenerate to $L$-smooth condition. Our result does not need other strong assumptions like bounded adaptor or large $\varepsilon$ (see Eq. (2)), either.

Furthermore, the conclusion of our convergence result is among the strongest.

- Our convergence result holds for every possible trajectory. This is much stronger than the common results of "convergence in expectation" and is technically challenging.
- In our convergence results, the setting of hyperparameters $(\beta_1, \beta_2)$ is close to practice. Specifically, our result holds for any $\beta_1$ and any $\beta_2$ close to 1, which matches the practical settings (for example, 0.9 and 0.999 in deep learning libraries).

We also provide counter-examples where SGD diverges under the same setting as in Theorem 1. Therefore, our results can shed new light on why Adam outperforms SGD in deep learning tasks.

The rest of this paper is organized as follows: Section 2 summarizes the related works; Section 3, introduces notations and assumptions; Section 4 presents our convergence result of Adam along with the proof ideas.

## 2 RELATED WORKS

**Convergence analysis for Adam.** When Adam is firstly proposed in (Kingma & Ba, 2015), the authors provide a convergence proof. However, Reddi et al. (2019) point out that the proof in (Kingma & Ba, 2015) has flaws. They further provide simple counterexamples with which Adam diverges. Ever since then, there have been many attempts to modify the update rules of Adam to ensure convergence: e.g., AMSGrad (Reddi et al., 2019) and AdaBound (Luo et al., 2019). Due to the limited space, we introduce more details of the Adam-variants in Appendix A.

On the other hand, vanilla Adam works well in practice, and the divergence is not reported unless for carefully constructed examples. This empirical phenomenon motivates researchers to rethink the counterexamples. The counterexamples states "for every $\beta_1 < \sqrt{\beta_2}$, there exists a problem that Adam diverges". That is to say, the divergence statement requires picking $(\beta_1, \beta_2)$ before fixing the problem, while in practice, the algorithmic parameters are often picked according to the problem. Based on this observation, a recent work (Zhang et al., 2022) proves that Adam can converge with $(\beta_1, \beta_2)$ picked after the problem is given.

Similar analyses on vanilla Adam and RMSProp are also shown under various conditions in (Zaheer et al., 2018b; Zou et al., 2019; Défossez et al., 2020; De et al., 2018; Guo et al., 2021; Shi et al., 2021). We list and compare these results in Table 1 for convenience. In summary, we emphasize that all the above works (including those for Adam-variants) require $L$-smooth condition. In addition, most of them require stronger assumptions such as bounded gradient or large $\epsilon$.

| | $(\mathbf{L_0}, \mathbf{L_1})$-**Smoothness** | Allow $\beta_1 \neq 0^{(a)}$ | Allow $\varepsilon = 0^{(b)}$ | Allow unbounded gradient | Trajectory-wise convergence |
|---|---|---|---|---|---|
| Zaheer et al. (2018b) | × | × | × | × | × |
| De et al. (2018)$^{(c)}$ | × | × | × | × | × |
| Défossez et al. (2020) | × | ✓ | × | × | × |
| Guo et al. (2021)$^{(d)}$ | × | ✓ | × | × | × |
| Huang et al. (2021)$^{(d)}$ | × | ✓ | × | × | × |
| Shi et al. (2021) | × | × | ✓ | ✓ | ✓ |
| Zhang et al. (2022) | × | ✓ | ✓ | ✓ | × |
| **This work** | ✓ | ✓ | ✓ | ✓ | ✓ |

Table 1: Comparison of existing convergence result of Adam. All the literature requires $L$-smooth condition, which is strictly stronger than $(L_0, L_1)$-smooth condition. We provide some explanation on the upper footmarks: $(a)$: When $\beta_1 = 0$, Adam is reduced to RMSProp (Hinton et al., 2012), the analysis of which is essentially simpler due to the lack of the momentum term; $(b)$: $\varepsilon$ is the stability hyperparameter in Adam (see Eq. (2)). In practice, $\varepsilon$ is often a small number such as $10^{-8}$. In theory, $\varepsilon$ should be arbitrarily small, including 0. $(c)$: Zaheer et al. (2018b) further requires the signs of the gradients to remain the same along the trajectory; $(d)$: Guo et al. (2021); Huang et al. (2021) require $\sqrt{\boldsymbol{\nu}_{k,i} + \varepsilon}$ to be lower bounded, which is equivalent to requiring $\varepsilon > 0$.

In this work, we focus on the convergence of vanilla Adam. In particular, our result is the first to relax $L$-smooth condition for Adam. Moreover our convergence analysis of Adam is proved under the mildest assumptions so far and with the strongest conclusions, which can be easily extended to other Adam-variants as well.

**Generalized smoothness assumption.** There are several attempts on relaxing $L$-smooth condition. Zhang et al. (2019a) propose $(L_0, L_1)$-smooth condition to theoretically explain the acceleration effect of clipped SGD over SGD. Similar results are also extended to clipped SGD with momentum (Zhang et al., 2020a) and differentially-private SGD (DP SGD) (Yang et al., 2022). Through extensive experiments, these works empirically showed that $(L_0, L_1)$-smooth condition holds when Adam outperforms SGD. However, they did not theoretically analyze Adam in this setting.

**Theoretical comparison between adaptive gradient methods and SGD.** The comparison between adaptive gradient methods and SGD is a popular topic. There are several theoretical works from

different perspectives. Ward et al. (2020); Défossez et al. (2020); Faw et al. (2022) show that AdaGrad can converge with any constant learning rate. They argue that "tuning-free" is one advantage of AdaGrad over SGD. This line of works has the following differences with us. First, they analyze AdaGrad while we focus on Adam. Second, they require $L$-smooth, while we do not have this condition. Third, their proposed convergence rate is no better than SGD, while we show Adam can converge faster.

Xie et al. (2022) study the escaping rate from saddle points of the continuous dynamics approximation of Adam and SGD. They show that adaptive learning rate can help escape saddle points efficiently. This work is orthogonal to ours: they compare the behavior of Adam and SGD *after* a certain stationary point is reached, while we focus on comparing the iteration complexity of approaching the stationary points.

Zhang et al. (2020b) prove that the (adaptive) clipped gradient method converges faster than SGD when the stochastic noises are heavy-tailed. As such, they argue that the heavy-tailed noise is one cause of SGD's poor performance. However, a recent work (Chen et al., 2021b) shows that (S)GD still performs poorly even when the stochastic noise is removed in the training process (by using the full-batch gradient), which shows that the effect of the stochastic noise may not be crucial. There are also recent works (Zhou et al., 2020; Zou et al., 2021) trying to compare the generalization behaviors of adaptive gradient methods and SGD. These works are orthogonal to the topic of this work. More detailed discussion is deferred to Appendix A.

## 3 PRELIMINARIES

This section introduces notations, definitions, and assumptions that are used throughout this work.

**Notations.** We list the notations that are used in the formal definition of the randomly-shuffled Adam and its convergence analysis.

- (Vector) We define $\boldsymbol{a} \odot \boldsymbol{b}$ as the Hadamard product (i.e., component-wise product) between two vectors $\boldsymbol{a}$ and $\boldsymbol{b}$ with the same dimension. We also define $\langle \boldsymbol{a}, \boldsymbol{b} \rangle$ as the $\ell^2$ inner product between $\boldsymbol{a}$ and $\boldsymbol{b}$, and $\|\boldsymbol{a}\|_p$ as the $\ell^p$ norm of $\boldsymbol{a}$ (specifically, we abbreviate the $\ell^2$ norm of $\boldsymbol{a}$ as $\|\boldsymbol{a}\|$). We define $\mathbb{1}_d$ as an all-one vector with dimension $d$.

- (Derivative) For a function $f(\boldsymbol{w}) : \mathbb{R}^d \to \mathbb{R}$, we define $\nabla f(\boldsymbol{w})$ as the gradient of $f$ at point $\boldsymbol{w}$, and $\partial_l f(\boldsymbol{w})$ as the $l$-th partial derivative of $f$ at point $\boldsymbol{w}$, i.e., $\partial_l f(\boldsymbol{w}) = (\nabla f(\boldsymbol{w}))_l$.

- (Array) We define $[m_1, m_2] \triangleq \{m_1, \cdots, m_2\}, \forall m_1, m_2 \in \mathbb{N}, m_1 \leq m_2$. Specifically, we use $[m] \triangleq \{1, \cdots, m\}$.

**Formal Definition of Adam.** Based on the $n$-sum problem Eq. (1), we provide the update rule of Adam as follows. We initialize $\boldsymbol{w}_{1,0}, \boldsymbol{m}_{1,-1}$, and $\boldsymbol{\nu}_{1,-1}$ as any point in $\mathbb{R}^d$. At the beginning of every outer loop $k \in \mathbb{N}^+$, we sample $\{\tau_{k,0}, \cdots, \tau_{k,n-1}\}$ as a random permutation of $\{0, 1, \cdots, n-1\}$. Then, in each inner loop $i \in [0, n-1]$, we respectively calculate the $1st$-order momentum $\boldsymbol{m}_{k,i}$, the $2st$-order momentum $\boldsymbol{\nu}_{k,i}$, and the parameter $\boldsymbol{w}_{k,i+1}$ as

$$
\begin{cases}
\boldsymbol{m}_{k,i} = \beta_1 \boldsymbol{m}_{k,i-1} + (1 - \beta_1) \nabla f_{\tau_{k,i}}(\boldsymbol{w}_{k,i}), \\
\boldsymbol{\nu}_{k,i} = \beta_2 \boldsymbol{\nu}_{k,i-1} + (1 - \beta_2) \nabla f_{\tau_{k,i}}(\boldsymbol{w}_{k,i}) \odot \nabla f_{\tau_{k,i}}(\boldsymbol{w}_{k,i}), , \\
\boldsymbol{w}_{k,i+1} = \boldsymbol{w}_{k,i} - \frac{\eta_k}{\sqrt{\boldsymbol{\nu}_{k,i}} + \varepsilon \mathbb{1}_d} \odot \boldsymbol{m}_{k,i}.
\end{cases} \tag{2}
$$

In the end of the outer loop $k$, Adam updates $\boldsymbol{w}_{k+1,0} = \boldsymbol{w}_{k,n}, \boldsymbol{\nu}_{k+1,-1} = \boldsymbol{\nu}_{k,n-1}, \boldsymbol{m}_{k+1,-1} = \boldsymbol{m}_{k,n-1}$ as a preparation for the next outer loop. $\boldsymbol{m}_{k,i}$ and $\boldsymbol{\nu}_{k,i}$ are weighted averages with hyper-paramter $\beta_1 \in [0, 1)$ and $\beta_2 \in [0, 1)$, respectively. We choose $\eta_k$ to be diminishing learning rate $\eta_k = \frac{\eta_1}{\sqrt{k}}$ In practice, $\varepsilon$ is adopted for numerical stability and it is often chosen to be $10^{-8}$ in practice. In our theory, we allow $\varepsilon$ to be an arbitrary non-negative constant including 0. We further emphasize that that our analysis holds for any random permutation order used to generate $\tau_{k,i}$. In the default setting in deep learning libraries, the pure Adam is implemented in the random shuffling fashion and it is the default setting for computer vision, NLP, and generative models.

We make two mild assumptions on the objective function (Eq. (1)).

**Assumption 1** ($(L_0, L_1)$-smooth condition)**.** $f_i(\boldsymbol{w})$ *satisfies* $(L_0, L_1)$-*smooth condition, i.e., there exist positive constants* $(L_0, L_1)$*, such that,* $\forall \boldsymbol{w}_1, \boldsymbol{w}_2 \in \mathbb{R}^d$ *satisfying* $\|\boldsymbol{w}_1 - \boldsymbol{w}_2\| \leq \frac{1}{L_1}$,

$$\|\nabla f_i(\boldsymbol{w}_1) - \nabla f_i(\boldsymbol{w}_2)\| \leq (L_0 + L_1\|\nabla f_i(\boldsymbol{w}_1)\|)\|\boldsymbol{w}_1 - \boldsymbol{w}_2\|. \tag{3}$$

Eq. (3) is firstly introduced by Zhang et al. (2019a). It is strictly weaker than the classical $L$-smooth condition (i.e., $L_1 = 0$ in Assumption 3, see [Remark 2.3, (Zhang et al., 2020a)] for details). For instance, Eq. (3) holds for a wide range of polynomials and even the exponential functions, while $L$-smooth condition does not hold for polynomials with degree larger than 2. More importantly, empirical observation in (Zhang et al., 2019a; 2020a) demonstrates that Eq. (3) is a better characterization of the loss landscape of neural networks than $L$-smooth condition, especially in the tasks where Adam significantly outperforms SGD.

**Assumption 2** (Affine Noise Variance)**.** $\forall \boldsymbol{w} \in \mathbb{R}^d$, *the gradients of* $\{f_i(\boldsymbol{w})\}_{i=0}^{n-1}$ *has the following connection with the gradient of* $f(\boldsymbol{w})$*:*

$$\sum_{i=0}^{n-1} \|\nabla f_i(\boldsymbol{w})\|^2 \leq D_1\|\nabla f(\boldsymbol{w})\|^2 + D_0.$$

Assumption 2 is quite general, which covers the "bounded variance" assumption (Ghadimi et al., 2016; Zaheer et al., 2018a; Huang et al., 2021) and the "strongly growth condition" in existing literature (Schmidt & Roux, 2013; Vaswani et al., 2019). Compare to the "bounded variance" assumption where the noise diminishes when the gradient is large, Assumption 2 allows the noise variance to grow with the gradient norm, and thus applies to a rich variety of problems (e.g., (Khani & Liang, 2020)). Furthermore, combining Assumptions 1 and 2, we conclude that $f$ satisfies $(nL_0 + L_1\sqrt{n}\sqrt{D_0}, L_1\sqrt{n}\sqrt{D_1})$-smooth condition. The proof is immediate and we defer it to Lemma 2 in the appendix.

## 4 MAIN RESULT

### 4.1 ADAM CONVERGES WITH $(L_0, L_1)$-SMOOTH CONDITION

We formally present our main result as follows:

**Theorem 1.** *Consider Adam defined as Eq. (2) with diminishing learning rate* $\eta_k = \frac{\eta_1}{\sqrt{k}}$*. Let Assumptions 1 and 2 hold. Suppose that* $\beta_1$ *and* $\beta_2$ *satisfies* $\beta_1^2 < \beta_2 < 1$ *and*

$$\delta(\beta_2) \triangleq \sqrt{d}\max\left\{1 - \frac{1}{\sqrt{\beta_2^{n-1} + 8n\frac{1-\beta_2^{n-1}}{\beta_2^n}}}, \sqrt{\frac{\beta_2}{1 - \frac{2(1-\beta_2)n}{\beta_2^n}} - 1}, \beta_2^{-n/2} - 1, 1 - \sqrt{\beta_2}\right\}\frac{n}{\beta_2^{\frac{\sqrt{n}}{2}}}$$

$$\leq \frac{1}{2(4 + \sqrt{2})\sqrt{D_1}\left(n - 1 + \frac{1+\beta_1}{1-\beta_1}\right)}. \tag{4}$$

*Then, we have that*

$$\min_{k\in[1,T]}\left\{\frac{\|\nabla f(\boldsymbol{w}_{k,0})\|}{\sqrt{D_1}}, \frac{\|\nabla f(\boldsymbol{w}_{k,0})\|^2}{\sqrt{D_0}}\right\} \leq \mathcal{O}\left(\frac{\log T}{\sqrt{T}}\right) + \mathcal{O}(\sqrt{D_0}\min\{\delta(\beta_2), \delta(\beta_2)^2\}). \tag{5}$$

Theorem 1 shows that Adam converges to the neighborhood of stationary points under $(L_0, L_1)$-smooth condition. The coefficient of $\frac{\log T}{\sqrt{T}}$ is $O(L_0^2 + L_1^2)$ with respect to $(L_0, L_1)$. In Section 4.2, we present a proof sketch to illustrate our proof idea.

**Remark 1** (The choice of hyperparameters $(\beta_1, \beta_2)$)**.** *In Theorem 1, the feasible range of* $(\beta_1, \beta_2)$ *subject to the two constraints:* $\beta_1^2 < \beta_2 < 1$ *and Inequality (4). We emphasize that the intersection of these two constraints is not empty. One can easily observe that* $\delta(\beta_2) \to 0$ *as* $\beta_2 \to 1$*, and thus the constraints hold when* $\beta_1$ *is fixed and* $\beta_2$ *approaches* $1$*. Therefore, Theorem 1 indicates that Adam can work when* $\beta_2$ *close enough to* $1$*. On the other hand, there is little restriction on* $\beta_1$ *except* $\beta_1^2 < \beta_2$ *(discussed in Appendix B). This agrees with* $\beta_2$ *selection in deep learning libraries (e.g.,* $0.999$ *in Pytorch). On the other hand, there are counterexamples in (Zhang et al., 2022), where Adam is showed to diverge if* $\beta_2$ *are chosen improperly (i.e., not close to* $1$*). Therefore, the condition that* $\beta_2$ *is close to* $1$ *is both sufficient and necessary for the convergence of Adam.*

**Remark 2** (Convergence to the neighborhood of stationary points). *When $D_0 \neq 0$, our theorem can only ensure that Adam converges to a neighborhood of stationary points. As discussed under Assumption 2, convergence to the bounded region is common for Adam analysis: even with L-smooth condition, Zhang et al. (2022) can only show that Adam with diminishing learning rates converges to a bounded region. This is because even with diminishing $\eta_k$, the effective learning rate $\frac{\eta_k}{\varepsilon \mathbb{1}_d + \sqrt{\boldsymbol{\nu}_{k,i}}}$ may not decay. We further conduct experiments and observe that Adam cannot reach the stationary point over a simple quadratic function with $D_0 \neq 0$. Specifically, we run Adam on a synthetic optimization target which is given as follows:*

$$f_0(x) = (x-3)^2 + \frac{1}{10}, f_i(x) = -\frac{1}{10}\left(x - \frac{10}{3}\right)^2 + \frac{1}{10}, \ i \in [1,9], \ f(x) = \sum_{i=0}^{9} f_i(x) = \frac{1}{10}x^2.$$

*Such an example is proposed by (Shi et al., 2021). One can easily observe that $f(x)$ and $\{f_i(x)\}_{i=0}^{9}$ satisfies Assumption 1 and Assumption 2 with $D_1 = 208$, $D_0 = 75\frac{5}{9}$, thus Theorem 1 applies. With the diminishing learning rate $\eta_t = \frac{1}{\sqrt{t}}$, we plot the distance between Adam's trajectory and the unique minimum 0 of $f$ in Figure 2. It can be observed that Adam does not converge to the unique minimum 0, but gets closer to the minimum if choosing $\beta_2$ closer to 1, which supports Theorem 1.*

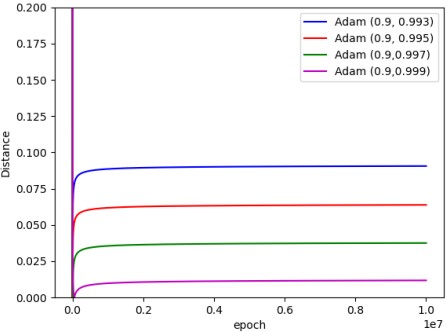

Figure 2: Performance of Adam on a synthetic objective satisfying $(L_0, L_1)$-smooth condition. Adam doesn't converge to the unique stationary point, but gets closer to the stationary point as $\beta_2 \to 1$.

*The good news is the neighborhood shrinks as $\beta_2 \to 1$, which explains the practical use of large $\beta_2$. Furthermore, we refine the bounded region $\{\boldsymbol{w} : \min\{\frac{\|\nabla f(\boldsymbol{w}))\|}{\sqrt{D_1}}, \frac{\|\nabla f(\boldsymbol{w})\|^2}{\sqrt{D_0}}\} \leq \mathcal{O}(\sqrt{D_0}\delta(\beta_2))\}$ in existing literature (Shi et al., 2021; Zhang et al., 2022) to $\{\boldsymbol{w} : \min\{\frac{\|\nabla f(\boldsymbol{w}))\|}{\sqrt{D_1}}, \frac{\|\nabla f(\boldsymbol{w})\|^2}{\sqrt{D_0}}\} \leq \mathcal{O}(\sqrt{D_0}\min\{\delta(\beta_2), \delta(\beta_2)^2\})\}$ through a more careful analysis. We emphasize that this change contributes to a sharp improvement since $\delta(\beta_2)$ is required to be close to 0.*

**Remark 3** (Dependence on the dimension $d$). *The convergence rate and the neighborhood have polynomial dependence over the dimension $d$, which is not desired as the convergence rate of SGD under L-smooth condition does not have such a dependence. We conjecture that such a dependence may be due to the limit of proof techniques. However, such a result suffices to provide a convergence rate separation between Adam and SGD since we will latter show that SGD may converge arbitrarily slow under $(L_0, L_1)$-smooth condition. Furthermore to the best of our knowledge, all the existing convergence analysis of Adam has the dependence of $d$ even under L-smooth condition e.g., (Défossez et al., 2020; Shi et al., 2021; Zhang et al., 2022). Though we believe that removing the dependence on $d$ is possible and worthy studying, it is technically difficult and deserves an independent work.*

**Remark 4** (Constant learning rate). *Following the same proof strategy of Theorem 1, one can show that with constant learning rate $\eta$, the conclusion (Eq. (5)) of Theorem 1 changes into $\min_{k \in [1,T]}\{\frac{\|\nabla f(w_{k,0})\|}{\sqrt{D_0}}, \frac{\|\nabla f(w_{k,0})\|^2}{\sqrt{D_1}}\} \leq \mathcal{O}(\frac{\log T}{\eta T}) + \mathcal{O}(\sqrt{D_0}\min\{\delta(\beta_2), \delta(\beta_2)^2\}) + \mathcal{O}(\eta)$. In other words, Adam will converge faster to the neighborhood (with rate $1/\sqrt{t} \to 1/t$), but the size of neighborhood is enlarged by an additional term $\mathcal{O}(\eta)$ due to the constant step-size.*

**Remark 5.** *Theorem 1 allows both $\beta_1 = 0$ (RMSprop) and $\beta_1 > 0$ (Adam). However, the convergence rate of these two cases are not distinguished, showing no benefit of momentum so far. At the current*

*stage, we take a first step to compare RMSProp & Adam to SGD. A more detailed comparison between RMSprop and Adam will be considered as future work. For completeness, we briefly discuss on the difficulty in analyzing the effect of momentum in Appendix B.*

### 4.2 PROOF IDEAS

Here, we briefly explain our proof idea for the convergence result. We will first prove Theorem 1 for RMSProp (i.e., $\beta_1 = 0$) to show the challenge brought by $(L_0, L_1)$-smooth condition and how we tackle it. We then show the additional difficulty when extending the result to general Adam and our corresponding intuition to solve it.

**Stage I: Convergence of RMSProp.** We start with the following descent lemma:

$$f(\boldsymbol{w}_{k+1,0}) - f(\boldsymbol{w}_{k,0}) \leq \underbrace{\langle \boldsymbol{w}_{k+1,0} - \boldsymbol{w}_{k,0}, \nabla f(\boldsymbol{w}_{k,0}) \rangle}_{\text{First Order}} + \underbrace{\frac{L_{loc}}{2} \|\boldsymbol{w}_{k+1,0} - \boldsymbol{w}_{k,0}\|^2}_{\text{Second Order}}, \tag{6}$$

where $L_{loc}$ is the local smoothness. We first bound the second order term by noticing that the norm of the epoch-$k$ update of RMSProp is in order $\mathcal{O}(\eta_k)$ when $0 = \beta_1^2 < \beta_2 < 1$ (a standard property with proof deferred to Lemma 3 in the appendix). When $k$ is large enough, the epoch-$k$ update is small enough and $(L_0, L_1)$-smooth condition applies, leading to $L_{loc} = \mathcal{O}(1) + \mathcal{O}(\|\nabla f(w)\|)$ and the second order term $= \mathcal{O}(\eta_k^2) + \mathcal{O}(\eta_k^2 \|\nabla f(w)\|)$. Therefore, in order to show $f(\boldsymbol{w}_{k,0})$ decreases, we need to prove the first order is negative and dominates the term $\mathcal{O}(\eta_k^2) + \mathcal{O}(\eta_k^2 \|\nabla f(w)\|)$. In other words, we need to lower bound the alignment between the update $\boldsymbol{w}_{k+1,0} - \boldsymbol{w}_{k,0} = -\eta_k \sum_i \frac{1}{\varepsilon \mathbb{1}_d + \sqrt{\boldsymbol{\nu}_{k,i}}} \odot \nabla f_{\tau_{k,i}}(\boldsymbol{w}_{k,i})$ and the negative gradient $-\nabla f(\boldsymbol{w}_{k,0})$. It is easy to prove such a property when $\boldsymbol{\nu}_{k,i}$ does not change, i.e., if we are analyzing SGD with preconditioning, in which case $\sum_i \frac{1}{\varepsilon \mathbb{1}_d + \sqrt{\boldsymbol{\nu}_{k,i}}} \odot \nabla f_{\tau_{k,i}}(\boldsymbol{w}_{k,i}) = \sum_i \frac{1}{\varepsilon \mathbb{1}_d + \sqrt{\boldsymbol{\nu}_{k,0}}} \odot \nabla f_{\tau_{k,i}}(\boldsymbol{w}_{k,i}) \approx \sum_i \frac{1}{\varepsilon \mathbb{1}_d + \sqrt{\boldsymbol{\nu}_{k,0}}} \odot \nabla f_{\tau_{k,i}}(\boldsymbol{w}_{k,0}) = \frac{1}{\varepsilon \mathbb{1}_d + \sqrt{\boldsymbol{\nu}_{k,0}}} \odot \nabla f(\boldsymbol{w}_{k,0})$. Can we extend the above methodology to RMSProp? We give an affirmative answer when $\beta_2$ is close to 1 and the gradient is large by providing the following lemma.

**Lemma 1** (Informal). *For any* $l \in [d]$ *and* $i \in [0, n-1]$, *if* $\max_{p \in [0, n-1]} |\partial_l f_p(\boldsymbol{w}_{k,0})| = \Omega(\sum_{r=1}^{k-1} \beta_2^{\frac{(k-1-r)}{2}} \eta_r \|\nabla f(\boldsymbol{w}_{r,0})\| + \eta_k)$, *then* $|\boldsymbol{\nu}_{l,k,i} - \boldsymbol{\nu}_{l,k,0}| = \mathcal{O}(\delta(\beta_2) \boldsymbol{\nu}_{l,k,0})$.

The proof idea is that (1). the squared maximum gradient can be bounded by $\boldsymbol{\nu}_{l,k,0}$ with an error term. Thus when the maximum gradient is larger than the error term, it can be only bounded by $\boldsymbol{\nu}_{l,k,0}$; (3). thus $\boldsymbol{\nu}_{l,k,i} = \beta_2^i \boldsymbol{\nu}_{l,k,0} + (1-\beta_2)\beta_2^{i-1}\nabla f_{\tau_{k,i}}(\boldsymbol{w}_{k,i}) + \cdots = \beta_2^i \boldsymbol{\nu}_{l,k,0} + (1-\beta_2^i)\mathcal{O}(\boldsymbol{\nu}_{l,k,0})$ gets close to $\boldsymbol{\nu}_{l,k,0}$ when $\beta_2$ is close to 1. The detailed proof is sophisticated due to the presence of $(L_0, L_1)$-smooth condition and we defer it to Corollary 2 for details. Therefore, if we denote those dimensions with large gradients (i.e., satisfying the requirement of Lemma 1) as $\mathbb{L}_{large}^k$ and the rest as $\mathbb{L}_{small}^k$, Lemma 1 indicates

$$\sum_{l \in \mathbb{L}_{large}^k} (\boldsymbol{w}_{l,k+1,0} - \boldsymbol{w}_{l,k,0}) \partial_l f(\boldsymbol{w}_{k,0}) = -\eta_k \sum_{l \in \mathbb{L}_{large}^k} \frac{\partial_l f(\boldsymbol{w}_{k,0})}{\sqrt{\boldsymbol{\nu}_{l,k,i}} + \varepsilon} \sum_i \partial_l f_{\tau_{k,i}}(\boldsymbol{w}_{k,i}) \approx -\eta_k \sum_{l \in \mathbb{L}_{large}^k} \frac{\partial_l f(\boldsymbol{w}_{k,0})^2}{\sqrt{\boldsymbol{\nu}_{l,k,0}} + \varepsilon}$$

$$\approx -\eta_k \sum_{l \in \mathbb{L}_{large}^k} \frac{\partial_l f(\boldsymbol{w}_{k,0})^2}{\max_{p \in [0, n-1]} |\partial_l f_p(\boldsymbol{w}_{k,0})| + \varepsilon} = -\Omega\left(\eta_k \min\left\{\frac{\|\nabla f(\boldsymbol{w}_{k,0})\|}{\sqrt{D_1}}, \frac{\|\nabla f(\boldsymbol{w}_{k,0})\|^2}{\sqrt{D_0}}\right\}\right).$$

Here in the second "$\approx$", we use again the squared maximum gradient is close to $\boldsymbol{\nu}_{l,k,0}$ if it is large. A formal derivation of the above result can be seen in Appendix D.2. How about the dimensions in $\mathbb{L}_{small}^k$? We will treat them as error terms. Specifically, $l \in \mathbb{L}_{small}^k$ indicates that $\partial_l f(\boldsymbol{w}_{k,0}) = \mathcal{O}(\sum_{r=1}^{k-1} \beta_2^{\frac{(k-1-r)}{2}} \eta_r \|\nabla f(\boldsymbol{w}_{r,0})\| + \eta_k)$. One can easily observe that $\frac{\partial_l f_{\tau_{k,i}}(\boldsymbol{w}_{k,i})}{\sqrt{\boldsymbol{\nu}_{l,k,i}} + \varepsilon}$ is bounded, and thus $\sum_{l \in \mathbb{L}_{small}^k} (\boldsymbol{w}_{l,k+1,0} - \boldsymbol{w}_{l,k,0}) \partial_l f(\boldsymbol{w}_{k,0})$ equals to

$$-\eta_k \sum_{l \in \mathbb{L}_{large}^k} \frac{\partial_l f(\boldsymbol{w}_{k,0})}{\sqrt{\boldsymbol{\nu}_{l,k,i}} + \varepsilon} \sum_i \partial_l f_{\tau_{k,i}}(\boldsymbol{w}_{k,i}) = \mathcal{O}\left(\eta_k \left(\sum_{r=1}^{k-1} \beta_2^{\frac{(k-1-r)}{2}} \eta_r \|\nabla f(\boldsymbol{w}_{r,0})\| + \eta_k\right)\right).$$

Putting all the estimates together, we conclude that $f(\boldsymbol{w}_{k+1,0}) - f(\boldsymbol{w}_{k,0})$ is smaller than $-\Omega(\eta_k \min\{\frac{\|\nabla f(\boldsymbol{w}_{k,0})\|}{\sqrt{D_1}}, \frac{\|\nabla f(\boldsymbol{w}_{k,0})\|^2}{\sqrt{D_0}}\}) + \mathcal{O}(\eta_k(\sum_{r=1}^k \beta_2^{\frac{(k-1-r)}{2}} \eta_r \|\nabla f(\boldsymbol{w}_{r,0})\|)) + \mathcal{O}(\eta_k^2)$. The accumulation of $\mathcal{O}(\eta_k^2)$ is in order $\log T$ and grows slowly. To derive the convergence result, we need to show

the first term dominates. However, the second term contains historical gradient information, which is not necessarily smaller than the first term in absolute value. How to deal with it? We observe that while the first and the second terms are not comparable for single $k$, summing over $k$ leads to $f(\boldsymbol{w}_{T+1,0}) - f(\boldsymbol{w}_{1,0})$ is smaller than

$$-\Omega\left(\sum_{k=1}^{T}\eta_k \min\{\frac{\|\nabla f(\boldsymbol{w}_{k,0})\|}{\sqrt{D_1}}, \frac{\|\nabla f(\boldsymbol{w}_{k,0})\|^2}{\sqrt{D_0}}\}\right) + \mathcal{O}\left(\sum_{k=1}^{T}\eta_k\left(\sum_{r=1}^{k}\beta_2^{\frac{(k-1-r)}{2}}\eta_r\|\nabla f(\boldsymbol{w}_{r,0})\|\right)\right) + \mathcal{O}\left(\sum_{k=1}^{T}\eta_k^2\right).$$

With a sum order change, one can easily observe that the last term equals to $\mathcal{O}(\sum_{k=1}^{T}\eta_k^2\|\nabla f(\boldsymbol{w}_{k,0})\|)$, which is also $\mathcal{O}(\sum_{k=1}^{T}\eta_k \min\{\frac{\|\nabla f(\boldsymbol{w}_{k,0})\|}{\sqrt{D_1}}, \frac{\|\nabla f(\boldsymbol{w}_{k,0})\|^2}{\sqrt{D_0}}\}) + \mathcal{O}(\eta_k^2)$ due to the mean value inequality $\eta_k^2\|\nabla f(\boldsymbol{w}_{k,0})\| \le \mathcal{O}(\eta_k^2) + \mathcal{O}(\eta_k^2\sqrt{\frac{D_1}{D_0}}\|\nabla f(\boldsymbol{w}_{k,0})\|^2)$. Thus the last term is dominated by the last to second term. This completes the proof for RMSProp.

**Remark 6** (Difficulty compared to $L$-smooth case). *First of all, the change of $\boldsymbol{\nu}_{k,i}$ is easier to bound without the historical gradient term. Secondly, under $L$-smooth condition, the error does not contain historical gradient information and is only in order of $\mathcal{O}(\eta_k^2)$, which is easy to bound.*

**Remark 7** (Intuition why SGD converges slowly). *If we replace Adam with SGD in the above analysis, the first order term may no longer dominates the second order term. This explains why SGD converges slowly. A detailed discussion can be found in Appendix B.*

**Stage II: Extend the proof to general Adam.** For Adam, the update norm is still bounded and the second order term can be bounded similarly. However, the analysis of the first order term becomes more challenging even though we still have $\boldsymbol{\nu}_{k,i} \approx \boldsymbol{\nu}_{k,0}$. Specifically, even with constant $\boldsymbol{\nu}_{k,i} = \boldsymbol{\nu}_{k,0}$, $-\eta_k\langle\sum_i \frac{\boldsymbol{m}_{k,i}}{\sqrt{\boldsymbol{\nu}_{k,i}}+\varepsilon}, -\nabla f(\boldsymbol{w}_{k,0})\rangle > 0$ is not necessarily correct, as the momentum $\boldsymbol{m}_{k,i}$ contains a heavy historical signal, and may push the update away from the negative gradient direction. How to deal with this challenge?

We observe that the alignment of $\boldsymbol{w}_{k+1,0} - \boldsymbol{w}_{k,0}$ and $-\nabla f(\boldsymbol{w}_{k,0})$ is required due to that our analysis is based on the potential function $f(\boldsymbol{w}_{k,0})$. However, while this potential function is suitable for the analysis of RMSProp, it is no longer appropriate for Adam based on the above discussion and we need to construct another potential function. Our construction of the potential function is based on the following observation: we revisit the update rule in Eq. (2) and rewrite it in the following manner:

$$\frac{\boldsymbol{m}_{k,i} - \beta_1\boldsymbol{m}_{k,i-1}}{1 - \beta_1} = \nabla f_{\tau_{k,i}}(\boldsymbol{w}_{k,i}).$$

Notice that the right-hand-side of the above equation contains no historical gradients but only the gradient of the current step! We then get $1/\sqrt{\boldsymbol{\nu}_{k,i}}$ into the play, leading to

$$\frac{\boldsymbol{w}_{k,i+1} - \beta_1\boldsymbol{w}_{k,i}}{1 - \beta_1} - \frac{\boldsymbol{w}_{k,i} - \beta_1\boldsymbol{w}_{k,i-1}}{1 - \beta_1} = \frac{\boldsymbol{w}_{k,i+1} - \boldsymbol{w}_{k,i} - \beta_1(\boldsymbol{w}_{k,i} - \boldsymbol{w}_{k,i-1})}{1 - \beta_1}$$

$$\approx -\frac{\eta_k}{\sqrt{\boldsymbol{\nu}_{k,0}}+\varepsilon\mathbb{1}_d}\odot\frac{\boldsymbol{m}_{k,i} - \beta_1\boldsymbol{m}_{k,i-1}}{1 - \beta_1} = -\frac{\eta_k}{\sqrt{\boldsymbol{\nu}_{k,0}}+\varepsilon\mathbb{1}_d}\odot\nabla f_{\tau_{k,i}}(\boldsymbol{w}_{k,i}).$$

One can see that the sequence $\{\boldsymbol{u}_{k,i} \triangleq \frac{\boldsymbol{w}_{k,i} - \beta_1\boldsymbol{w}_{k,i-1}}{1-\beta_1}\}$ are (approximately) doing SGD within one epoch (with coordinate-wise but constant learning rate $\boldsymbol{\nu}_{k,i}$)! Further notice that $\boldsymbol{u}_{k,i} = \boldsymbol{w}_{k,i} + \beta_1\frac{\boldsymbol{w}_{k,i}-\boldsymbol{w}_{k,i-1}}{1-\beta_1}$ is close to $\boldsymbol{w}_{k,i}$. Therefore, we choose our potential function as $f(\boldsymbol{u}_{k,i})$. The Taylor's expansion of $f$ at $\boldsymbol{u}_{k,0}$ then provides a new descent lemma, i.e.,

$$f(\boldsymbol{u}_{k+1,0}) - f(\boldsymbol{u}_{k,0}) \le \underbrace{\langle\boldsymbol{u}_{k+1,0} - \boldsymbol{u}_{k,0}, \nabla f(\boldsymbol{u}_{k,0})\rangle}_{\text{First Order}} + \underbrace{\frac{L_0 + L_1\|\nabla f(\boldsymbol{w}_{k,0})\|}{2}\|\boldsymbol{w}_{k+1,0} - \boldsymbol{w}_{k,0}\|^2}_{\text{Second Order}}, \quad (7)$$

By noticing $\boldsymbol{w}_{k,i} \approx \boldsymbol{u}_{k,i} \approx \boldsymbol{u}_{k,0}$, the first order term can be further approximated as follows,

$$\langle\boldsymbol{u}_{k+1,0} - \boldsymbol{u}_{k,0}, \nabla f(\boldsymbol{u}_{k,0})\rangle \approx -\langle\frac{\eta_k}{\sqrt{\boldsymbol{\nu}_{k,0}}+\varepsilon\mathbb{1}_d}\odot\nabla f(\boldsymbol{w}_{k,0}), \nabla f(\boldsymbol{w}_{k,0})\rangle, \quad (8)$$

which is negative, and the challenge is addressed.

**Remark 8.** *Zhang et al. (2022) also encounter the misalignment between $\boldsymbol{w}_{k+1,0} - \boldsymbol{w}_{k,0}$ and $-\nabla f(\boldsymbol{w}_{k,0})$, but they do not use the potential function and can only derive in-expectation convergence. A detailed discussion can be found in Appendix B.*

**Remark 9.** *There are similar potential functions in existing works for other optimizers, but none of them succeeds to derive the result for Adam. We defer a detailed discussion to Appendix B.*

### 4.3 SGD MAY CONVERGE ARBITRARILY SLOW WITH $(L_0, L_1)$-SMOOTH CONDITION

As the $(L_0, L_1)$-smooth condition appears to be a more precise characterization of the landscape of neural networks than $L$-smooth condition, one may wonder how SGD performs under such a circumstance. Here, we find examples that satisfy $(L_0, L_1)$-smooth condition, but GD and SGD diverges. This implies that under this realistic smoothness assumption, (S)GD can be arbitrarily slow.

**GD may converge arbitrarily slow with** $(L_0, L_1)$**-smooth condition.** To begin with, we notice that Zhang et al. (2019a) have already provided counterexamples over which GD converges arbitrarily slowly. However, their examples use constant learning rate GD, which, unfortunately, does NOT match the setting of Theorem 1 (i.e., diminishing learning rate and stochastic update), and thus does not suffice to provide a fair comparison between GD and (full-batch) Adam. To fill this gap in setting, we provide the following example:

**Example 1.** *There exists a domain $\mathcal{X}$ and a target function $f : \mathbb{R}^{\mathcal{X}} \to \mathbb{R}$, such that for every initial learning rate $\eta_1$, there exists a region $\mathcal{E} \subset \mathcal{X}$ with infinite measure, such that GD with diminishing learning rate $\eta_t = \frac{\eta_1}{\sqrt{t}}$ and initialized in $\mathcal{E}$ will drive $f(\boldsymbol{w}_t)$ and $\|\nabla f(\boldsymbol{w}_t)\|$ to infinity.*

We defer a concrete construction of the example to Appendix C.1. We emphasize here that it is a classification problem with linear model, which shares similar properties with practical classification problems. The intuition of this example is that with a fixed initial learning rate, a bad initialization may lead to arbitrarily worse local smoothness due to globally unbounded smoothness, and make the first-step stepsize relatively large. This leads to an even worse (larger) local smoothness at the next step. Also, the increase of the local smoothness can be rapid enough to offset the decrease in learning rate, and eventually drive the loss to divergence. On the contrary, the update norm of Adam is bounded regardless of the initialization, and thus Adam does not suffer from the same problem. One may notice that the divergence issue in the above example can be equivalently viewed as "given initialization, a large learning rate will bring divergence". A natural question is that whether reducing the learning rate will lead to convergence. The answer could be yes, but it is true only if some additional assumption is provided (for example, Assumption 4 in (Zhang et al., 2019a), which assumes the gradient is bounded over a loss sub-level set). However, in this case, the feasible initial learning rate depends on the local smoothness over the trajectory. Since the initial gradient can be arbitrarily large, the initial learning rate needs to be arbitrarily small, which thus leads to a slow convergence rate if bad initialization is picked. This is exactly the intuition of the lower bound in (Zhang et al., 2019a). Please refer to a detailed discussion in Appendix B.

Similar negative results also apply to SGD. Actually, the situation is even worse: SGD still suffer from divergence for learning rate $1/L$, which can lead to convergence for GD.

**Example 2.** *There exists a domain $\mathcal{X}$, a initial point $\boldsymbol{w}_0 \in \mathcal{X}$ and a target function $f = \sum_{i=0}^{n-1} f_i$, over which SGD with initial learning rate $\eta_1 = \frac{1}{L}$ and diminishing learning rate $\eta_t = \frac{\eta_1}{\sqrt{t}}$ diverges while GD with the same setting converges, where $L$ is the smoothness upper bound of the loss sub-level set $\{\boldsymbol{w} \in \mathcal{X} : f(\boldsymbol{w}) \leq f(\boldsymbol{w}_0)\}$.*

We defer the concrete construction to Appendix C.2. The intuition of this example is that even with learning rate $1/L$, a non-full-batch update can step out the loss sub-level set and increase the local smoothness coefficient. Similarly as before, keep reducing the learning rate might lead to convergence. But this only happens when additional assumptions is imposed. Even if so, the convergence rate could be arbitrarily slow.

## 5 CONCLUSION

In this paper, we take the first step to theoretically understand the adaptivity in Adam. Specifically, we prove that *Adam can converge arbitrarily faster than SGD*. Our assumption is realistic and close to practical settings. On the other hand, there is still a lot to explore for Adam's performance. For example, it is interesting to analyze how momentum helps in Adam's optimization. Further, it is interesting to investigate whether Adam can handle more sharp smooth conditions, e.g., smoothness is bounded by a high order polynomial of the gradient norm.

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

## A   ADDITIONAL RELATED WORKS

In this section, we provide discussions on more related works.

**New variants of Adam.**   Ever since Reddi et al. (2019) pointed out the non-convergence issue of
Adam, many new variants of Adam are designed. For instance, Zou et al. (2019); Gadat & Gavra
(2020); Chen et al. (2018b; 2021a) replaced the constant hyperparameters by iterate-dependent ones
e.g. $\beta_{1t}$ or $\beta_{2t}$. AMSGrad (Reddi et al., 2019) and AdaFom (Chen et al., 2018b) enforced $\{v_t\}$ to
be an non-decreasing. Similarly, AdaBound (Luo et al., 2019) imposed constraints $v_t \in [C_l, C_u]$ to
prevent the learning rate from vanishing or exploding. Similarly, Zhou et al. (2018b) adopted a new
estimate of $v_t$ to correct the bias. In addition, there are attempts to combine Adam with Nesterov
momentum (Dozat, 2016) as well as warm-up techniques (Liu et al., 2020a). There are also some
works providing theoretical analysis on the variants of Adam. For instance, Zhou et al. (2018a)
studied the convergence of AdaGrad and AMSGrad. Gadat & Gavra (2020) studied the asymptotic
behavior of a subclass of adaptive gradient methods from landscape point of view. Their analysis
applies to Adam-variants with $\beta_1 = 0$ and $\beta_2$ increasing along the iterates. In summary, all these
works require $L$-smooth condition. Though the new variants of Adam can probably converge, the
convergence rate is no better than SGD.

**Generalization ability of Adaptive gradient methods.**   The generalization ability of Adam is a
hot debate topic. For instance, Wang et al. (2021) studied the implicit bias of adaptive optimization
algorithms on homogeneous neural networks. They proved that the convergent direction of Adam and
RMSProp is the same as SGD. Zhou et al. (2020); Xie et al. (2022); Zou et al. (2021) argue that Adam
preferred sharp local-min while GD prefers the wide ones. As such, they argue that Adam generalizes
worse than SGD. Zou et al. (2021) prove that Adam generalizes worse than SGD over a specific
model. There are also several attempts to improve the generalization ability of Adam. For instance,
Padam (Chen et al., 2018a) introduced a partial adaptive parameter to improve the generalization
performance. AdamW (Loshchilov & Hutter, 2017) improved regularization in Adam by decoupling
the weight decay from the gradient-based update.

## B   ADDITIONAL DISCUSSIONS

### B.1   RESTRICTION OF $\beta_1$ IN THEOREM 1

We discuss a bit more on $\beta_1$. First, The requirement $\beta_1^2 < \beta_2$ is a standard condition in Adam-family
literature (Reddi et al., 2018; Zou et al., 2019; Défossez et al., 2020; Shi et al., 2021). Since $\beta_2$
is suggested to be large, this condition covers flexible choice of $\beta_1 \in [0, 1)$. Second, although the
dependency between $\beta_1$ and $\beta_2$ in Inequality (4) seems complicated, the effect of $\beta_1$ on Inequality
(4) can be omitted when adopting the practical value such as $\beta_1 = 0.9$. This is because the term
"$\frac{1+\beta_1}{1-\beta_1}$" is way smaller than the term "$(n-1)$" in the right-hand-side of Inequality (4). For example,
when $\beta_1 = 0.9$ (the default setting in deep learning libraries), $\frac{1+\beta_1}{1-\beta_1} = 19$. This number is much
smaller than $n$ on real datasets (for instance, on CIFAR-10 (He et al., 2016), $n = 50k/128 \approx 390$;
on ImageNet (You et al., 2017), $n = 1.2m/8k = 150$). As a result, constraint (4) is inactive for
practical choice of $\beta_1$.

### B.2   COMPARISONS OF OPTIMIZERS OVER THE FINE-TUNING TASK

For the case where the gradient along the trajectory is small, $(L_0, L_1)$-smooth condition will degener-
ate to $L$-smooth condition, and thus SGD works well. This may explain the phenomenon that SGD is
also adopted in some finetuning tasks, as pretraining can be viewed as selecting a good initialization
(and we can expect that the gradient is small along the trajectory). While the above discussion is
highly intuitive, it is an interesting future work to formally prove it.

### B.3   ON THE EFFECT OF MOMENTUM

Since Adam and RMSProp both converge under $(L_0, L_1)$ smooth condition, a natural question is
what the effect of momentum is in Adam. However, this is a highly non-trivial question. The effect

of momentum is not clear even for momentum SGD in non-convex optimization, let alone for Adam. We believe that this question is interesting yet beyond the scope of this paper and leave it as a future work.

### B.4 Convergence result of RMSProp

As mentioned in Section 4.2, we provide the convergence result of RMSProp as a corollary of Theorem 1 here.

**Corollary 1.** *Let all the conditions in Theorem 1 hold. Then, for RMSProp, if*

$$\delta(\beta_2) \triangleq \sqrt{d} \max \left\{ 1 - \frac{1}{\sqrt{\beta_2^{n-1} + 8n\frac{1-\beta_2^{n-1}}{\beta_2^n}}}, \sqrt{\frac{\beta_2}{1 - \frac{2(1-\beta_2)n}{\beta_2^n}}} - 1, \beta_2^{-n/2} - 1, 1 - \sqrt{\beta_2} \right\} \frac{n}{\beta_2^{\frac{\sqrt{n}}{2}}}$$

$$\leq \frac{1}{2(4+\sqrt{2})\sqrt{D_1}n}, \tag{9}$$

*then, we have that*

$$\min_{k \in [1,T]} \left\{ \frac{\|\nabla f(\boldsymbol{w}_{k,0})\|}{\sqrt{D_1}}, \frac{\|\nabla f(\boldsymbol{w}_{k,0})\|^2}{\sqrt{D_0}} \right\} \leq \mathcal{O}\left(\frac{\log T}{\sqrt{T}}\right) + \mathcal{O}(\sqrt{D_0} \min\{\delta(\beta_2), \delta(\beta_2)^2\}). \tag{10}$$

Since Adam and RMSProp both converge under $(L_0, L_1)$-smooth condition, a natural question is what the effect of momentum is in Adam. However, this is a highly non-trivial question. Theoretically, the effect of momentum is not clear even for momentum SGD in non-convex optimization, let alone for Adam. Practically, there are existing experiments suggesting that the performance of well-tuned RMSProp can match that of well-tuned Adam (Choi et al., 2019). We believe that this question is interesting but beyond the scope of this paper and leave it as a future work.

### B.5 Advantage of Adam over the GD/SGD with gradient clipping

(Zhang et al., 2019a) shows that GD/SGD with gradient clipping converges under $(L_0, L_1)$ smooth condition. A natural question is that what is the benefit of Adam over GD/SGD with gradient clipping. Honestly, we are not able to answer this question yet, but one potential advantage may be that Adam can handle more complex noise. It is not known whether SGD with gradient clipping can converge in our setting, as the existing analyses of GD/SGD with gradient clipping under $(L_0, L_1)$-smooth condition all assume that the distance between stochastic gradient and true gradient are bounded with probability 1 ([Zhang et al., 2019] and [Zhang et al. 2020]), which is strictly stronger than our affine variance noise assumption. It will be interesting to either provide counterexample that SGD with clipping does not converge under $(L_0, L_1)$-smooth condition with affine noise assumption, or prove that SGD with clipping does converge and find other perspective to demonstrate the advantage of Adam over SGD with clipping.

### B.6 Intuition why Adam converges faster than SGD

We first illustrate this from the proof. If we analyze SGD following the same methodology of Section 4.2 for SGD, the first order term in Eq. (6) for SGD is $\langle -\eta_k \sum_{i=0}^{n-1} \nabla f_{\tau_{k,i}}(\boldsymbol{w}_{k,i}), \nabla f_{\tau_{k,i}}(\boldsymbol{w}_{k,0})\rangle \approx \langle -\eta_k \sum_{i=0}^{n-1} \nabla f_{\tau_{k,i}}(\boldsymbol{w}_{k,0}), \nabla f_{\tau_{k,i}}(\boldsymbol{w}_{k,0})\rangle = -\eta_k \|\nabla f_{\tau_{k,i}}(\boldsymbol{w}_{k,0})\|^2$. However, when it comes to bound the second order term, we will encounter two problems. First, it is not known whether the update of SGD is bounded, and thus whether we can bound $L_{loc}$ using $(L_0, L_1)$-smooth condition. Second, even if we succeed to bound $L_{loc}$ by $O(\|\nabla f(\boldsymbol{w}_{k,i})\|) + O(1)$, the second order term has the form $O(\eta_k^2 \|\nabla f(\boldsymbol{w}_{k,i})\|^3) + O(\eta_k^2 \|\nabla f(\boldsymbol{w}_{k,i})\|^2)$, and may be larger than the first order term when the gradient is large. We invite the readers to see the counterexample in Section 4.3 to learn more.

We then provide an example. When the local smoothness constant coefficientL varies drastically in the domain, it is difficult to decide a pre-decide stepsize that guarantees to work along SGD trajectories. There are two cases:

1. When entering a sharp local region, the pre-determined steps might be too large and cause divergence of SGD. 2. When entering a flat region, the pre-determined steps might be too small to

make progress. It will cause slow convergence of SGD. Situation gets worse when using diminishing steps.

For Adam, the effective stepsize involves $\frac{1}{\sqrt{\boldsymbol{\nu}_{k,i}}+\varepsilon\mathbb{1}_d}$, which is adaptive changing along the trajectory. Adam can better handle the above two cases: 1. When entering the sharp local region , the gradient is usually large and $\frac{1}{\sqrt{\boldsymbol{\nu}_{k,i}}+\varepsilon\mathbb{1}_d}$ will gradually decrease in this region. Finally it will reach a small enough stepsize to slip into the sharp region. 2. When entering the flat region, the gradient is consistently small and thus $\frac{1}{\sqrt{\boldsymbol{\nu}_{k,i}}+\varepsilon\mathbb{1}_d}$ will gradually increase, leading to larger stepsize and thus Adam will converge faster.

### B.7 DISCUSSION ON THE POTENTIAL FUNCTION

We notice that similar potential functions have already been applied in the analysis of other momentum-based optimizers, e.g., momentum (S)GD in (Ghadimi et al., 2015) and (Liu et al., 2020b) and Adam-type optimizers (except Adam) in (Chen et al., 2018b). However, extending the proof to Adam is highly-nontrivial and fails to provide a convergence result for Adam. The key difficulty lies in showing that the first order expansion of $f(\boldsymbol{u}_{k,0})$ is positive, which further requires that the adaptive learning rate does not change much within one epoch. This is hard for Adam as the adaptive learning rate of Adam can be nonmonotonic. The lack of L-smooth condition makes the proof even challenging due to the unbounded error brought by gradient norms.

### B.8 DISCUSSION ON THE IN-EXPECTATION RESULT IN (ZHANG ET AL., 2022)

Zhang et al. (2022) handle the misalignment between $\boldsymbol{w}_{k+1,0} - \boldsymbol{w}_{k,0}$ and $-\nabla f(\boldsymbol{w}_{k,0})$ by additionally assuming that the random permutation to generate $\{\tau_{k,0}, \cdots, \tau_{k,n-1}\}$ is uniformly sampled from all possible permutation orders. Based on this additional assumption, the expectation of the momentum can be shown close to the gradient and thus they derive an in-expectation result. In contrast, we do not need such an assumption. Further, their proof is more complex than ours as they need to deal with the historical gradient signal in the momentum. Our potential function allows us to offer a trajectory-wise convergence result with simplified proof.

### B.9 INSIGHT FOR PRACTITIONERS

First, Adam receives great popularity among practitioners (with more than 100k citations). It is important to theoretically understand this current algorithm.

Second, we provide new suggestions for practitioners : When running experiments on tasks such as Transformers and LSTM training, we suggest using Adam instead of SGD. Though this is a folk result based on engineering experience. It is firstly theoretically justified.Third, we provide suggestions for hyperparameter tuning (based on the convergence conditions in Theorem 1): when running Adam, we suggest tune up beta2 and try different $\beta_1 < \sqrt{\beta_2}$. This suggestion would save much effort of grid-searching the $(\beta_1, \beta_2)$ combination.

## C DIVERGING EXAMPLE FOR (S)GD

In this section, we discuss the divergence issue for (S)GD under the $(L_0, L_1)$-smooth condition. This section is organized as follows: we will first consider the simple full-batch case, i.e., $n = 1$ and $f(\boldsymbol{w}) = f_0(\boldsymbol{w})$ (therefore there is no random reshuffling and no randomness in the process of training), in which case we will show GD may suffer from divergence under the $(L_0, L_1)$ assumption while Adam converges to the stationary point. We then step further to the random reshuffling case and show that the randomness may brought additional divergence issues.

### C.1 COMPARE GD AND FULL-BATCH ADAM

To start with, we notice there is an example in Zhang et al. (2019a) over which GD will converge arbitrarily slow. We re-state their result as follows:

**Property 1** (Theorem 4, Zhang et al. (2019a))**.** *For every fixed learning rate $\eta \in \mathbb{R}^+$ and tolerate error bound $\varepsilon \in \mathbb{R}^+$, there exists a domain $\mathcal{X}$, an optimization problem $f : \mathcal{X} \to \mathbb{R}$, and an*

*initialization point $\boldsymbol{w}_1 \in \mathcal{X}$ satisfying $f$ is lower with $(L_0, L_1)$ smoothness and satisfying $M \triangleq \sup\{\|\nabla f(\boldsymbol{w})\| \|\boldsymbol{w}$ such that $f(\boldsymbol{w}) \leq f(\boldsymbol{w}_0)\} < \infty$, such that GD over $f$ starting from $\boldsymbol{w}_1$ with constant learning rate $\eta$, such that any $t \in \mathbb{Z}^+$ smaller than $\frac{L_1 M(f(\boldsymbol{w}_1) - \min_{\boldsymbol{w} \in \mathcal{X}} f(\boldsymbol{w}) - 5\varepsilon/8)}{8\varepsilon^2(\log M + 1)}$ satisfies $\|\nabla f(\boldsymbol{w}_t)\| > \varepsilon$.*

By Property 1, we know that the convergence of GD can be arbitrarily slow depending on the gradient upper bound $M$ of the loss sub-level set $\{\boldsymbol{w} : f(\boldsymbol{w}) \leq f(\boldsymbol{w}_1)\}$, which further depends on the initialization. In other words, the performance of GD sensitively relies on the initialization.

However, Property 1 does not suffice to show Adam performs better than GD, as in Property 1 the learning rate is constant while in Theorem 1, it is decaying. To fill this gap in setting, we provide the following example satisfying Assumptions 2 and 1, on which GD with decaying learning rate diverges if initialized badly.

**Example 3** (Example 1, restated). *Let $n = 1$ and $p = 2$ with $f_0((w^1, w^2)) = e^{-\frac{\sqrt{3}}{2}w^1 - \frac{1}{2}w^2} + e^{\frac{\sqrt{3}}{2}w^1 - \frac{1}{2}w^2}$ and $f((w^1, w^2)) = f_0((w^1, w^2))$. Then, $f$ satisfies Assumptions 1 and 2 . However, for every initial learning rate $\eta_1$, there exists a region $\mathcal{E}$ with infinite measure over $\mathbb{R}^2$ such that, with GD with decaying learning rate $\eta_k = \frac{\eta_1}{\sqrt{k}}$ and start point in $\mathcal{E}$, $f$ and $\nabla f$ diverge to infinity.*

*Proof.* **Notations.** To simplify the notation, we define $\boldsymbol{x}_1 = (\frac{\sqrt{3}}{2}, \frac{1}{2})$ and $\boldsymbol{x}_2 = (-\frac{\sqrt{3}}{2}, \frac{1}{2})$. It can be observed that $\|\boldsymbol{x}_1\| = \|\boldsymbol{x}_2\| = 1$ and $\langle \boldsymbol{x}_1, \boldsymbol{x}_2 \rangle = -\frac{1}{2}$. We also denote the parameter given by the $k$-th iteration of GD as $\boldsymbol{w}_k$. By definition, we have

$$\boldsymbol{w}_{k+1} = \boldsymbol{w}_k + \eta_k e^{-\langle \boldsymbol{x}_1, \boldsymbol{w}_k \rangle} \boldsymbol{x}_1 + \eta_k e^{-\langle \boldsymbol{x}_2, \boldsymbol{w}_k \rangle} \boldsymbol{x}_2.$$

We further define $g(x) \triangleq \eta_1 e^{\frac{3}{8}x} - 17x - 16\eta_1 e^{-\frac{15}{16}x}$. As $g(x) \to \infty$ as $x \to \infty$, there exists a large enough positive constant $C > 1$, such that for every $x > C$, $g(x) > 0$.

**Verify the assumptions.** Assumption 2 immediately follows as $n = 1$. We then check Assumption 1. We have

$$\nabla f(\boldsymbol{w}) = -e^{-\langle \boldsymbol{w}, \boldsymbol{x}_1 \rangle} \boldsymbol{x}_1 - e^{-\langle \boldsymbol{w}, \boldsymbol{x}_2 \rangle} \boldsymbol{x}_2, \nabla^2 f(\boldsymbol{w}) = \boldsymbol{x}_1^\top \boldsymbol{x}_1 e^{-\langle \boldsymbol{w}, \boldsymbol{x}_1 \rangle} + \boldsymbol{x}_2^\top \boldsymbol{x}_2 e^{-\langle \boldsymbol{w}, \boldsymbol{x}_2 \rangle}.$$

Therefore, $\|\nabla^2 f_0(\boldsymbol{w})\|_2 \leq e^{-\langle \boldsymbol{w}, \boldsymbol{x}_1 \rangle} + e^{-\langle \boldsymbol{w}, \boldsymbol{x}_2 \rangle}$, while

$$\|\nabla f(\boldsymbol{w})\| \geq \langle -e^{-\langle \boldsymbol{w}, \boldsymbol{x}_1 \rangle} \boldsymbol{x}_1 - e^{-\langle \boldsymbol{w}, \boldsymbol{x}_2 \rangle} \boldsymbol{x}_2, (0, 1) \rangle \geq \frac{1}{2} e^{-\langle \boldsymbol{w}, \boldsymbol{x}_1 \rangle} + \frac{1}{2} e^{-\langle \boldsymbol{w}, \boldsymbol{x}_2 \rangle}.$$

As a conclusion, Assumption 1 is satisfied with $L_0 = 0$ and $L_1 = 2$.

**Initialization.** We define region $\mathcal{E}$ as

$$\mathcal{E} \triangleq \{a_1 \boldsymbol{x}_1 + \boldsymbol{b_1} : a_1 > C, a_1 > 16\|\boldsymbol{b_1}\|\} \cup \{a_1 \boldsymbol{x}_2 + \boldsymbol{b_1} : a_1 > C, a_1 > 16\|\boldsymbol{b_1}\|\}.$$

It should be noticed that $\mathcal{E}$ has infinite measure.

**Iteration.** As $\{a_1 \boldsymbol{x}_1 + \boldsymbol{b_0} : a_1 > C, a_1 > 16\|\boldsymbol{b_1}\|\}$ is symmetric with $\{a_1 \boldsymbol{x}_2 + \boldsymbol{b_1} : a_1 > C, a_1 > 16\|\boldsymbol{b_1}\|\}$, we analyze the case when $\boldsymbol{w}_1$ is in the former set without loss of generality, i.e., $\boldsymbol{w}_1 = a_1 \boldsymbol{x}_1 + \boldsymbol{b_1}$ with $a_1 > 0$ and $a_1 > 16\|\boldsymbol{b_1}\|$.

We will prove by induction that $\forall k \geq 1$, $\boldsymbol{w}_k = a_k \boldsymbol{x}_{i_k} + \boldsymbol{b_k}$, where $i_k$ equals to 1 if $k$ is odd and 2 if $k$ is even. $a_k$ and $\boldsymbol{b_k}$ satisfies $a_k \geq 16\|\boldsymbol{b_k}\|$ and $a_k \geq 17^{k-1}C$.

For $k = 1$, the proof directly follows by the initialization. For $k \neq 1$, suppose that the above property holds for all the iterations before the $k$-th iteration. Suppose $k$ is even. Then,

$$\boldsymbol{w}_{k-1} = a_{k-1} \boldsymbol{x}_1 + \boldsymbol{b_{k-1}},$$

which leads to

$$\|\boldsymbol{w}_{k-1}\| \leq a_{k-1} + \|\boldsymbol{b_{k-1}}\| \leq \frac{17}{16} a_{k-1},$$

$$\langle \boldsymbol{w}_{k-1}, \boldsymbol{x}_1 \rangle \geq a_{k-1} - \|\boldsymbol{b_{k-1}}\| \geq \frac{15}{16} a_{k-1},$$

and

$$\langle \boldsymbol{w}_{k-1}, \boldsymbol{x}_2 \rangle \leq -\frac{1}{2}a_{k-1} + \|\boldsymbol{b_{k-1}}\| \leq -\frac{7}{16}a_{k-1}.$$

Consequently,

$$\begin{aligned}
\boldsymbol{w}_k =& \boldsymbol{w}_{k-1} + \eta_{k-1}e^{-\langle \boldsymbol{x}_1, \boldsymbol{w}_{k-1}\rangle}\boldsymbol{x}_1 + \eta_{k-1}e^{-\langle \boldsymbol{x}_2, \boldsymbol{w}_{k-1}\rangle}\boldsymbol{x}_2 \\
\leq& \eta_{k-1}e^{-\langle \boldsymbol{x}_2, \boldsymbol{w}_{k-1}\rangle}\boldsymbol{x}_2 + \boldsymbol{w}_{k-1} + \eta_{k-1}e^{-\langle \boldsymbol{x}_1, \boldsymbol{w}_{k-1}\rangle}\boldsymbol{x}_1.
\end{aligned}$$

The norm of $\boldsymbol{w}_{k-1} + \eta_{k-1}e^{-\langle \boldsymbol{x}_1, \boldsymbol{w}_{k-1}\rangle}\boldsymbol{x}_1$ can be bounded as

$$\|\boldsymbol{w}_{k-1} + \eta_{k-1}e^{-\langle \boldsymbol{x}_1, \boldsymbol{w}_{k-1}\rangle}\boldsymbol{x}_1\| \leq \|\boldsymbol{w}_{k-1}\| + \eta_{k-1}e^{-\langle \boldsymbol{x}_1, \boldsymbol{w}_{k-1}\rangle} \leq \frac{17}{16}a_{k-1} + \eta_{k-1}e^{-\frac{15}{16}a_{k-1}},$$

while the coefficient of $\boldsymbol{x}_2$ can be lower bounded as

$$\eta_{k-1}e^{-\langle \boldsymbol{x}_2, \boldsymbol{w}_{k-1}\rangle} \geq \eta_{k-1}e^{\frac{7}{16}a_{k-1}} \geq \eta_1 e^{\frac{7}{16}a_{k-1}-\frac{1}{2}\log k - 1} \geq \eta_1 e^{\frac{3}{8}a_{k-1}+\frac{1}{16}17^{k-1}C-\frac{1}{2}(k-2)} \geq \eta_1 e^{\frac{3}{8}a_{k-1}}.$$

Therefore,

$$\eta_{k-1}e^{-\langle \boldsymbol{x}_2, \boldsymbol{w}_{k-1}\rangle} - 16\|\boldsymbol{w}_{k-1} + \eta_{k-1}e^{-\langle \boldsymbol{x}_1, \boldsymbol{w}_{k-1}\rangle}\boldsymbol{x}_1\| \geq \eta_1 e^{\frac{3}{8}a_{k-1}} - 17a_{k-1} + 16\eta_{k-1}e^{-\frac{15}{16}a_{k-1}}$$
$$\geq g(a_{k-1}).$$

As by the induction condition $a_{k-1} \geq 17^{k-1}C \geq C$, we have that $g(a_{k-1}) \geq 0$. Therefore, choosing $a_k = \eta_{k-1}e^{-\langle \boldsymbol{x}_2, \boldsymbol{w}_{k-1}\rangle}$ and $\boldsymbol{b_k} = \|\boldsymbol{w}_{k-1} + \eta_{k-1}e^{-\langle \boldsymbol{x}_1, \boldsymbol{w}_{k-1}\rangle}\boldsymbol{x}_1\|$ completes the proof for $k$.

The proof of the case that $k$ is odd follows the same routine by simply exchanging $\boldsymbol{x}_1$ and $\boldsymbol{x}_2$ in the above proof.

Consequently, for $\|\nabla f(\boldsymbol{w}_k)\|$, we have that when $k$ is even

$$\|\nabla f(\boldsymbol{w}_k)\| \geq \|\boldsymbol{x}_1 e^{-\langle \boldsymbol{x}_2, \boldsymbol{w}_k\rangle}\| - \|\boldsymbol{x}_2 e^{-\langle \boldsymbol{x}_1, \boldsymbol{w}_k\rangle}\| \geq e^{\frac{5}{16}a_k} - e^{-\frac{15}{16}a_k} \to \infty.$$

Similar claim holds for the case $k$ is odd. We then complete the proof that $\|\nabla f(\boldsymbol{w}_k)\| \to \infty$ as $k \to \infty$. The claim that $f(\boldsymbol{w}_k) \to \infty$ as $k \to \infty$ can be derived following the same routine.

The proof is completed. □

On the other hand, as $f$ in Example 1 satisfies Assumption 1 and Assumption 2 (with $D_0 = 0$), Theorem 1 applies and full batch Adam will converge to a stationary point regardless of the initialization. Example 1 indicates that even for the full-batch case, Adam appears to be less sensitive to the initialization compared with GD under $(L_0, L_1)$-smooth condition.

However, under the same setting and fixing the initialization point, tuning down the initial learning rate $\eta_1$ may help GD converge. Specifically, when gradient norm of the loss sub-level set $\{\boldsymbol{w} : f(\boldsymbol{w}) \leq f(\boldsymbol{w}_0)\}$ is upper bounded, if we choose the learning rate to be sufficiently small (for example, smaller than $2/(L_0 + L_1 M)$[1] where $M$ is the gradient norm upper bound of the loss sub-level set), we can make the loss keeps decreasing as GD iterates, and the parameter stays in the loss sub-level set. However, such a learning rate can be rather small when $M$ is large and leads to a slow convergence of GD, which is exactly the intuition behind Property 1.

Furthermore, when $n \geq 2$ and SGD is adopted, tuning down the learning rate may not help any more. We will discuss such an issue in the following section.

### C.2 Compare SGD and Adam

First of all, Example 3 can be simply extended to provide a divergent example for SGD.

---

[1]$2/(L_0 + L_1 M)$ serves as the largest reasonable setting of the learning rate for GD, as $f$ over the loss sub-level set has smoothness upper bound $(L_0 + L_1 M)$.

**Example 4** (Example 1, restated). *Let $n = 2$ and $p = 2$ with $f_0((w^1, w^2)) = e^{-\frac{\sqrt{3}}{2}w^1 - \frac{1}{2}w^2}$, $f_1(x) = e^{\frac{\sqrt{3}}{2}w^1 - \frac{1}{2}w^2}$, and $f((w^1, w^2)) = f_0((w^1, w^2))$. Then, $f$ satisfies Assumptions 1 and 2 (with $D_0 = 0$). However, for every initial learning rate $\eta_1$, there exists a region $\mathcal{E}$ with infinite measure over $\mathbb{R}^2$ such that, with SGD with decaying learning rate $\eta_k = \frac{\eta_1}{\sqrt{k}}$ and start point in $\mathcal{E}$, $f$ and $\nabla f$ diverge to infinity.*

The proof follows the similar routine as Example 3 and we omit it here. Note as $D_0 = 0$ in this case, Adam still converges to the stationary point.

On the other hand, as discussed in the previous section, for SGD over $(L_0, L_1)$-smooth condition, a small learning rate may no longer help and SGD will still diverge. We illustrate this idea by consider the following example.

**Example 5** (Diverging Example for SGD with Small Learning Rate). *Specifically, we rescale the $\boldsymbol{x}_1$ and $\boldsymbol{x}_2$ in Example 3 as $\tilde{\boldsymbol{x}}_1 = \boldsymbol{x}_1$ and $\tilde{\boldsymbol{x}}_2 = 50\boldsymbol{x}_2$. We choose initialization point $\boldsymbol{w}_1 = \tilde{\boldsymbol{x}}_1$, and choose learning rate as $\eta_1 = 1/L$ with decaying learning rate $\eta_t = \frac{\eta_1}{\sqrt{t}}$, where $L$ is defined as the smoothness upper bound of the loss sub-level set, i.e.,*

$$L \triangleq \arg\max \left\{ \|\nabla^2 f(\boldsymbol{w})\|_2 : f(\boldsymbol{w}) \le f(\boldsymbol{w}_1) \right\}.$$

By the definition of the previous section, GD on such an example will converge to the stationary point, as the iterates will stay in the sub-level set, the smoothness will be small, and the loss will decrease. However, we run SGD on this example and find that the loss and gradient norm will explode in three steps (Figure 3).

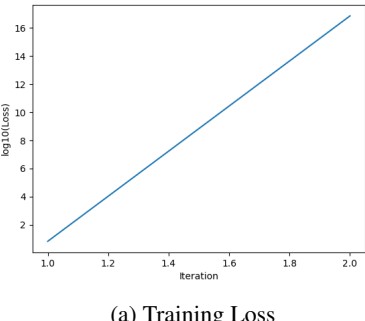
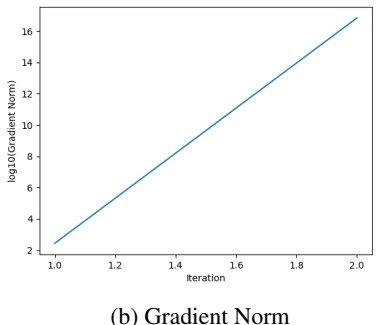

(a) Training Loss          (b) Gradient Norm

Figure 3: Example on which SGD with decaying learning rate and small initial learning rate diverges. We only plot the first two steps because the values exceed the math range in python in the third step.

The intuition behind Example 5 is that the update calculate from a batch in SGD does not essentially reduce the loss and may no longer keeps the iterates stay in the loss sub-level set. This can eventually make the loss diverging, or keep the parameter away from the saddle point in the training of neural networks.

## D    PROOF OF THEOREM 1

This appendix provides the formal proof of Theorem 1. Specifically, we will first make some preparations by ① showing notations and ② proving lemmas which characterize several basic properties of the Adam optimizer, and then prove Theorem 1 based on the lemmas.

**Remark 10.** *In the remaining proof of this paper, we assume **without the loss of generality that** $\eta_1$ **is small enough**, such that the following requirements are fulfilled (with notations explained latter):*

- *$2C_2\sqrt{d}\eta_1 \le \frac{1}{L_1}$. This will latter ensure that we can directly apply the definition of $(L_0, L_1)$-smooth condition (Assumption 1) to parameter sequence $\{\boldsymbol{w}_{k,i}\}_{k,i}$;*

- $\frac{1}{4(2\sqrt{2}+1)} \geq \sqrt{D_1}C_{11}\eta_1$. *This will latter ensure the second-order term is smaller than the first-order term at the end of the proof.*

*The proof can be easily extended to general cases (while certainly more cumbersome) by selecting large enough $K$ and using the epoch $K$ as a new start point and derive the results after epoch $K$ (this is due to $\eta_k$ is decaying, and $K$ is finite, the epochs before epoch $k$ can be uniformly bounded and we then derive the desired result for all epochs).*

*Without the loss of generality, we also take the following initialization: $\boldsymbol{w}_{1,0} = \boldsymbol{w}_0$, $\boldsymbol{m}_{1,-1} = \nabla f_{\tau_{1,-1}}(\boldsymbol{w}_0)$ ($\tau_{1,-1}$ can be any integer in $[0, n-1]$), and $\boldsymbol{\nu}_{l,1,-1} = \max_j\{\partial_l f_j(\boldsymbol{w}_0)^2\}$, $\forall l$ (here the maximum is taken component-wisely). We take the initialization to have a more concise proof, while the proof can be easily extended to all the initialization as the information of the initialization in the exponentially decayed average of Adam (both in $\boldsymbol{m}_{k,i}$ and $\boldsymbol{\nu}_{k,i}$) decays rapidly with $k$ increasing.*

## D.1 PRELIMINARIES

### D.1.1 NOTATIONS

Here we provide a complete list of notations used in the appendix for a clear reference.

- We use $(k_1, i_1) \leq (<)(k_2, i_2)$ for $\forall k_1, k_2 \in \mathbb{N}^+$ and $i_1, i_2 \in \{0, \cdots, n-1\}$, if either $k_1 < k_2$ or $k_1 = k_2$ and $i_1 \leq (<)i_2$

- We define function $g(x) : [0, 1] \to \mathbb{R}^{/-}$ as

$$g(\beta_2) \triangleq \max\left\{ \frac{1}{\sqrt{\beta_2^{n-1}}} - 1, 1 - \frac{1}{\sqrt{\beta_2^{n-1} + 8n\frac{1-\beta_2^{n-1}}{\beta_2^n}}}, 1 - \sqrt{\beta_2}, \sqrt{\frac{\beta_2}{\left(1 - (1-\beta_2)\frac{2n}{\beta_2^n}\right)} - 1} \right\}.$$

- We define constants $\{C_i\}_{i=1}^{10}$ as follows:

$$C_1 \triangleq \frac{(1-\beta_1)^2}{1-\beta_2} \frac{1}{1 - \frac{\beta_1^2}{\beta_2}} + 1,$$

$$C_2 \triangleq nC_1 + \frac{\beta_1}{1-\beta_1}C_1\left(1 + \sqrt{2}\right),$$

$$C_3 \triangleq C_1\left(n(L_0 + L_1\sqrt{D_0}) + 2\sqrt{2}(L_0 + L_1\sqrt{D_0})\frac{\sqrt{1-\beta_2}}{1-\sqrt{\beta_2}}\frac{\sqrt{\beta_2}}{1-\sqrt{\beta_2}} + 8\sqrt{2n}L_0\frac{1}{1-\beta_2^n}\right),$$

$$C_4 \triangleq 4L_1C_1\sqrt{D_1}\frac{\sqrt{1-\beta_2}}{1-\sqrt{\beta_2}}$$

$$C_5 \triangleq n^2(1 + n\sqrt{d}C_1\eta_1 L_1\sqrt{n}\sqrt{D_1})\left(C_4 + \frac{dC_4\sqrt{D_1}}{1-\sqrt{\beta_2^n}}\right),$$

$$C_6 \triangleq \left(dC_3 + \frac{C_4n\sqrt{D_1}}{1-\sqrt{\beta_2^n}}\right)\eta_1^2,$$

$$C_7 \triangleq 3n\left(C_4 + \frac{dC_4}{1-\sqrt{\beta_2^n}}\right)\left(nL_0 + L_1\sqrt{n}\sqrt{D_0}\right)n^2\sqrt{d}C_1\eta_1^3 + \left(dC_3 + \frac{C_2C_4n\sqrt{D_1}}{1-\sqrt{\beta_2^n}}\right)\eta_1^2,$$

$$C_8 \triangleq \sqrt{\frac{2n^2}{\beta_2^n}}L_1\sqrt{D_1}n\sqrt{n} + dg(\beta_2)\left(n - 1 + \frac{1+\beta_1}{1-\beta_1}\right)\frac{\sqrt{2}n}{\beta_2^{\frac{n}{2}}}L_1C_1\sqrt{D_1}\left(1 + \frac{1}{1-\beta_2^n}\right)(n + n^{\frac{5}{2}}\sqrt{d}C_1\eta_1 L_1\sqrt{D_1}) + 2\frac{\beta_1}{(1-\beta_1)\eta_1}\sqrt{d}C_1,$$

$$C_9 \triangleq \sqrt{\frac{2n^2}{\beta_2^n}}d(n^2L_0 + n\sqrt{n}L_1\sqrt{D_0})C_1\eta_1^2 + g(\beta_2)\left(n - 1 + \frac{1+\beta_1}{1-\beta_1}\right)\frac{\sqrt{2}n}{\beta_2^{\frac{n}{2}}}\left(n + \frac{2\sqrt{2}\beta_1}{1-\beta_1}\right)C_1(L_0 + L_1\sqrt{D_0})d\sqrt{d}\eta_1^2,$$

$$C_{10} \triangleq 3dg(\beta_2)\left(n - 1 + \frac{1+\beta_1}{1-\beta_1}\right)\frac{\sqrt{2}n}{\beta_2^{\frac{n}{2}}}L_1C_1\sqrt{D_1}\left(1 + \frac{1}{1-\beta_2^n}\right)n\left(nL_0 + L_1\sqrt{n}\sqrt{D_0}\right)n\sqrt{d}C_1\eta_1^3 + C_9,$$

$$C_{11} \triangleq (\frac{1}{2} + C_2)C_5 + C_8 + \frac{3L_1\sqrt{n}\sqrt{D_1}C_2^2d}{2},$$

$$C_{12} \triangleq (\frac{1}{2} + C_2)C_6 + C_9 + \frac{nL_0 + L_1\sqrt{n}\sqrt{D_0}}{2}3C_2^2d\eta_1^2,$$

$$C_{13} \triangleq (\frac{1}{2} + C_2)C_7 + C_{10} + \frac{nL_0 + L_1\sqrt{n}\sqrt{D_0}}{2}3C_2^2d\eta_1^2.$$

### D.1.2 Auxiliary Lemmas

Here we provide auxiliary lemmas describing basic properties of Adam and the descent lemma under $(L_0, L_1)$-smooth condition.

**Smoothness of $f$**

**Lemma 2.** *With Assumptions 2 and 1, $f$ satisfies $(nL_0 + L_1\sqrt{n}\sqrt{D_0}, L_1\sqrt{n}\sqrt{D_1})$-smooth condition.*

*Proof.* $\forall \boldsymbol{w}_1, \boldsymbol{w}_2 \in \mathbb{R}^d$ satisfying $\|\boldsymbol{w}_1 - \boldsymbol{w}_2\| \leq \frac{1}{L_1}$,

$$\|\nabla f(\boldsymbol{w}_1) - \nabla f(\boldsymbol{w}_2)\| \leq \sum_{i=0}^{n-1} \|\nabla f_i(\boldsymbol{w}_1) - \nabla f_i(\boldsymbol{w}_2)\| \leq \sum_{i=0}^{n-1}(L_0 + L_1\|\nabla f_i(\boldsymbol{w}_1)\|)\|\boldsymbol{w}_1 - \boldsymbol{w}_2\|$$

$$\leq \left( nL_0 + L_1\sqrt{n}\sqrt{\sum_{i=0}^{n-1} \|\nabla f_i(\boldsymbol{w}_1)\|^2} \right) \|\boldsymbol{w}_1 - \boldsymbol{w}_2\|$$

$$\leq (nL_0 + L_1\sqrt{n}\sqrt{D_0 + D_1\|\nabla f(\boldsymbol{w}_1)\|^2})\|\boldsymbol{w}_1 - \boldsymbol{w}_2\|$$

$$\leq (nL_0 + L_1\sqrt{n}(\sqrt{D_0} + \sqrt{D_1}\|\nabla f(\boldsymbol{w}_1)\|))\|\boldsymbol{w}_1 - \boldsymbol{w}_2\|$$

$$\leq (nL_0 + L_1\sqrt{n}\sqrt{D_0} + L_1\sqrt{n}\sqrt{D_1}\|\nabla f(\boldsymbol{w}_1)\|)\|\boldsymbol{w}_1 - \boldsymbol{w}_2\|.$$

The proof is completed. $\qquad\square$

**Basic Properties of Adam**

The following lemma characterizes the update norm of Adam.

**Lemma 3** (Bounded Update). *If $\beta_1 < \sqrt{\beta_2}$, we have $\forall k \in \mathbb{N}^+$, $i \in \{0, \cdots, n-1\}$,*

$$\frac{|\boldsymbol{m}_{l,k,i}|}{\sqrt{\boldsymbol{\nu}_{l,k,i}} + \varepsilon} \leq C_1,$$

*where*

$$C_1 \triangleq \frac{(1-\beta_1)^2}{1-\beta_2} \frac{1}{1 - \frac{\beta_1^2}{\beta_2}} + 1.$$

*Furthermore, we have $|\boldsymbol{w}_{l,k,i+1} - \boldsymbol{w}_{l,k,i}| \leq C_1\eta_k$, and thus $\|\boldsymbol{w}_{k,i+1} - \boldsymbol{w}_{k,i}\| \leq C_1\eta_k\sqrt{d}$.*

*Proof.* By the definition of $\boldsymbol{m}_{k,i}$, we have

$$
(\boldsymbol{m}_{l,k,i})^2
$$

$$
= \left( (1-\beta_1) \sum_{j=0}^{i} \beta_1^{(k-1)n+i-((k-1)n+j)} \partial_l f_{\tau_{k,j}}(\boldsymbol{w}_{k,j}) \right.
$$

$$
\left. + (1-\beta_1) \sum_{m=1}^{k-1} \sum_{j=0}^{n-1} \beta_1^{(k-1)n+i-((m-1)n+j)} \partial_l f_{\tau_{m,j}}(\boldsymbol{w}_{m,j}) + \beta_1^{(k-1)n+i+1} \partial_l f_{\tau_{1,-1}}(\boldsymbol{w}_{1,0}) \right)^2
$$

$$
\leq \left( (1-\beta_1) \sum_{j=0}^{i} \beta_1^{(k-1)n+i-((k-1)n+j)} |\partial_l f_{\tau_{k,j}}(\boldsymbol{w}_{k,j})| \right.
$$

$$
\left. + (1-\beta_1) \sum_{m=1}^{k-1} \sum_{j=0}^{n-1} \beta_1^{(k-1)n+i-((m-1)n+j)} |\partial_l f_{\tau_{m,j}}(\boldsymbol{w}_{m,j})| + \beta_1^{(k-1)n+i+1} \max_{s\in[n]} |\partial_l f_s(\boldsymbol{w}_{1,0})| \right)^2
$$

$$
\overset{(\star)}{\leq} \left( (1-\beta_2) \sum_{j=0}^{i} \beta_2^{(k-1)n+i-((k-1)n+j)} |\partial_l f_{\tau_{k,j}}(\boldsymbol{w}_{k,j})|^2 \right.
$$

$$
\left. + (1-\beta_2) \sum_{m=1}^{k-1} \sum_{j=0}^{n-1} \beta_2^{(k-1)n+i-((m-1)n+j)} |\partial_l f_{\tau_{m,j}}(\boldsymbol{w}_{m,j})|^2 + \beta_2^{(k-1)n+i+1} \max_{s\in[n]} |\partial_l f_s(\boldsymbol{w}_{1,0})|^2 \right)
$$

$$
\cdot \left( \frac{(1-\beta_1)^2}{1-\beta_2} \sum_{j=0}^{(k-1)n+i} \left( \frac{\beta_1^2}{\beta_2} \right)^j + \left( \frac{\beta_1^2}{\beta_2} \right)^{(k-1)n+i+1} \right)
$$

$$
\overset{(*)}{=} \left( \frac{(1-\beta_1)^2}{1-\beta_2} \sum_{j=0}^{(k-1)n+i} \left( \frac{\beta_1^2}{\beta_2} \right)^j + \left( \frac{\beta_1^2}{\beta_2} \right)^{(k-1)n+i+1} \right) \boldsymbol{\nu}_{l,k,i}
$$

$$
\leq \left( \frac{(1-\beta_1)^2}{1-\beta_2} \frac{1}{1-\frac{\beta_1^2}{\beta_2}} + 1 \right) \boldsymbol{\nu}_{l,k,i} = C_1 \boldsymbol{\nu}_{l,k,i},
$$

where Eq. $(\star)$ is due to the Cauchy-Schwartz's Inequality, and Eq. $(*)$ is due to the definition of $\boldsymbol{\nu}_{l,1,-1}$. We complete the proof of the first claim. The second claim then follows directly from the update rule

$$
\boldsymbol{w}_{l,k,i+1} - \boldsymbol{w}_{l,k,i} = \eta_k \frac{\boldsymbol{m}_{l,k,i}}{\sqrt{\boldsymbol{\nu}_{l,k,i}} + \varepsilon}.
$$

The proof is completed. $\qquad\square$

Based on Lemma 3, we then provide estimations for the norms of the momentum and the adaptor.

**Lemma 4** (Estimation of the norm of the momentum). *We have for all $l \in [d], k \in \mathbb{Z}^+, i \in [n]$,*

$$
|\boldsymbol{m}_{l,k,i}| \leq \max_{i'\in[n]} |\partial_l f_{i'}(\boldsymbol{w}_{k,0})| + \left( n + \frac{2\sqrt{2}\beta_1}{1-\beta_1} \right) C_1 (L_0 + L_1\sqrt{D_0}) \sqrt{d}\eta_k + L_1 C_1 \sqrt{D_1} \eta_k \sum_{j=0}^{i-1} \|\nabla f(\boldsymbol{w}_{k,j})\|
$$

$$
+ L_1 C_1 \sqrt{D_1} \sum_{t=1}^{k-1} \eta_{k-t} \sum_{j=0}^{n-1} \beta_1^{tn+i-j} \|\nabla f(\boldsymbol{w}_{k-t,j})\|.
$$

*Similarly, $l \in [d], k \in \mathbb{Z}^+/\{1\}$,*

$$
|\boldsymbol{m}_{l,k-1,n-1}| \leq \max_{i'\in[n]} |\partial_l f_{i'}(\boldsymbol{w}_{k,0})| + \sum_{t=1}^{k-1} \sum_{j=0}^{n-1} \beta_1^{tn-1-j} C_1 \eta_{k-t} \sqrt{d} L_1 \sqrt{D_1} \|\nabla f(\boldsymbol{w}_{k-t,j})\| + \frac{2\sqrt{2}(L_0 + L_1\sqrt{D_0})C_1\sqrt{d}\eta_k}{1-\beta_1}.
$$

*Proof.* To begin with, for any $t \in [k-1]$ and any $j \in [0, n-1]$, we have the following estimation for $\partial_l f_i(\boldsymbol{w}_{k-t,j})$:

$$
|\partial_l f_i(\boldsymbol{w}_{k-t,j})|
$$
$$
\overset{(\star)}{\leq} |\partial_l f_i(\boldsymbol{w}_{k,0})| + \sum_{p=j}^{n-1} |\partial_l f_i(\boldsymbol{w}_{k-t,p}) - \partial_l f_i(\boldsymbol{w}_{k-t,p+1})| + \sum_{r=1}^{t-1}\sum_{p=0}^{n-1} |\partial_l f_i(\boldsymbol{w}_{k-r,p}) - \partial_l f_i(\boldsymbol{w}_{k-r,p+1})|
$$
$$
\leq |\partial_l f_i(\boldsymbol{w}_{k,0})| + \sum_{p=j}^{n-1}(L_0 + L_1\|\nabla f_i(\boldsymbol{w}_{k-t,p})\|)\|\boldsymbol{w}_{k-t,p} - \boldsymbol{w}_{k-t,p+1}\|
$$
$$
+ \sum_{r=1}^{t-1}\sum_{p=0}^{n-1}(L_0 + L_1\|\nabla f_i(\boldsymbol{w}_{k-r,p})\|)\|\boldsymbol{w}_{k-r,p} - \boldsymbol{w}_{k-r,p+1}\|
$$
$$
\leq |\partial_l f_i(\boldsymbol{w}_{k,0})| + \sum_{p=j}^{n-1}(L_0 + L_1\|\nabla f_i(\boldsymbol{w}_{k-t,p})\|)C_1\eta_{k-t}\sqrt{d} + \sum_{r=1}^{t-1}\sum_{p=0}^{n-1}(L_0 + L_1\|\nabla f_i(\boldsymbol{w}_{k-r,p})\|)C_1\eta_{k-r}\sqrt{d}
$$
$$
\leq |\partial_l f_i(\boldsymbol{w}_{k,0})| + \sum_{p=j}^{n-1}\left(L_0 + L_1\sqrt{\sum_{i'\in[n]}\|\nabla f_{i'}(\boldsymbol{w}_{k-t,p})\|^2}\right)C_1\eta_{k-t}\sqrt{d}
$$
$$
+ \sum_{r=1}^{t-1}\sum_{p=0}^{n-1}\left(L_0 + L_1\sqrt{\sum_{i'\in[n]}\|\nabla f_{i'}(\boldsymbol{w}_{k-r,p})\|^2}\right)C_1\eta_{k-r}\sqrt{d}
$$
$$
\leq |\partial_l f_i(\boldsymbol{w}_{k,0})| + \sum_{p=j}^{n-1}\left(L_0 + L_1\sqrt{D_1}\|\nabla f(\boldsymbol{w}_{k-t,p})\| + L_1\sqrt{D_0}\right)C_1\eta_{k-t}\sqrt{d}
$$
$$
+ \sum_{r=1}^{t-1}\sum_{p=0}^{n-1}\left(L_0 + L_1\sqrt{D_1}\|\nabla f(\boldsymbol{w}_{k-r,p})\| + L_1\sqrt{D_0}\right)C_1\eta_{k-r}\sqrt{d}
$$
$$
\overset{(*)}{\leq} |\partial_l f_i(\boldsymbol{w}_{k,0})| + \sum_{p=j}^{n-1}L_1\sqrt{D_1}\|\nabla f(\boldsymbol{w}_{k-t,p})C_1\eta_{k-t}\sqrt{d} + \sum_{r=1}^{t-1}\sum_{p=0}^{n-1}L_1\sqrt{D_1}\|\nabla f(\boldsymbol{w}_{k-r,p})\|C_1\eta_{k-r}\sqrt{d}
$$
$$
+ 2(L_0 + L_1\sqrt{D_0})C_1\sqrt{d}\eta_{k-1}(tn-j)
$$
$$
\leq |\partial_l f_i(\boldsymbol{w}_{k,0})| + \sum_{p=j}^{n-1}L_1\sqrt{D_1}\|\nabla f(\boldsymbol{w}_{k-t,p})C_1\eta_{k-t}\sqrt{d} + \sum_{r=1}^{t-1}\sum_{p=0}^{n-1}L_1\sqrt{D_1}\|\nabla f(\boldsymbol{w}_{k-r,p})\|C_1\eta_{k-r}\sqrt{d}
$$
$$
+ 2\sqrt{2}(L_0 + L_1\sqrt{D_0})C_1\sqrt{d}\eta_k(tn-j).
$$

where Inequality $(\star)$ is due to $(L_0, L_1)$-smooth condition, and Inequality $(*)$ is due to $\forall a, b \in \mathbb{N}^+, a > b, \sum_{i=0}^{b}\frac{1}{\sqrt{a-i}} \leq 2\frac{b+1}{a}$.

Similarly, we have that for any $j \in [0, n-1]$,

$$
|\partial_l f_i(\boldsymbol{w}_{k,j})| \leq |\partial_l f_i(\boldsymbol{w}_{k,0})| + \sum_{p=0}^{j-1}|\partial_l f_i(\boldsymbol{w}_{k,p+1}) - \partial_l f_i(\boldsymbol{w}_{k,p})|
$$
$$
\leq |\partial_l f_i(\boldsymbol{w}_{k,0})| + \sum_{p=0}^{j-1}\left(L_0 + L_1\sqrt{D_1}\|\nabla f(\boldsymbol{w}_{k,p})\| + L_1\sqrt{D_0}\right)C_1\eta_k\sqrt{d}
$$
$$
= |\partial_l f_i(\boldsymbol{w}_{k,0})| + \sum_{p=0}^{j-1}L_1\sqrt{D_1}\|\nabla f(\boldsymbol{w}_{k,p})\|C_1\eta_k\sqrt{d} + j(L_0 + L_1\sqrt{D_0})C_1\sqrt{d}\eta_k.
$$

Therefore, the norm of $m_{l,k,i}$ can be bounded as

$$|m_{l,k,i}|$$

$$\leq (1-\beta_1)\sum_{j=0}^{i}\beta_1^{(k-1)n+i-((k-1)n+j)}|\partial_l f_{\tau_{k,j}}(\boldsymbol{w}_{k,j})| + (1-\beta_1)\sum_{t=1}^{k-1}\sum_{j=0}^{n-1}\beta_1^{tn+i-j}|\partial_l f_{\tau_{k-t,j}}(\boldsymbol{w}_{k-t,j})|$$

$$+ \beta_1^{(k-1)n+i+1}|\partial_l f_{\tau_{1,0}}(\boldsymbol{w}_{1,0})|$$

$$\leq (1-\beta_1)\sum_{j=0}^{i}\beta_1^{(k-1)n+i-((k-1)n+j)}|\partial_l f_{\tau_{k,j}}(\boldsymbol{w}_{k,0})| + (1-\beta_1)\sum_{t=1}^{k-1}\sum_{j=0}^{n-1}\beta_1^{tn+i-j}|\partial_l f_{\tau_{k-t,j}}(\boldsymbol{w}_{k,0})|$$

$$+ \beta_1^{(k-1)n+i+1}|\partial_l f_{\tau_{1,0}}(\boldsymbol{w}_{k,0})|$$

$$+ (1-\beta_1)\sum_{j=0}^{i}\beta_1^{(k-1)n+i-((k-1)n+j)}\left(\sum_{p=0}^{j-1}C_1\eta_k\sqrt{d}L_1\sqrt{D_1}\|\nabla f(\boldsymbol{w}_{k,p})\| + (L_0+L_1\sqrt{D_0})C_1\eta_k\sqrt{d}j\right)$$

$$+ (1-\beta_1)\sum_{t=1}^{k-1}\sum_{j=0}^{n-1}\beta_1^{tn+i-j}\left(\sum_{p=j}^{n-1}C_1\eta_{k-t}\sqrt{d}L_1\sqrt{D_1}\|\nabla f(\boldsymbol{w}_{k-t,p})\|\right.$$

$$+ \sum_{r=1}^{t-1}\sum_{p=0}^{n-1}C_1\eta_{k-r}\sqrt{d}L_1\sqrt{D_1}\|\nabla f(\boldsymbol{w}_{k-r,p})\| + 2\sqrt{2}(L_0+L_1\sqrt{D_0})C_1\sqrt{d}\eta_k(tn-j)\bigg)$$

$$+ \beta_1^{(k-1)n+i+1}\left(\sum_{t=1}^{k-1}\sum_{p=0}^{n-1}L_1\sqrt{D_1}\|\nabla f(\boldsymbol{w}_{k-r,p})\|C_1\eta_{k-r}\sqrt{d} + 2\sqrt{2}(L_0+L_1\sqrt{D_0})C_1\sqrt{d}\eta_k(k-1)n\right)$$

$$\overset{(\star)}{\leq} \max_{i\in[n]}|\partial_l f_i(\boldsymbol{w}_{k,0})| + \left(n+\frac{2\sqrt{2}\beta_1}{1-\beta_1}\right)\sqrt{d}C_1(L_0+L_1\sqrt{D_0})\eta_k + L_1C_1\sqrt{D_1}\eta_k\sum_{j=0}^{i-1}\|\nabla f(\boldsymbol{w}_{k,j})\|$$

$$+ L_1C_1\sqrt{D_1}\sum_{t=1}^{k-1}\eta_{k-t}\sum_{j=0}^{n-1}\beta_1^{tn+i-j}\|\nabla f(\boldsymbol{w}_{k-t,j})\|,$$

where Inequality $(\star)$ is due to a exchange in the sum order.

Following the same routine, we have

$$|m_{l,k,-1}|$$

$$\leq (1-\beta_1)\sum_{t=1}^{k-1}\sum_{j=0}^{n-1}\beta_1^{tn-1-j}|\partial_l f_{\tau_{k-t,j}}(\boldsymbol{w}_{k-t,j})| + \beta_1^{(k-1)n}|\partial_l f_{\tau_{1,0}}(\boldsymbol{w}_{1,0})|$$

$$\leq (1-\beta_1)\sum_{t=1}^{k-1}\sum_{j=0}^{n-1}\beta_1^{tn-1-j}|\partial_l f_{\tau_{k-t,j}}(\boldsymbol{w}_{k,0})| + \beta_1^{(k-1)n}|\partial_l f_{\tau_{1,0}}(\boldsymbol{w}_{k,0})|$$

$$+ (1-\beta_1)\sum_{t=1}^{k-1}\sum_{j=0}^{n-1}\beta_1^{tn-1-j}C_1\sqrt{d}\left(\sum_{p=j}^{n-1}L_1\sqrt{D_1}\|\nabla f(\boldsymbol{w}_{k-t,p})\|\eta_{k-t} + \sum_{r=1}^{t-1}\sum_{p=0}^{n-1}L_1\sqrt{D_1}\|\nabla f(\boldsymbol{w}_{k-r,p})\|\eta_{k-r}\right.$$

$$+ 2\sqrt{2}(L_0+L_1\sqrt{D_0})C_1\sqrt{d}\eta_k(tn-j)\bigg)$$

$$+ \beta_1^{(k-1)n}\left(\sum_{t=1}^{k-1}\sum_{p=0}^{n-1}L_1\sqrt{D_1}\|\nabla f(\boldsymbol{w}_{k-r,p})\|C_1\eta_{k-r}\sqrt{d} + 2\sqrt{2}(L_0+L_1\sqrt{D_0})C_1\sqrt{d}\eta_k(k-1)n\right)$$

$$\leq \max_{i\in[n]}|\partial_l f_i(\boldsymbol{w}_{k,0})| + \sum_{t=1}^{k-1}\sum_{j=0}^{n-1}\beta_1^{tn-1-j}C_1\eta_{k-t}\sqrt{d}L_1\sqrt{D_1}\|\nabla f(\boldsymbol{w}_{k-t,j})\|$$

$$+ \frac{2\sqrt{2}(L_0+L_1\sqrt{D_0})C_1\sqrt{d}\eta_k}{1-\beta_1}.$$

The proof is completed. □

**Lemma 5** (Estimation of the norm of the adaptor). *We have for all $l \in [d], k \in \mathbb{Z}^+$,*

$$
|\boldsymbol{\nu}_{l,k,0}| \geq \beta_2^n \frac{1-\beta_2}{1-\beta_2^n} \sum_{i\in[n]} \partial_l f_i(\boldsymbol{w}_{k,0})^2 - \sqrt{\sum_{i\in[n]} |\partial_l f_i(\boldsymbol{w}_{k,0})^2|} \left( 8\sqrt{2n}\eta_k C_1 L_0 \frac{1-\beta_2}{(1-\beta_2^n)^2}\beta_2^n \right.
$$
$$
\left. + 4L_1 C_1 \frac{1-\beta_2}{1-\beta_2^n} \frac{\sqrt{1-\beta_2}}{1-\sqrt{\beta_2}} \left( \sum_{t=1}^{k-1} \beta_2^n \sqrt{\beta_2}^{-(r-1)n} \eta_{k-t} \sum_{j=0}^{n-1} (\sqrt{D_1}\|\nabla f(\boldsymbol{w}_{k-t,j})\| + \sqrt{D_0}) \right) \right),
$$

*and*

$$
|\boldsymbol{\nu}_{l,k,0}| \leq 2\max_{i\in[n]} \partial_l f_i(\boldsymbol{w}_{k,0})^2 + 2\left( 2\sqrt{2}\eta_k C_1(L_0 + L_1\sqrt{D_0}) \frac{\sqrt{1-\beta_2}}{1-\sqrt{\beta_2}} \frac{\sqrt{\beta_2}}{1-\sqrt{\beta_2}} \right.
$$
$$
\left. + L_1 C_1\sqrt{D_1} \sum_{t=1}^{k-1} \eta_{k-t} \frac{\sqrt{1-\beta_2}}{1-\sqrt{\beta_2}} \sum_{j=0}^{n-1} \sqrt{\beta_2}^{-(t-1)n} \|\nabla f(\boldsymbol{w}_{k-t,j})\| \right)^2.
$$

*Proof.* By the definition of $\boldsymbol{\nu}_{l,k,0}$, we have

$$\boldsymbol{\nu}_{l,k,0}$$

$$
=(1-\beta_2)\partial_l f_{\tau_{k,0}}(\boldsymbol{w}_{k,0})^2 + \sum_{t=1}^{k-1}\sum_{j=0}^{n-1}(1-\beta_2)\beta_2^{tn-j}\partial_l f_{\tau_{k-t,j}}(\boldsymbol{w}_{k-t,j})^2
$$
$$
+ \beta_2^{(k-1)n+1} \max_{i\in[n]} \partial_l f_i(\boldsymbol{w}_{1,0})^2
$$
$$
\geq (1-\beta_2)\partial_l f_{\tau_{k,0}}(\boldsymbol{w}_{k,0})^2 + \sum_{t=1}^{k-1}\sum_{j=0}^{n-1}(1-\beta_2)\beta_2^{tn}\partial_l f_{\tau_{k-t,j}}(\boldsymbol{w}_{k-t,j})^2
$$
$$
+ \beta_2^{(k-1)n+1}\frac{1}{n}\sum_{i=1}^{n} \partial_l f_i(\boldsymbol{w}_{1,0})^2
$$
$$
= (1-\beta_2)\partial_l f_{\tau_{k,0}}(\boldsymbol{w}_{k,0})^2 + \sum_{t=1}^{k-1}\sum_{j=0}^{n-1}(1-\beta_2)\beta_2^{tn}(\partial_l f_{\tau_{k-t,j}}(\boldsymbol{w}_{k,0}) + \partial_l f_{\tau_{k-t,j}}(\boldsymbol{w}_{k-t,j}) - \partial_l f_{\tau_{k-t,j}}(\boldsymbol{w}_{k,0}))^2
$$
$$
+ \beta_2^{(k-1)n+1}\frac{1}{n}\sum_{i=1}^{n}(\partial_l f_i(\boldsymbol{w}_{k,0}) + \partial_l f_i(\boldsymbol{w}_{1,0}) - \partial_l f_i(\boldsymbol{w}_{k,0}))^2
$$
$$
\geq (1-\beta_2)\partial_l f_{\tau_{k,0}}(\boldsymbol{w}_{k,0})^2 + \sum_{t=1}^{k-1}\sum_{j=0}^{n-1}(1-\beta_2)\beta_2^{tn}\partial_l f_{\tau_{k-t,j}}(\boldsymbol{w}_{k,0})^2
$$
$$
+ \beta_2^{(k-1)n+1}\frac{1}{n}\sum_{i=1}^{n} \partial_l f_i(\boldsymbol{w}_{k,0})^2
$$
$$
- \sum_{t=1}^{k-1}\sum_{j=0}^{n-1}(1-\beta_2)\beta_2^{tn}|\partial_l f_{\tau_{k-t,j}}(\boldsymbol{w}_{k,0})||\partial_l f_{\tau_{k-t,j}}(\boldsymbol{w}_{k,0}) - \partial_l f_{\tau_{k-t,j}}(\boldsymbol{w}_{k-t,j})|
$$
$$
- \beta_2^{(k-1)n+1}\frac{1}{n}\sum_{i=1}^{n}|\partial_l f_i(\boldsymbol{w}_{k,0})||\partial_l f_i(\boldsymbol{w}_{k,0}) - \partial_l f_i(\boldsymbol{w}_{1,0})|
$$
$$
\overset{(\star)}{\geq} \left( \beta_2^n \frac{1-\beta_2^{(k-1)n}}{1-\beta_2^n}(1-\beta_2) + \frac{\beta_2^{(k-1)n+1}}{n} \right) \sum_{i\in[n]} \partial_l f_i(\boldsymbol{w}_{k,0})^2
$$
$$
- \sum_{t=1}^{k-1}\sum_{j=0}^{n-1}(1-\beta_2)\beta_2^{tn}|\partial_l f_{\tau_{k-t,j}}(\boldsymbol{w}_{k,0})| \left( \sum_{r=1}^{t}\sum_{p=0}^{n-1} L_1\sqrt{D_1}\|\nabla f(\boldsymbol{w}_{k-r,p})\|C_1\eta_{k-r}\sqrt{d} + 2\sqrt{2}(L_0 + L_1\sqrt{D_0})C_1\sqrt{d}\eta_k tn \right)
$$
$$
- \beta_2^{(k-1)n+1}\frac{1}{n}\sum_{i=1}^{n}|\partial_l f_i(\boldsymbol{w}_{k,0})| \left( \sum_{r=1}^{k-1}\sum_{p=0}^{n-1} L_1\sqrt{D_1}\|\nabla f(\boldsymbol{w}_{k-r,p})\|C_1\eta_{k-r}\sqrt{d} + 2\sqrt{2}(L_0 + L_1\sqrt{D_0})C_1\sqrt{d}\eta_k(k-1)n \right)
$$

$$
\geq \beta_2^n \frac{1-\beta_2}{1-\beta_2^n} \sum_{i\in[n]} \partial_l f_i(\boldsymbol{w}_{k,0})^2
$$

$$
- \sum_{t=1}^{k-1} \sum_{j=0}^{n-1} (1-\beta_2)\beta_2^{tn} |\partial_l f_{\tau_{k-t,j}}(\boldsymbol{w}_{k,0})| \left( \sum_{r=1}^{t} \sum_{p=0}^{n-1} L_1\sqrt{D_1}\|\nabla f(\boldsymbol{w}_{k-r,p})\|C_1\eta_{k-r}\sqrt{d} + 2\sqrt{2}(L_0+L_1\sqrt{D_0})C_1\sqrt{d}\eta_k tn \right)
$$

$$
- \beta_2^{(k-1)n+1} \frac{1}{n} \sum_{i=1}^{n} |\partial_l f_i(\boldsymbol{w}_{k,0})| \left( \sum_{r=1}^{k-1} \sum_{p=0}^{n-1} L_1\sqrt{D_1}\|\nabla f(\boldsymbol{w}_{k-r,p})\|C_1\eta_{k-r}\sqrt{d} + 2\sqrt{2}(L_0+L_1\sqrt{D_0})C_1\sqrt{d}\eta_k(k-1)n \right)
$$

$$
\geq \beta_2^n \frac{1-\beta_2}{1-\beta_2^n} \sum_{i\in[n]} \partial_l f_i(\boldsymbol{w}_{k,0})^2 - 8\sqrt{2}\eta_k C_1 L_0 \frac{1-\beta_2}{(1-\beta_2^n)^2}\beta_2^n \sum_{i\in[n]} |\partial_l f_i(\boldsymbol{w}_{k,0})|
$$

$$
- 4L_1 C_1 \frac{1-\beta_2}{1-\beta_2^n} \sum_{i\in[n]} |\partial_l f_i(\boldsymbol{w}_{k,0})| \left( \sum_{r=1}^{k-1} \beta_2^{rn}\eta_{k-r} \sum_{j=0}^{n-1} \|\nabla f_i(\boldsymbol{w}_{k-r,j})\| \right)
$$

$$
\geq \beta_2^n \frac{1-\beta_2}{1-\beta_2^n} \sum_{i\in[n]} \partial_l f_i(\boldsymbol{w}_{k,0})^2 - 8\sqrt{2}\eta_k C_1 L_0 \frac{1-\beta_2}{(1-\beta_2^n)^2}\beta_2^n \sum_{i\in[n]} |\partial_l f_i(\boldsymbol{w}_{k,0})|
$$

$$
- 4L_1 C_1 \frac{1-\beta_2}{1-\beta_2^n} \|\nabla f_i(\boldsymbol{w}_{k,0})\| \left( \sum_{r=1}^{k-1} \beta_2^{rn}\eta_{k-r} \sum_{j=0}^{n-1} (\sqrt{D_1}\|\nabla f(\boldsymbol{w}_{k-r,j})\| + \sqrt{D_0}) \right)
$$

$$
\geq \beta_2^n \frac{1-\beta_2}{1-\beta_2^n} \sum_{i\in[n]} \partial_l f_i(\boldsymbol{w}_{k,0})^2 - 8\sqrt{2n}\eta_k C_1 L_0 \frac{1-\beta_2}{(1-\beta_2^n)^2}\beta_2^n \sqrt{\sum_{i\in[n]} |\partial_l f_i(\boldsymbol{w}_{k,0})^2|}
$$

$$
- 4L_1 C_1 \frac{1-\beta_2}{1-\beta_2^n} \sqrt{\sum_{i\in[n]} |\partial_l f_i(\boldsymbol{w}_{k,0})^2|} \left( \sum_{r=1}^{k-1} \beta_2^{rn}\eta_{k-r} \sum_{j=0}^{n-1} (\sqrt{D_1}\|\nabla f(\boldsymbol{w}_{k-r,j})\| + \sqrt{D_0}) \right)
$$

$$
\geq \beta_2^n \frac{1-\beta_2}{1-\beta_2^n} \sum_{i\in[n]} \partial_l f_i(\boldsymbol{w}_{k,0})^2 - 8\sqrt{2n}\eta_k C_1 L_0 \frac{1-\beta_2}{(1-\beta_2^n)^2}\beta_2^n \sqrt{\sum_{i\in[n]} |\partial_l f_i(\boldsymbol{w}_{k,0})^2|}
$$

$$
- 4L_1 C_1 \frac{1-\beta_2}{1-\beta_2^n} \frac{\sqrt{1-\beta_2}}{1-\sqrt{\beta_2}} \sqrt{\sum_{i\in[n]} |\partial_l f_i(\boldsymbol{w}_{k,0})^2|} \left( \sum_{r=1}^{k-1} \beta_2^n \sqrt{\beta_2}^{(r-1)n}\eta_{k-r} \sum_{j=0}^{n-1} (\sqrt{D_1}\|\nabla f(\boldsymbol{w}_{k-r,j})\| + \sqrt{D_0}) \right).
$$

The first claim is proved.

As for the upper bound, we have

$$\boldsymbol{\nu}_{l,k,0}$$

$$=(1-\beta_2)\partial_l f_{\tau_{k,0}}(\boldsymbol{w}_{k,0})^2 + \sum_{t=1}^{k-1}\sum_{j=0}^{n-1}(1-\beta_2)\beta_2^{tn-j}\partial_l f_{\tau_{k-t,j}}(\boldsymbol{w}_{k-t,j})^2$$

$$+ \beta_2^{(k-1)n+1}\max_{i\in[n]}\partial_l f_i(\boldsymbol{w}_{1,0})^2$$

$$\leq 2(1-\beta_2)\partial_l f_{\tau_{k,0}}(\boldsymbol{w}_{k,0})^2 + 2\sum_{t=1}^{k-1}\sum_{j=0}^{n-1}(1-\beta_2)\beta_2^{tn-j}\partial_l f_{\tau_{k-t,j}}(\boldsymbol{w}_{k,0})^2$$

$$+ 2\beta_2^{(k-1)n+1}\max_{i\in[n]}\partial_l f_i(\boldsymbol{w}_{k,0})^2$$

$$+ 2\sum_{t=1}^{k-1}\sum_{j=0}^{n-1}(1-\beta_2)\beta_2^{tn-j}\left(\sum_{p=j}^{n-1}L_1\sqrt{D_1}\|\nabla f(\boldsymbol{w}_{k-t,p})C_1\eta_{k-t}\sqrt{d}\right.$$

$$\left.+ \sum_{r=1}^{t-1}\sum_{p=0}^{n-1}L_1\sqrt{D_1}\|\nabla f(\boldsymbol{w}_{k-r,p})\|C_1\eta_{k-r}\sqrt{d} + 2\sqrt{2}(L_0+L_1\sqrt{D_0})C_1\sqrt{d}\eta_k(tn-j)\right)^2$$

$$+ 2\beta_2^{(k-1)n+1}\left(\sum_{r=1}^{k-1}\sum_{p=0}^{n-1}L_1\sqrt{D_1}\|\nabla f(\boldsymbol{w}_{k-r,p})\|C_1\eta_{k-r}\sqrt{d} + 2\sqrt{2}(L_0+L_1\sqrt{D_0})C_1\sqrt{d}\eta_k(k-1)n\right)^2$$

$$\leq 2\max_{i\in[n]}\partial_l f_i(\boldsymbol{w}_{k,0})^2 + 2\left(\sum_{t=1}^{k-1}\sum_{j=0}^{n-1}\sqrt{1-\beta_2}\sqrt{\beta_2}^{tn-j}\left(\sum_{p=j}^{n-1}L_1\sqrt{D_1}\|\nabla f(\boldsymbol{w}_{k-t,p})C_1\eta_{k-t}\sqrt{d}\right.\right.$$

$$\left.+ \sum_{r=1}^{t-1}\sum_{p=0}^{n-1}L_1\sqrt{D_1}\|\nabla f(\boldsymbol{w}_{k-r,p})\|C_1\eta_{k-r}\sqrt{d} + 2\sqrt{2}(L_0+L_1\sqrt{D_0})C_1\sqrt{d}\eta_k(tn-j)\right)$$

$$\left.+ \sqrt{\beta_2}^{(k-1)n+1}\left(\sum_{r=1}^{k-1}\sum_{p=0}^{n-1}L_1\sqrt{D_1}\|\nabla f(\boldsymbol{w}_{k-r,p})\|C_1\eta_{k-r}\sqrt{d} + 2\sqrt{2}(L_0+L_1\sqrt{D_0})C_1\sqrt{d}\eta_k(k-1)n\right)\right)^2$$

$$\leq 2\max_{i\in[n]}\partial_l f_i(\boldsymbol{w}_{k,0})^2 + 2\left(2\sqrt{2}\eta_k C_1(L_0+L_1\sqrt{D_0})\frac{\sqrt{1-\beta_2}}{1-\sqrt{\beta_2}}\frac{\sqrt{\beta_2}}{1-\sqrt{\beta_2}}\right.$$

$$\left.+ L_1 C_1\sqrt{D_1}\sum_{t=1}^{k-1}\eta_{k-t}\frac{\sqrt{1-\beta_2}}{1-\sqrt{\beta_2}}\sum_{j=0}^{n-1}\sqrt{\beta_2}^{tn-j}\|\nabla f(\boldsymbol{w}_{k-t,j})\|\right)^2$$

$$\leq 2\max_{i\in[n]}\partial_l f_i(\boldsymbol{w}_{k,0})^2 + 2\left(2\sqrt{2}\eta_k C_1(L_0+L_1\sqrt{D_0})\frac{\sqrt{1-\beta_2}}{1-\sqrt{\beta_2}}\frac{\sqrt{\beta_2}}{1-\sqrt{\beta_2}}\right.$$

$$\left.+ L_1 C_1\sqrt{D_1}\sum_{t=1}^{k-1}\eta_{k-t}\frac{\sqrt{1-\beta_2}}{1-\sqrt{\beta_2}}\sum_{j=0}^{n-1}\sqrt{\beta_2}^{(t-1)n}\|\nabla f(\boldsymbol{w}_{k-t,j})\|\right)^2.$$

The proof is completed. $\qquad\square$

We then immediately have the following corollary when $\max_{i\in[n]}|\partial_l f_i(\boldsymbol{w}_{k,0})|$ is large enough.

**Corollary 2** (Lemma 1, formal). *If*

$$\max_{i\in[n]}|\partial_l f_i(\boldsymbol{w}_{k,0})| \geq 4L_1 C_1\frac{\sqrt{1-\beta_2}}{1-\sqrt{\beta_2}}\left(\sum_{r=1}^{k-1}\sqrt{\beta_2}^{(r-1)n}\eta_{k-r}\sum_{j=0}^{n-1}(\sqrt{D_1}\|\nabla f(\boldsymbol{w}_{k-r,j})\| + \sqrt{D_0})\right)$$

$$+ 2\sqrt{2}\eta_k C_1(L_0+L_1\sqrt{D_0})\frac{\sqrt{1-\beta_2}}{1-\sqrt{\beta_2}}\frac{\sqrt{\beta_2}}{1-\sqrt{\beta_2}} + 8\sqrt{2n}\eta_k C_1 L_0\frac{1}{1-\beta_2^n}$$

$$+ \eta_k C_1\left(n(L_0+L_1\sqrt{D_0}) + L_1\sqrt{D_1}\left(\sum_{p=0}^{n-1}\|\nabla f(\boldsymbol{w}_{k,p})\|\right)\right), \tag{11}$$

*then*

$$\frac{\beta_2^n}{2}\frac{1}{n}\sum_{i\in[n]}\partial_l f_i(\boldsymbol{w}_{k,0})^2 \leq \boldsymbol{\nu}_{l,k,0} \leq 4\max_{i\in[n]}\partial_l f_i(\boldsymbol{w}_{k,0})^2.$$

*Furthermore, if Eq. (11) holds, we have* $\forall i \in \{0, \cdots, n-1\}$,

$$\beta_2^{n-1} \boldsymbol{\nu}_{l,k,0} \leq \boldsymbol{\nu}_{l,k,i} \leq \left(\beta_2^{n-1} + 8n\frac{1-\beta_2^{n-1}}{\beta_2^n}\right)\boldsymbol{\nu}_{l,k,0},$$

*and*

$$\frac{1}{\beta_2}\left(1 - (1-\beta_2)\frac{2n}{\beta_2^n}\right)\boldsymbol{\nu}_{l,k,0} \leq \boldsymbol{\nu}_{l,k,-1} \leq \frac{1}{\beta_2}\boldsymbol{\nu}_{l,k,0},$$

*Proof.* The first claim is derived by directly applying the range of $\max_{i\in[n]}|\partial_l f_i(\boldsymbol{w}_{k,0})|$ into Lemma 5.

As for the second claim, we have

$$\boldsymbol{\nu}_{l,k,i} = \beta_2^i \boldsymbol{\nu}_{l,k,0} + (1-\beta_2)(\partial_l f_{\tau_k,i}(\boldsymbol{w}_{k,i})^2 + \cdots + \beta_2^{i-1}\partial_l f_{\tau_k,i}(\boldsymbol{w}_{k,1})^2).$$

On the other hand, since $\forall j \in \{0, \cdots, n-1\}$

$$|\partial_l f_i(\boldsymbol{w}_{k,j})| \leq \max_{p\in[n]}|\partial_l f_p(\boldsymbol{w}_{k,0})| + \eta_k C_1\left(j(L_0 + L_1\sqrt{D_0}) + L_1\sqrt{D_1}\left(\sum_{p=0}^{j-1}\|\nabla f(\boldsymbol{w}_{k,p})\|\right)\right)$$

$$\leq \max_{p\in[n]}|\partial_l f_p(\boldsymbol{w}_{k,0})| + \eta_k C_1\left(n(L_0 + L_1\sqrt{D_0}) + L_1\sqrt{D_1}\left(\sum_{p=0}^{n-1}\|\nabla f(\boldsymbol{w}_{k,p})\|\right)\right),$$

we have

$$\beta_2^{n-1}\boldsymbol{\nu}_{l,k,0} \leq \boldsymbol{\nu}_{l,k,i}$$

$$\leq \beta_2^i\boldsymbol{\nu}_{l,k,0} + 2(1-\beta_2)\max_{p\in[n]}\partial_l f_p(\boldsymbol{w}_{k,0})^2(1 + \cdots + \beta_2^{i-1})$$

$$+ 2(1-\beta_2)(1 + \cdots + \beta_2^{i-1})\eta_k^2 C_1^2\left(n(L_0 + L_1\sqrt{D_0}) + L_1\sqrt{D_1}\left(\sum_{p=0}^{n-1}\|\nabla f(\boldsymbol{w}_{k,p})\|\right)\right)^2$$

$$= \beta_2^i\boldsymbol{\nu}_{l,k,0} + 2(1-\beta_2^i)\max_{p\in[n]}\partial_l f_p(\boldsymbol{w}_{k,0})^2 + 2(1-\beta_2^i)\eta_k^2 C_1^2\left(n(L_0 + L_1\sqrt{D_0}) + L_1\sqrt{D_1}\left(\sum_{p=0}^{n-1}\|\nabla f(\boldsymbol{w}_{k,p})\|\right)\right)^2.$$

Therefore, if Eq. (11) holds, we then have

$$\boldsymbol{\nu}_{l,k,i} \leq \beta_2^i\boldsymbol{\nu}_{l,k,0} + 4(1-\beta_2^i)\max_{p\in[n]}\partial_l f_p(\boldsymbol{w}_{k,0})^2$$

$$\leq \beta_2^i\boldsymbol{\nu}_{l,k,0} + 4\frac{n}{n}(1-\beta_2^i)\sum_{p\in[n]}\partial_l f_p(\boldsymbol{w}_{k,0})^2 \leq \left(\beta_2^i + 8n\frac{1-\beta_2^i}{\beta_2^n}\right)\boldsymbol{\nu}_{l,k,0}$$

$$\leq \left(\beta_2^{n-1} + 8n\frac{1-\beta_2^{n-1}}{\beta_2^n}\right)\boldsymbol{\nu}_{l,k,0}.$$

Following the same routine, we have

$$\beta_2\boldsymbol{\nu}_{l,k,-1} \leq \boldsymbol{\nu}_{l,k,0},$$

and if Eq. (11) holds,

$$\boldsymbol{\nu}_{l,k,-1} = \frac{1}{\beta_2}\left(\boldsymbol{\nu}_{l,k,0} - (1-\beta_2)\partial_l f_{\tau_k,0}(\boldsymbol{w}_{k,0})^2\right)$$

$$\geq \frac{1}{\beta_2}\left(\boldsymbol{\nu}_{l,k,0} - (1-\beta_2)\max_p \partial_l f_p(\boldsymbol{w}_{k,0})^2\right)$$

$$\geq \boldsymbol{\nu}_{l,k,0}\frac{1}{\beta_2}\left(1 - (1-\beta_2)\frac{2n}{\beta_2^n}\right).$$

The proof of the second claim is completed. $\qquad\square$

**Remark 11.** *For brevity, we denote*

$$C_3 \triangleq C_1 \left( n(L_0 + L_1\sqrt{D_0}) + 2\sqrt{2}(L_0 + L_1\sqrt{D_0})\frac{\sqrt{1-\beta_2}}{1-\sqrt{\beta_2}}\frac{\sqrt{\beta_2}}{1-\sqrt{\beta_2}} + 8\sqrt{2n}L_0\frac{1}{1-\beta_2^n} \right),$$

$$C_4 \triangleq 4L_1 C_1 \sqrt{D_1}\frac{\sqrt{1-\beta_2}}{1-\sqrt{\beta_2}}.$$

*The right-hand-size of Eq. (11) is smaller than*

$$\max_{i\in[n]} |\partial_l f_i(\boldsymbol{w}_{k,0})| \geq C_3\eta_k + C_4 \sum_{r=1}^{k-1} \sqrt{\beta_2}^{(r-1)n} \eta_{k-r} \sum_{j=0}^{n-1} \|\nabla f(\boldsymbol{w}_{k-r,j})\|$$

$$+ C_4 n \sum_{r=1}^{k-1} \sqrt{\beta_2}^{(r-1)n}\eta_{k-r} + \eta_k C_4 \left( \sum_{j=0}^{n-1} \|\nabla f(\boldsymbol{w}_{k,j})\| \right). \tag{12}$$

*Furthermore, we define $g(\beta_2)$ as*

$$g(\beta_2) \triangleq \max \left\{ \frac{1}{\sqrt{\beta_2}^{n-1}} - 1, 1 - \frac{1}{\sqrt{\beta_2^{n-1} + 8n\frac{1-\beta_2^{n-1}}{\beta_2^n}}}, 1 - \sqrt{\beta_2}, \sqrt{\frac{\beta_2}{\left(1-(1-\beta_2)\frac{2n}{\beta_2^n}\right)} - 1} \right\},$$

*and the conclusion of Corollary 2 can be translated into that if Eq. (12) holds,*

$$\left| \frac{1}{\sqrt{\boldsymbol{\nu}_{l,k,i}}} - \frac{1}{\sqrt{\boldsymbol{\nu}_{l,k,0}}} \right| \leq g(\beta_2)\frac{1}{\sqrt{\boldsymbol{\nu}_{l,k,0}}},$$

*and*

$$\left| \frac{1}{\sqrt{\boldsymbol{\nu}_{l,k,-1}}} - \frac{1}{\sqrt{\boldsymbol{\nu}_{l,k,0}}} \right| \leq g(\beta_2)\frac{1}{\sqrt{\boldsymbol{\nu}_{l,k,0}}}.$$

In the end of this section, we draw the connection of the gradients across the epoch.

**Lemma 6.** $\forall k \in \mathbb{N}^+, i \in \{0, \cdots, n-1\}$,

$$\|\nabla f(\boldsymbol{w}_{k,i})\| \leq (1 + n\sqrt{d}C_1\eta_1 L_1\sqrt{n}\sqrt{D_1})\|\nabla f(\boldsymbol{w}_{k,0})\| + \left(nL_0 + L_1\sqrt{n}\sqrt{D_0}\right) n\sqrt{d}C_1\eta_k.$$

*Proof.* By Assumption 1, we have

$$\|\nabla f(\boldsymbol{w}_{k,i})\|$$

$$\leq \|\nabla f(\boldsymbol{w}_{k,0})\| + \left( nL_0 + L_1 \sum_{i=1}^{n} \|\nabla f_i(\boldsymbol{w}_{k,0})\| \right) \|\boldsymbol{w}_{k,i} - \boldsymbol{w}_{k,0}\|$$

$$\leq \|\nabla f(\boldsymbol{w}_{k,0})\| + \left( nL_0 + L_1 \sum_{i=1}^{n} \|\nabla f_i(\boldsymbol{w}_{k,0})\| \right) i\sqrt{d}C_1\eta_k$$

$$\leq \|\nabla f(\boldsymbol{w}_{k,0})\| + \left( nL_0 + L_1\sqrt{n}\sqrt{\sum_{i=1}^{n} \|\nabla f_i(\boldsymbol{w}_{k,0})\|^2} \right) i\sqrt{d}C_1\eta_k$$

$$\leq \|\nabla f(\boldsymbol{w}_{k,0})\| + \left( nL_0 + L_1\sqrt{n}\sqrt{D_1}\|\nabla f(\boldsymbol{w}_{k,0})\| + L_1\sqrt{n}\sqrt{D_0} \right) i\sqrt{d}C_1\eta_k$$

$$\leq (1 + n\sqrt{d}C_1\eta_1 L_1\sqrt{n}\sqrt{D_1})\|\nabla f(\boldsymbol{w}_{k,0})\| + \left( nL_0 + L_1\sqrt{n}\sqrt{D_0} \right) n\sqrt{d}C_1\eta_k.$$

The proof is completed. $\square$

**Descent Lemma Under $(L_0, L_1)$-smooth condition** We need a descent lemma assuming $(L_0, L_1)$-smooth condition similar as the case assuming $L$ smoothness. Specifically, for a function $h : \mathcal{X} \to \mathbb{R}$

satisfying $L$-smooth condition and two points $\boldsymbol{w}$ and $\boldsymbol{v}$ in the domain $\mathcal{X}$, by Taylor's expansion, we have

$$h(\boldsymbol{w}) \leq h(\boldsymbol{v}) + \langle \nabla h(\boldsymbol{v}), \boldsymbol{w} - \boldsymbol{v} \rangle + \frac{L}{2} \|\boldsymbol{w} - \boldsymbol{v}\|^2.$$

This is called "Descent Lemma" by existing literature Sra (2014), as it guarantees that the loss decreases with proper parameter update. Parallel to the above inequality, we have the following Descent Lemma under $(L_0, L_1)$ smoothness.

**Lemma 7.** *Assume that function $h : \mathcal{X} \to \mathbb{R}$ satisfies $(L_0, L_1)$-smooth condition, i.e., $\forall \boldsymbol{w}, \boldsymbol{v} \in \mathcal{X}$ satisfying $\|\boldsymbol{w} - \boldsymbol{v}\| \leq \frac{1}{L_1}$,*

$$\|\nabla h(\boldsymbol{w}) - \nabla h(\boldsymbol{v})\| \leq (L_0 + L_1 \|\nabla h(\boldsymbol{v})\|) \|\boldsymbol{w} - \boldsymbol{v}\|.$$

*Then, for any three points $\boldsymbol{u}, \boldsymbol{w}, \boldsymbol{v} \in \mathcal{X}$ satisfying $\|\boldsymbol{w} - \boldsymbol{u}\| \leq \frac{1}{L_1}$ and $\|\boldsymbol{v} - \boldsymbol{u}\| \leq \frac{1}{L_1}$. Then,*

$$h(\boldsymbol{w}) \leq h(\boldsymbol{v}) + \langle \nabla h(\boldsymbol{u}), \boldsymbol{w} - \boldsymbol{v} \rangle + \frac{1}{2}(L_0 + L_1 \|\nabla h(\boldsymbol{u})\|)(\|\boldsymbol{v} - \boldsymbol{u}\| + \|\boldsymbol{w} - \boldsymbol{u}\|)\|\boldsymbol{w} - \boldsymbol{v}\|.$$

*Proof.* By the Fundamental theorem of calculus, we have

$$
\begin{aligned}
h(\boldsymbol{w}) =& h(\boldsymbol{v}) + \int_0^1 \langle \nabla h(\boldsymbol{v} + a(\boldsymbol{w} - \boldsymbol{v})), \boldsymbol{w} - \boldsymbol{v} \rangle \mathrm{d}a \\
=& h(\boldsymbol{v}) + \langle \nabla h(\boldsymbol{u}), \boldsymbol{w} - \boldsymbol{v} \rangle + \int_0^1 \langle \nabla h(\boldsymbol{v} + a(\boldsymbol{w} - \boldsymbol{v})) - \nabla h(\boldsymbol{u}), \boldsymbol{w} - \boldsymbol{v} \rangle \mathrm{d}a \\
\leq& h(\boldsymbol{v}) + \langle \nabla h(\boldsymbol{u}), \boldsymbol{w} - \boldsymbol{v} \rangle + \int_0^1 \|\nabla h(\boldsymbol{v} + a(\boldsymbol{w} - \boldsymbol{v})) - \nabla h(\boldsymbol{u})\| \|\boldsymbol{w} - \boldsymbol{v}\| \mathrm{d}a \\
\overset{(\star)}{\leq}& h(\boldsymbol{v}) + \langle \nabla h(\boldsymbol{u}), \boldsymbol{w} - \boldsymbol{v} \rangle + \int_0^1 (L_0 + L_1 \|\nabla h(\boldsymbol{u})\|) \|\boldsymbol{v} + a(\boldsymbol{w} - \boldsymbol{v}) - \boldsymbol{u}\| \|\boldsymbol{w} - \boldsymbol{v}\| \mathrm{d}a \\
\leq& h(\boldsymbol{v}) + \langle \nabla h(\boldsymbol{u}), \boldsymbol{w} - \boldsymbol{v} \rangle + \int_0^1 (L_0 + L_1 \|\nabla h(\boldsymbol{u})\|)((1-a)\|\boldsymbol{v} - \boldsymbol{u}\| + a\|\boldsymbol{w} - \boldsymbol{u}\|)\|\boldsymbol{w} - \boldsymbol{v}\| \mathrm{d}a \\
\leq& h(\boldsymbol{v}) + \langle \nabla h(\boldsymbol{u}), \boldsymbol{w} - \boldsymbol{v} \rangle + \frac{1}{2}(L_0 + L_1 \|\nabla h(\boldsymbol{u})\|)(\|\boldsymbol{v} - \boldsymbol{u}\| + \|\boldsymbol{w} - \boldsymbol{u}\|)\|\boldsymbol{w} - \boldsymbol{v}\|,
\end{aligned}
$$

where Inequality $(\star)$ is due to

$$\|\boldsymbol{v} + a(\boldsymbol{w} - \boldsymbol{v}) - \boldsymbol{u}\| = \|(1-a)(\boldsymbol{v} - \boldsymbol{u}) + a(\boldsymbol{w} - \boldsymbol{u})\| \leq (1-a)\|\boldsymbol{v} - \boldsymbol{u}\| + a\|\boldsymbol{w} - \boldsymbol{u}\| \leq \frac{1}{L_1},$$

and thus the definition of $(L_0, L_1)$-smooth condition can be applied.

The proof is completed. $\qquad\square$

### D.2 PROOF OF ADAM'S CONVERGENCE

*Proof of Theorem 1.* We define $\boldsymbol{u}_k \triangleq \frac{\boldsymbol{w}_{k,0} - \beta_1 \boldsymbol{w}_{k,-1}}{1 - \beta_1}$ (with $\boldsymbol{w}_{1,-1} \triangleq \boldsymbol{w}_{1,0} = \boldsymbol{w}_0$), and let $\boldsymbol{u}_{l,k}$ be the $i$-th component of $\boldsymbol{u}_k$, $\forall k \in \mathbb{N}^+$, $l \in [d]$. Then, by Lemma 3, we immediately have $\forall l \in [d]$, $|\boldsymbol{u}_{l,k} - \boldsymbol{w}_{l,k,0}|$ is bounded as

$$
\begin{aligned}
|\boldsymbol{u}_{l,k} - \boldsymbol{w}_{l,k,0}| =& \left| \frac{\boldsymbol{w}_{l,k,0} - \beta_1 \boldsymbol{w}_{l,k,-1}}{1 - \beta_1} - \boldsymbol{w}_{l,k,0} \right| \\
=& \frac{\beta_1}{1 - \beta_1} |\boldsymbol{w}_{l,k,0} - \boldsymbol{w}_{l,k,-1}| \leq \frac{\beta_1}{1 - \beta_1} C_1 \eta_1 \frac{1}{\sqrt{k-1}} \quad\quad (13) \\
\leq& \frac{\sqrt{2}\beta_1}{1 - \beta_1} C_1 \eta_1 \frac{1}{\sqrt{k}} \leq \frac{\sqrt{2}\beta_1}{1 - \beta_1} C_1 \eta_k \leq C_2 \eta_k, \quad\quad (14)
\end{aligned}
$$

and

$$
\begin{aligned}
&|\boldsymbol{u}_{l,k+1} - \boldsymbol{u}_{l,k}| \\
&= \left| \frac{\boldsymbol{w}_{l,k+1,0} - \beta_1 \boldsymbol{w}_{l,k+1,-1}}{1 - \beta_1} - \frac{\boldsymbol{w}_{l,k,0} - \beta_1 \boldsymbol{w}_{l,k,-1}}{1 - \beta_1} \right| \\
&= \left| (\boldsymbol{w}_{l,k+1,0} - \boldsymbol{w}_{l,k,0}) + \frac{\beta_1}{1 - \beta_1} (\boldsymbol{w}_{l,k+1,0} - \boldsymbol{w}_{l,k+1,-1}) - \frac{\beta_1}{1 - \beta_1} (\boldsymbol{w}_{l,k,0} - \boldsymbol{w}_{l,k,-1}) \right| \\
&\leq \left| (\boldsymbol{w}_{l,k+1,0} - \boldsymbol{w}_{l,k,0}) + \frac{\beta_1}{1 - \beta_1} (\boldsymbol{w}_{l,k+1,0} - \boldsymbol{w}_{l,k+1,-1}) - \frac{\beta_1}{1 - \beta_1} (\boldsymbol{w}_{l,k,0} - \boldsymbol{w}_{l,k,-1}) \right| \\
&\leq n C_1 \eta_1 \frac{1}{\sqrt{k}} + \frac{\beta_1}{1 - \beta_1} C_1 \eta_1 \left( \frac{1}{\sqrt{k}} + \frac{\sqrt{2}}{\sqrt{k}} \right) = C_2 \eta_1 \frac{1}{\sqrt{k}} = C_2 \eta_k,
\end{aligned} \tag{15}
$$

where $C_2$ is defined as $C_2 \triangleq n C_1 + \frac{\beta_1}{1 - \beta_1} C_1 \left( 1 + \sqrt{2} \right)$.

We then analyze the change of the Lyapunov function $f(\boldsymbol{u}_k)$ along the iterations. Specifically, by Lemma 7, we have

$$
\begin{aligned}
&f(\boldsymbol{u}_{k+1}) \\
&\leq f(\boldsymbol{u}_k) + \langle \nabla f(\boldsymbol{w}_{k,0}), \boldsymbol{u}_{k+1} - \boldsymbol{u}_k \rangle + \frac{n L_0 + L_1 \sum_{i \in [n]} \|\nabla f_i(\boldsymbol{w}_{k,0})\|}{2} (\|\boldsymbol{w}_{k,0} - \boldsymbol{u}_k\| + \|\boldsymbol{w}_{k,0} - \boldsymbol{u}_{k+1}\|) \|\boldsymbol{u}_{k+1} - \boldsymbol{u}_k\| \\
&\leq f(\boldsymbol{u}_k) + \langle \nabla f(\boldsymbol{w}_{k,0}), \boldsymbol{u}_{k+1} - \boldsymbol{u}_k \rangle + \frac{n L_0 + L_1 \sum_{i \in [n]} \|\nabla f_i(\boldsymbol{w}_{k,0})\|}{2} 3 C_2^2 d \eta_k^2 \\
&\leq f(\boldsymbol{u}_k) + \langle \nabla f(\boldsymbol{w}_{k,0}), \boldsymbol{u}_{k+1} - \boldsymbol{u}_k \rangle + \frac{n L_0 + L_1 \sqrt{n} \sqrt{\sum_{i \in [n]} \|\nabla f_i(\boldsymbol{w}_{k,0})\|^2}}{2} 3 C_2^2 d \eta_k^2 \\
&\leq f(\boldsymbol{u}_k) + \langle \nabla f(\boldsymbol{w}_{k,0}), \boldsymbol{u}_{k+1} - \boldsymbol{u}_k \rangle + \frac{n L_0 + L_1 \sqrt{n} \sqrt{D_0 + D_1 \|\nabla f(\boldsymbol{w}_{k,0})\|^2}}{2} 3 C_2^2 d \eta_k^2 \\
&\leq f(\boldsymbol{u}_k) + \langle \nabla f(\boldsymbol{w}_{k,0}), \boldsymbol{u}_{k+1} - \boldsymbol{u}_k \rangle + \frac{n L_0 + L_1 \sqrt{n} (\sqrt{D_0} + \sqrt{D_1} \|\nabla f(\boldsymbol{w}_{k,0})\|)}{2} 3 C_2^2 d \eta_k^2 \\
&\overset{(*)}{=} f(\boldsymbol{u}_k) + \sum_{l \in \mathbb{L}_{large}^k} \partial_l f(\boldsymbol{w}_{k,0})(\boldsymbol{u}_{l,k+1} - \boldsymbol{u}_{l,k}) + \sum_{l \in \mathbb{L}_{small}^k} \partial_l f(\boldsymbol{w}_{k,0})(\boldsymbol{u}_{l,k+1} - \boldsymbol{u}_{l,k}) \\
&\quad + \frac{n L_0 + L_1 \sqrt{n} \sqrt{D_0}}{2} 3 C_2^2 d \eta_k^2 + \frac{3 L_1 \sqrt{n} \sqrt{D_1} C_2^2 d \eta_k^2}{2} \|\nabla f(\boldsymbol{w}_{k,0})\|.
\end{aligned}
$$

Here in Eq. $(*)$, $\mathbb{L}_{large}$ and $\mathbb{L}_{small}$ are respectively defined as

$$
\begin{aligned}
\mathbb{L}_{large}^k &= \{l : l \in [d], \text{s.t. Eq. (12) holds}\}, \\
\mathbb{L}_{small}^k &= \{l : l \in [d], \text{s.t. Eq. (12) doesn't hold}\}.
\end{aligned}
$$

Apparently, $\mathbb{L}_{large} \cup \mathbb{L}_{small} = [d]$. We then tackle $\sum_{l \in \mathbb{L}_{large}^k} \partial_l f(\boldsymbol{w}_{k,0})(\boldsymbol{u}_{l,k+1} - \boldsymbol{u}_{l,k})$ and $\sum_{l \in \mathbb{L}_{small}^k} \partial_l f(\boldsymbol{w}_{k,0})(\boldsymbol{u}_{l,k+1} - \boldsymbol{u}_{l,k})$ respectively.

①**Analysis for** $\sum_{l \in \mathbb{L}_{small}^k} \partial_l f(\boldsymbol{w}_{k,0})(\boldsymbol{u}_{l,k+1} - \boldsymbol{u}_{l,k})$**:** By directly applying the range of $\max_{i \in [n]} |\partial_l f_i(\boldsymbol{w}_{k,0})|$, we have

$$
\begin{aligned}
&\frac{1}{n} \left| \sum_{l \in \mathbb{L}_{small}^k} \partial_l f(\boldsymbol{w}_{k,0})(\boldsymbol{u}_{l,k+1} - \boldsymbol{u}_{l,k}) \right| \\
&\leq d C_2 \eta_k \left( C_3 \eta_k + C_4 \sum_{r=1}^{k-1} \sqrt{\beta_2}^{(r-1)n} \eta_{k-r} \sum_{j=0}^{n-1} \|\nabla f(\boldsymbol{w}_{k-r,j})\| + C_4 n \sum_{r=1}^{k-1} \sqrt{\beta_2}^{(r-1)n} \eta_{k-r} + \eta_k C_4 \left( \sum_{p=0}^{n-1} \|\nabla f(\boldsymbol{w}_{k,p})\| \right) \right).
\end{aligned}
$$

Summing over $k$ from $1$ to $t$ then leads to

$$\frac{1}{n}\sum_{k=1}^{T}\left|\sum_{l\in\mathbb{L}_{small}^{k}}\partial_l f(\boldsymbol{w}_{k,0})(\boldsymbol{u}_{l,k+1}-\boldsymbol{u}_{l,k})\right|$$

$$\leq\sum_{k=1}^{T}dC_2C_3\eta_k^2+dC_2C_4\sum_{k=1}^{T}\eta_k\sum_{r=1}^{k-1}\sqrt{\beta_2}^{(r-1)n}\eta_{k-r}\sum_{j=0}^{n-1}\|\nabla f(\boldsymbol{w}_{k-r,j})\|+C_2C_4n\sum_{k=1}^{T}\eta_k\sum_{r=1}^{k-1}\sqrt{\beta_2}^{(r-1)n}\eta_{k-r}+C_2C_4\sum_{k=1}^{T}\eta_k^2\sum_{p=0}^{n-1}\|\nabla.$$

$$\leq\sum_{k=1}^{T}dC_2C_3\eta_k^2+\frac{dC_2C_4}{1-\sqrt{\beta_2^n}}\sum_{k=1}^{T-1}\eta_k^2\sum_{j=0}^{n-1}\|\nabla f(\boldsymbol{w}_{k,j})\|+\frac{C_2C_4n}{1-\sqrt{\beta_2^n}}\sum_{k=1}^{T-1}\eta_k^2+C_2C_4\sum_{k=1}^{T}\eta_k^2\sum_{p=0}^{n-1}\|\nabla f(\boldsymbol{w}_{k,p})\|$$

$$\leq\left(dC_2C_3+\frac{C_2C_4n}{1-\sqrt{\beta_2^n}}\right)\eta_1^2(1+\ln T)+\left(C_2C_4+\frac{dC_2C_4}{1-\sqrt{\beta_2^n}}\right)\sum_{k=1}^{T}\eta_k^2\sum_{j=0}^{n-1}\|\nabla f(\boldsymbol{w}_{k,j})\|,$$

which by Lemma 6 further leads to

$$\sum_{k=1}^{T}\left|\sum_{l\in\mathbb{L}_{small}^{k}}\partial_l f(\boldsymbol{w}_{k,0})(\boldsymbol{u}_{l,k+1}-\boldsymbol{u}_{l,k})\right|$$

$$\leq n\left(C_2C_4+\frac{dC_2C_4}{1-\sqrt{\beta_2^n}}\right)\sum_{k=1}^{T}\eta_k^2\sum_{j=0}^{n-1}\left((1+n\sqrt{d}C_1\eta_1L_1\sqrt{n})\|\nabla f(\boldsymbol{w}_{k,0})\|+\left(nL_0+L_1\sqrt{n}\sqrt{D_0}\right)n\sqrt{d}C_1\eta_k\right)$$

$$+n\left(dC_2C_3+\frac{C_2C_4n}{1-\sqrt{\beta_2^n}}\right)\eta_1^2(1+\ln T)$$

$$\leq n^2(1+n\sqrt{d}C_1\eta_1L_1\sqrt{n})\left(C_2C_4+\frac{dC_2C_4}{1-\sqrt{\beta_2^n}}\right)\sum_{k=1}^{T}\eta_k^2\|\nabla f(\boldsymbol{w}_{k,0})\|+\left(dC_2C_3+\frac{C_2C_4n}{1-\sqrt{\beta_2^n}}\right)\eta_1^2(1+\ln T)$$

$$+n\left(C_2C_4+\frac{dC_2C_4}{1-\sqrt{\beta_2^n}}\right)\left(nL_0+L_1\sqrt{n}\sqrt{D_0}\right)n^2\sqrt{d}C_1\sum_{k=1}^{T}\eta_k^3$$

$$\leq n^2(1+n\sqrt{d}C_1\eta_1L_1\sqrt{n}\sqrt{D_1})\left(C_2C_4+\frac{dC_2C_4\sqrt{D_1}}{1-\sqrt{\beta_2^n}}\right)\sum_{k=1}^{T}\eta_k^2\|\nabla f(\boldsymbol{w}_{k,0})\|+\left(dC_2C_3+\frac{C_2C_4n\sqrt{D_1}}{1-\sqrt{\beta_2^n}}\right)\eta_1^2(1+\ln T)$$

$$+3n\left(C_2C_4+\frac{dC_2C_4}{1-\sqrt{\beta_2^n}}\right)\left(nL_0+L_1\sqrt{n}\sqrt{D_0}\right)n^2\sqrt{d}C_1\eta_1^3. \tag{16}$$

We further define

$$C_5\triangleq n^2(1+n\sqrt{d}C_1\eta_1L_1\sqrt{n}\sqrt{D_1})\left(C_4+\frac{dC_4\sqrt{D_1}}{1-\sqrt{\beta_2^n}}\right),$$

$$C_6\triangleq\left(dC_3+\frac{C_4n\sqrt{D_1}}{1-\sqrt{\beta_2^n}}\right)\eta_1^2,$$

$$C_7\triangleq 3n\left(C_4+\frac{dC_4}{1-\sqrt{\beta_2^n}}\right)\left(nL_0+L_1\sqrt{n}\sqrt{D_0}\right)n^2\sqrt{d}C_1\eta_1^3+\left(dC_3+\frac{C_2C_4n\sqrt{D_1}}{1-\sqrt{\beta_2^n}}\right)\eta_1^2,$$

and thus

$$\sum_{k=1}^{T}\left|\sum_{l\in\mathbb{L}_{small}^{k}}\partial_l f(\boldsymbol{w}_{k,0})(\boldsymbol{u}_{l,k+1}-\boldsymbol{u}_{l,k})\right|\leq C_2\left(C_5\sum_{k=1}^{T}\eta_k^2\|\nabla f(\boldsymbol{w}_{k,0})\|+C_6\ln T+C_7\right).$$

②**Analysis for** $\sum_{l\in\mathbb{L}_{large}^{k}}\partial_l f(\boldsymbol{w}_{k,0})(\boldsymbol{u}_{l,k+1}-\boldsymbol{u}_{l,k})$**:** This term requires a more sophisticated analysis. To begin with, we provide a decomposition of $\boldsymbol{u}_{k+1}-\boldsymbol{u}_k$. According to the definition of $\boldsymbol{u}_k$, we

have

$$
\begin{aligned}
& \boldsymbol{u}_{k+1} - \boldsymbol{u}_k \\
={} & \frac{(\boldsymbol{w}_{k+1,0} - \beta_1 \boldsymbol{w}_{k+1,-1}) - (\boldsymbol{w}_{k,0} - \beta_1 \boldsymbol{w}_{k,-1})}{1 - \beta_1} \\
={} & \frac{(\boldsymbol{w}_{k+1,0} - \boldsymbol{w}_{k,0}) - \beta_1 (\boldsymbol{w}_{k+1,-1} - \boldsymbol{w}_{k,-1})}{1 - \beta_1} \\
={} & \frac{\sum_{i=0}^{n-1} (\boldsymbol{w}_{k,i+1} - \boldsymbol{w}_{k,i}) - \beta_1 \sum_{i=0}^{n-1} (\boldsymbol{w}_{k,i} - \boldsymbol{w}_{k,i-1})}{1 - \beta_1} \\
={} & \frac{(\boldsymbol{w}_{k+1,0} - \boldsymbol{w}_{k+1,-1}) + (1 - \beta_1) \sum_{i=0}^{n-2} (\boldsymbol{w}_{k,i+1} - \boldsymbol{w}_{k,i}) - \beta_1 (\boldsymbol{w}_{k,0} - \boldsymbol{w}_{k,-1})}{1 - \beta_1} \\
\overset{(\star)}{=}{} & - \frac{\frac{\eta_k}{\sqrt{\boldsymbol{\nu}_{k,n-1}}} \odot \boldsymbol{m}_{k,n-1} + (1 - \beta_1) \sum_{i=0}^{n-2} \frac{\eta_k}{\sqrt{\boldsymbol{\nu}_{k,i}}} \odot \boldsymbol{m}_{k,i} - \beta_1 \frac{\eta_{k-1}}{\sqrt{\boldsymbol{\nu}_{k-1,n-1}}} \odot \boldsymbol{m}_{k-1,n-1}}{1 - \beta_1} \\
={} & - \frac{\eta_k}{\sqrt{\boldsymbol{\nu}_{k,0}}} \odot \frac{\boldsymbol{m}_{k,n-1} + (1 - \beta_1) \sum_{i=0}^{n-2} \boldsymbol{m}_{k,i} - \beta_1 \boldsymbol{m}_{k-1,n-1}}{1 - \beta_1} - \eta_k \left( \left( \frac{1}{\sqrt{\boldsymbol{\nu}_{k,n-1}}} - \frac{1}{\sqrt{\boldsymbol{\nu}_{k,0}}} \right) \odot \frac{\boldsymbol{m}_{k,n-1}}{1 - \beta_1} \right. \\
& \left. + \sum_{i=0}^{n-2} \left( \frac{1}{\sqrt{\boldsymbol{\nu}_{k,i}}} - \frac{1}{\sqrt{\boldsymbol{\nu}_{k,0}}} \right) \odot \boldsymbol{m}_{k,i} - \frac{\beta_1}{1 - \beta_1} \left( \frac{1}{\sqrt{\boldsymbol{\nu}_{k-1,n-1}}} - \frac{1}{\sqrt{\boldsymbol{\nu}_{k,0}}} \right) \odot \boldsymbol{m}_{k-1,n-1} \right) - \frac{\beta_1}{1 - \beta_1} (\eta_{k-1} - \eta_k) \frac{1}{\sqrt{\boldsymbol{\nu}_{k-1,n-1}}} \odot \boldsymbol{m}_{k-}
\end{aligned}
$$

$$(17)$$

Here equation $(\star)$ is due to a direct application of the update rule of $\boldsymbol{w}_{k,i}$.

We then analyze the above three terms respectively, namely, we define

$$
a_l^1 \triangleq - \frac{\eta_k}{\sqrt{\boldsymbol{\nu}_{l,k,0}}} \frac{\boldsymbol{m}_{l,k,n-1} + (1 - \beta_1) \sum_{i=0}^{n-2} \boldsymbol{m}_{l,k,i} - \beta_1 \boldsymbol{m}_{l,k-1,n-1}}{1 - \beta_1} = - \frac{\eta_k}{\sqrt{\boldsymbol{\nu}_{l,k,0}}} \sum_{i=0}^{n-1} \partial_l f_{\tau_{k,i}}(\boldsymbol{w}_{k,i}),
$$

$$
\begin{aligned}
a_l^2 \triangleq - \eta_k \left( \left( \frac{1}{\sqrt{\boldsymbol{\nu}_{l,k,n-1}}} - \frac{1}{\sqrt{\boldsymbol{\nu}_{l,k,0}}} \right) \frac{\boldsymbol{m}_{l,k,n-1}}{1 - \beta_1} + \sum_{i=0}^{n-2} \left( \frac{1}{\sqrt{\boldsymbol{\nu}_{l,k,i}}} - \frac{1}{\sqrt{\boldsymbol{\nu}_{l,k,0}}} \right) \boldsymbol{m}_{l,k,i} \right. \\
\left. - \frac{\beta_1}{1 - \beta_1} \left( \frac{1}{\sqrt{\boldsymbol{\nu}_{l,k-1,n-1}}} - \frac{1}{\sqrt{\boldsymbol{\nu}_{l,k,0}}} \right) \boldsymbol{m}_{l,k-1,n-1} \right),
\end{aligned}
$$

$$
a_l^3 \triangleq - \frac{\beta_1}{1 - \beta_1} (\eta_{k-1} - \eta_k) \frac{1}{\sqrt{\boldsymbol{\nu}_{l,k-1,n-1}}} \boldsymbol{m}_{l,k-1,n-1}.
$$

One can then easily observe that by Eq. (17),

$$
\sum_{l \in \mathbb{L}_{large}^k} \partial_l f(\boldsymbol{w}_{k,0}) (\boldsymbol{u}_{l,k+1} - \boldsymbol{u}_{l,k}) = \sum_{l \in \mathbb{L}_{large}^k} \partial_l f(\boldsymbol{w}_{k,0}) a_l^1 + \sum_{l \in \mathbb{L}_{large}^k} \partial_l f(\boldsymbol{w}_{k,0}) a_l^2 + \sum_{l \in \mathbb{L}_{large}^k} \partial_l f(\boldsymbol{w}_{k,0}) a_l^3.
$$

**②. (A) Tackling Term $\sum_{l \in \mathbb{L}_{large}^k} \partial_l f(\boldsymbol{w}_{k,0}) a_l^1$:**

We have

$$
\begin{aligned}
& \sum_{l \in \mathbb{L}_{large}^k} \partial_l f(\boldsymbol{w}_{k,0}) a_l^1 \\
={} & - \sum_{l \in \mathbb{L}_{large}^k} \partial_l \frac{\eta_k}{\sqrt{\boldsymbol{\nu}_{l,k,0}}} \partial_l f(\boldsymbol{w}_{k,0}) \left( \sum_{i=0}^{n-1} \partial_l f_{\tau_{k,i}}(\boldsymbol{w}_{k,0}) \right) - \sum_{l \in \mathbb{L}_{large}^k} \frac{\eta_k}{\sqrt{\boldsymbol{\nu}_{l,k,0}}} \partial_l f(\boldsymbol{w}_{k,0}) \left( \sum_{i=0}^{n-1} (\partial_l f_{\tau_{k,i}}(\boldsymbol{w}_{k,i}) - \partial_l f_{\tau_{k,i}}(\boldsymbol{w}_{k,0})) \right) \\
={} & - \sum_{l \in \mathbb{L}_{large}^k} \frac{\eta_k}{\sqrt{\boldsymbol{\nu}_{l,k,0}}} \partial_l f(\boldsymbol{w}_{k,0})^2 - \sum_{l \in \mathbb{L}_{large}^k} \frac{\eta_k}{\sqrt{\boldsymbol{\nu}_{l,k,0}}} \partial_l f(\boldsymbol{w}_{k,0}) \left( \sum_{i=0}^{n-1} (\partial_l f_{\tau_{k,i}}(\boldsymbol{w}_{k,i}) - \partial_l f_{\tau_{k,i}}(\boldsymbol{w}_{k,0})) \right) \\
\overset{(\star)}{=}{} & - \sum_{l \in \mathbb{L}_{large}^k} \frac{\eta_k}{\sqrt{\boldsymbol{\nu}_{l,k,0}}} \partial_l f(\boldsymbol{w}_{k,0})^2 + \mathcal{O}\left( \eta_k^2 \right) + \mathcal{O}\left( \eta_k^2 \|\nabla f(\boldsymbol{w}_{k,0})\| \right),
\end{aligned}
$$

where Eq. $(\star)$ is due to

$$
\left| \sum_{l \in \mathbb{L}^k_{large}} \frac{\eta_k}{\sqrt{\boldsymbol{\nu}_{l,k,0}}} \partial_l f(\boldsymbol{w}_{k,0}) \left( \sum_{i=0}^{n-1} (\partial_l f_{\tau_{k,i}}(\boldsymbol{w}_{k,i}) - \partial_l f_{\tau_{k,i}}(\boldsymbol{w}_{k,0})) \right) \right|
$$

$$
\overset{(*)}{\leq} \eta_k \sqrt{\frac{2n^2}{\beta_2^n}} \left( \sum_{l \in \mathbb{L}^k_{large}} \sum_{i=0}^{n-1} |\partial_l f_{\tau_{k,i}}(\boldsymbol{w}_{k,i}) - \partial_l f_{\tau_{k,i}}(\boldsymbol{w}_{k,0})| \right)
$$

$$
\leq \eta_k \sqrt{\frac{2n^2}{\beta_2^n}} \left( \sqrt{d} \sum_{i=0}^{n-1} \|\nabla f_{\tau_{k,i}}(\boldsymbol{w}_{k,i}) - \nabla f_{\tau_{k,i}}(\boldsymbol{w}_{k,0})\| \right)
$$

$$
\overset{(\circ)}{\leq} \eta_k \sqrt{\frac{2n^2}{\beta_2^n}} \sqrt{d} \sum_{i=0}^{n-1} (L_0 + L_1 \|\nabla f_{\tau_{k,i}}(\boldsymbol{w}_{k,0})\|) \|\boldsymbol{w}_{k,i} - \boldsymbol{w}_{k,0}\|
$$

$$
\leq \eta_k \sqrt{\frac{2n^2}{\beta_2^n}} \sqrt{d} (nL_0 + L_1\sqrt{D_1}\sqrt{n}\|\nabla f(\boldsymbol{w}_{k,0})\| + \sqrt{n}L_1\sqrt{D_0})n\sqrt{d}C_1\eta_k
$$

$$
\overset{(\bullet)}{\leq} \sqrt{\frac{2n^2}{\beta_2^n}} d(n^2 L_0 + n\sqrt{n}L_1\sqrt{D_0})C_1\eta_k^2 + \eta_k^2 d\sqrt{\frac{2n^2}{\beta_2^n}} L_1\sqrt{D_1}n\sqrt{n}\|\nabla f(\boldsymbol{w}_{k,0})\|.
$$

Here Eq. $(*)$ is due to Corollary 2, Eq. $(\circ)$ is due to $f_i$ is $(L_0, L_1)$-smooth, $\forall i$, and Eq. $(\bullet)$ is due to Lemma 3.

**②. (B) Tackling Term $\sum_{l \in \mathbb{L}^k_{large}} \partial_l f(\boldsymbol{w}_{k,0}) a_l^2$:**

We have for any $l \in \mathbb{L}_{max}$,

$$
|\partial_l f(\boldsymbol{w}_{k,0}) a_l^2|
$$

$$
\leq \eta_k |\partial_l f(\boldsymbol{w}_{k,0})| \left( \left| \frac{1}{\sqrt{\boldsymbol{\nu}_{l,k,n-1}}} - \frac{1}{\sqrt{\boldsymbol{\nu}_{l,k,0}}} \right| \frac{|\boldsymbol{m}_{l,k,n-1}|}{1 - \beta_1} + \sum_{i=0}^{n-2} \left| \frac{1}{\sqrt{\boldsymbol{\nu}_{l,k,i}}} - \frac{1}{\sqrt{\boldsymbol{\nu}_{l,k,0}}} \right| |\boldsymbol{m}_{l,k,i}| \right.
$$

$$
\left. - \frac{\beta_1}{1 - \beta_1} \left| \frac{1}{\sqrt{\boldsymbol{\nu}_{l,k-1,n-1}}} + \frac{1}{\sqrt{\boldsymbol{\nu}_{l,k,0}}} \right| |\boldsymbol{m}_{l,k-1,n-1}| \right)
$$

$$
\overset{(\star)}{\leq} \eta_k g(\beta_2) \frac{|\partial_l f(\boldsymbol{w}_{k,0})|}{\sqrt{\boldsymbol{\nu}_{l,k,0}}} \left( \frac{|\boldsymbol{m}_{l,k,n-1}|}{1 - \beta_1} + \sum_{i=0}^{n-2} |\boldsymbol{m}_{l,k,i}| + \frac{\beta_1}{1 - \beta_1} |\boldsymbol{m}_{l,k-1,n-1}| \right)
$$

$$
\overset{(*)}{\leq} \eta_k g(\beta_2) \left( n - 1 + \frac{1 + \beta_1}{1 - \beta_1} \right) \frac{|\partial_l f(\boldsymbol{w}_{k,0})|}{\sqrt{\boldsymbol{\nu}_{l,k,0}}} \left( \max_{i \in [n]} |\partial_l f_i(\boldsymbol{w}_{k,0})| \right)
$$

$$
+ \eta_k^2 g(\beta_2) \left( n - 1 + \frac{1 + \beta_1}{1 - \beta_1} \right) \frac{\sqrt{2}n}{\beta_2^{\frac{n}{2}}} \left( n + \frac{2\sqrt{2}\beta_1}{1 - \beta_1} \right) C_1 (L_0 + L_1\sqrt{D_0})\sqrt{d}
$$

$$
+ \eta_k^2 g(\beta_2) \left( n - 1 + \frac{1 + \beta_1}{1 - \beta_1} \right) \frac{\sqrt{2}n}{\beta_2^{\frac{n}{2}}} L_1 C_1 \sqrt{D_1} \sum_{j=0}^{n-1} \|\nabla f(\boldsymbol{w}_{k,j})\|
$$

$$
+ \eta_k g(\beta_2) \left( n - 1 + \frac{1 + \beta_1}{1 - \beta_1} \right) \frac{\sqrt{2}n}{\beta_2^{\frac{n}{2}}} L_1 C_1 \sqrt{D_1} \sum_{t=1}^{k-1} \eta_{k-t} \sum_{j=0}^{n-1} \beta_1^{tn-1-j} \|\nabla f(\boldsymbol{w}_{k-t,j})\|,
$$

where Inequality $(\star)$ is due to Corollary 2, and $g(\beta_2)$ is defined in Lemma 11, and Inequality $(*)$ is due to Lemma 4, by which we have $\forall i \in \{-1, \cdots, n-1\}$

$$
|\boldsymbol{m}_{l,k,i}| \leq \max_{i' \in [n]} |\partial_l f_{i'}(\boldsymbol{w}_{k,0})| + \left( n + \frac{2\sqrt{2}\beta_1}{1 - \beta_1} \right) C_1 (L_0 + L_1\sqrt{D_0})\sqrt{d}\eta_k + L_1 C_1 \sqrt{D_1} \eta_k \sum_{j=0}^{n-1} \|\nabla f(\boldsymbol{w}_{k,j})\|
$$

$$
+ L_1 C_1 \sqrt{D_1} \sum_{t=1}^{k-1} \eta_{k-t} \sum_{j=0}^{n-1} \beta_1^{tn-1-j} \|\nabla f(\boldsymbol{w}_{k-t,j})\|.
$$

Therefore, summing over $\mathbb{L}_{large}^k$ and $k$ leads to

$$\sum_{k=1}^{T}\left|\sum_{l\in\mathbb{L}_{large}^k}\partial_l f(\boldsymbol{w}_{k,0})a_l^2\right|$$

$$\leq \sum_{k=1}^{T}\sum_{l\in\mathbb{L}_{large}^k}\eta_k g(\beta_2)\left(n-1+\frac{1+\beta_1}{1-\beta_1}\right)\frac{|\partial_l f(\boldsymbol{w}_{k,0})|}{\sqrt{\boldsymbol{\nu}_{l,k,0}}}\left(\max_{i\in[n]}|\partial_l f_i(\boldsymbol{w}_{k,0})|\right)$$

$$+\sum_{k=1}^{T}\eta_k^2 g(\beta_2)\left(n-1+\frac{1+\beta_1}{1-\beta_1}\right)\frac{\sqrt{2}n}{\beta_2^{\frac{n}{2}}}\left(n+\frac{2\sqrt{2}\beta_1}{1-\beta_1}\right)C_1(L_0+L_1\sqrt{D_0})d\sqrt{d}$$

$$+dg(\beta_2)\left(n-1+\frac{1+\beta_1}{1-\beta_1}\right)\frac{\sqrt{2}n}{\beta_2^{\frac{n}{2}}}L_1 C_1\sqrt{D_1}\sum_{k=1}^{T}\eta_k^2\sum_{j=0}^{n-1}\|\nabla f(\boldsymbol{w}_{k,j})\|$$

$$+dg(\beta_2)\left(n-1+\frac{1+\beta_1}{1-\beta_1}\right)\frac{\sqrt{2}n}{\beta_2^{\frac{n}{2}}}L_1 C_1\sqrt{D_1}\sum_{k=1}^{T}\eta_k\sum_{t=1}^{k-1}\eta_{k-t}\sum_{j=0}^{n-1}\beta_1^{(t-1)n}\|\nabla f(\boldsymbol{w}_{k-t,j})\|$$

$$\leq \sum_{k=1}^{T}\sum_{l\in\mathbb{L}_{large}^k}\eta_k g(\beta_2)\left(n-1+\frac{1+\beta_1}{1-\beta_1}\right)\frac{|\partial_l f(\boldsymbol{w}_{k,0})|}{\sqrt{\boldsymbol{\nu}_{l,k,0}}}\left(\max_{i\in[n]}|\partial_l f_i(\boldsymbol{w}_{k,0})|\right)$$

$$+g(\beta_2)\left(n-1+\frac{1+\beta_1}{1-\beta_1}\right)\frac{\sqrt{2}n}{\beta_2^{\frac{n}{2}}}\left(n+\frac{2\sqrt{2}\beta_1}{1-\beta_1}\right)C_1(L_0+L_1\sqrt{D_0})d\sqrt{d}\eta_1(1+\ln T)$$

$$+dg(\beta_2)\left(n-1+\frac{1+\beta_1}{1-\beta_1}\right)\frac{\sqrt{2}n}{\beta_2^{\frac{n}{2}}}L_1 C_1\sqrt{D_1}\left(1+\frac{1}{1-\beta_2^n}\right)\sum_{k=1}^{T}\eta_k^2\sum_{j=0}^{n-1}\|\nabla f(\boldsymbol{w}_{k,j})\|$$

$$\overset{(\star)}{\leq}\sum_{k=1}^{T}\sum_{l\in\mathbb{L}_{large}^k}\eta_k g(\beta_2)\left(n-1+\frac{1+\beta_1}{1-\beta_1}\right)\frac{|\partial_l f(\boldsymbol{w}_{k,0})|}{\sqrt{\boldsymbol{\nu}_{l,k,0}}}\left(\max_{i\in[n]}|\partial_l f_i(\boldsymbol{w}_{k,0})|\right)$$

$$+g(\beta_2)\left(n-1+\frac{1+\beta_1}{1-\beta_1}\right)\frac{\sqrt{2}n}{\beta_2^{\frac{n}{2}}}\left(n+\frac{2\sqrt{2}\beta_1}{1-\beta_1}\right)C_1(L_0+L_1\sqrt{D_0})d\sqrt{d}\eta_1(1+\ln T)$$

$$+dg(\beta_2)\left(n-1+\frac{1+\beta_1}{1-\beta_1}\right)\frac{\sqrt{2}n}{\beta_2^{\frac{n}{2}}}L_1 C_1\sqrt{D_1}\left(1+\frac{1}{1-\beta_2^n}\right)$$

$$\cdot\sum_{k=1}^{T}\eta_k^2\sum_{j=0}^{n-1}\left((1+n\sqrt{d}C_1\eta_1 L_1\sqrt{n}\sqrt{D_1})\|\nabla f(\boldsymbol{w}_{k,0})\|+\left(nL_0+L_1\sqrt{n}\sqrt{D_0}\right)n\sqrt{d}C_1\eta_k\right)$$

$$\leq \sum_{k=1}^{T}\sum_{l\in\mathbb{L}_{large}^k}\eta_k g(\beta_2)\left(n-1+\frac{1+\beta_1}{1-\beta_1}\right)\frac{|\partial_l f(\boldsymbol{w}_{k,0})|}{\sqrt{\boldsymbol{\nu}_{l,k,0}}}\left(\max_{i\in[n]}|\partial_l f_i(\boldsymbol{w}_{k,0})|\right)$$

$$+g(\beta_2)\left(n-1+\frac{1+\beta_1}{1-\beta_1}\right)\frac{\sqrt{2}n}{\beta_2^{\frac{n}{2}}}\left(n+\frac{2\sqrt{2}\beta_1}{1-\beta_1}\right)C_1(L_0+L_1\sqrt{D_0})d\sqrt{d}\eta_1^2(1+\ln T)$$

$$+dg(\beta_2)\left(n-1+\frac{1+\beta_1}{1-\beta_1}\right)\frac{\sqrt{2}n}{\beta_2^{\frac{n}{2}}}L_1 C_1\sqrt{D_1}\left(1+\frac{1}{1-\beta_2^n}\right)(n+n^{\frac{5}{2}}\sqrt{d}C_1\eta_1 L_1\sqrt{D_1})\sum_{k=1}^{T}\eta_k^2\|\nabla f(\boldsymbol{w}_{k,0})\|$$

$$+3dg(\beta_2)\left(n-1+\frac{1+\beta_1}{1-\beta_1}\right)\frac{\sqrt{2}n}{\beta_2^{\frac{n}{2}}}L_1 C_1\sqrt{D_1}\left(1+\frac{1}{1-\beta_2^n}\right)n\left(nL_0+L_1\sqrt{n}\sqrt{D_0}\right)n\sqrt{d}C_1\eta_1^3.$$

where Inequality $(\star)$ is due to Lemma 6.

②. **(C) Tackling Term** $\sum_{l\in\mathbb{L}_{large}^k}\partial_l f(\boldsymbol{w}_{k,0})a_l^3$**:**

For any $l \in \mathbb{L}_{large}^k$,

$$|\partial_l f(\boldsymbol{w}_{k,0}) a_l^3|$$
$$\leq \frac{\beta_1}{1-\beta_1} |\eta_{k-1} - \eta_k| \frac{1}{\sqrt{\boldsymbol{\nu}_{l,k-1,n-1}}} |\boldsymbol{m}_{l,k-1,n-1}| |\partial_l f(\boldsymbol{w}_{k,0})|$$
$$\leq \frac{\beta_1 \eta_1}{(1-\beta_1)} \frac{1}{\sqrt{k}\sqrt{k-1}(\sqrt{k}+\sqrt{k-1})} C_1 |\partial_l f(\boldsymbol{w}_{k,0})|$$
$$= \frac{\beta_1 \eta_k}{(1-\beta_1)} \frac{1}{\sqrt{k-1}(\sqrt{k}+\sqrt{k-1})} C_1 |\partial_l f(\boldsymbol{w}_{k,0})|.$$

Summing over $k$ and $\mathbb{L}_{large}^k$ then leads to

$$\sum_{k=1}^T \sum_{l \in \mathbb{L}_{large}^k} |\partial_l f(\boldsymbol{w}_{k,0}) a_l^3|$$
$$\leq \frac{\beta_1}{(1-\beta_1)} \sum_{k=1}^T \sum_{l \in \mathbb{L}_{large}^k} \frac{\eta_k}{\sqrt{k-1}(\sqrt{k}+\sqrt{k-1})} C_1 |\partial_l f(\boldsymbol{w}_{k,0})|$$
$$\leq 2 \frac{\beta_1}{(1-\beta_1)\eta_1} \sqrt{d} C_1 \sum_{k=1}^T \eta_k^2 \|\nabla f(\boldsymbol{w}_{k,0})\|.$$

**Put ②.(A), ②.(B), and ②.(C) together.** We have

$$\sum_{k=1}^T \sum_{l \in \mathbb{L}_{large}^k} \partial_l f(\boldsymbol{w}_{k,0})(\boldsymbol{u}_{l,k+1} - \boldsymbol{u}_{l,k})$$
$$\leq - \sum_{k=1}^T \sum_{l \in \mathbb{L}_{large}^k} \frac{\eta_k}{\sqrt{\boldsymbol{\nu}_{l,k,0}}} \partial_l f(\boldsymbol{w}_{k,0})^2 + \sum_{k=1}^T \sum_{l \in \mathbb{L}_{large}^k} \eta_k g(\beta_2) \left(n-1+\frac{1+\beta_1}{1-\beta_1}\right) \frac{|\partial_l f(\boldsymbol{w}_{k,0})|}{\sqrt{\boldsymbol{\nu}_{l,k,0}}} \left(\max_{i \in [n]} |\partial_l f_i(\boldsymbol{w}_{k,0})|\right)$$
$$+ C_8 \sum_{k=1}^T \eta_k^2 \|\nabla f(\boldsymbol{w}_{k,0})\| + C_9 \ln T + C_{10}, \tag{18}$$

where $C_5$, $C_6$, and $C_7$ are constants defined as

$$C_8 \triangleq \sqrt{\frac{2n^2}{\beta_2^n}} L_1 \sqrt{D_1} n \sqrt{n} + dg(\beta_2) \left(n-1+\frac{1+\beta_1}{1-\beta_1}\right) \frac{\sqrt{2}n}{\beta_2^{\frac{n}{2}}} L_1 C_1 \sqrt{D_1} \left(1 + \frac{1}{1-\beta_2^n}\right)(n + n^{\frac{5}{2}}\sqrt{d}C_1\eta_1 L_1 \sqrt{D_1}) + 2\frac{\beta_1}{(1-\beta_1)\eta_1}\sqrt{d}$$

$$C_9 \triangleq \sqrt{\frac{2n^2}{\beta_2^n}} d(n^2 L_0 + n\sqrt{n}L_1\sqrt{D_0}) C_1 \eta_1^2 + g(\beta_2) \left(n-1+\frac{1+\beta_1}{1-\beta_1}\right) \frac{\sqrt{2}n}{\beta_2^{\frac{n}{2}}} \left(n + \frac{2\sqrt{2}\beta_1}{1-\beta_1}\right) C_1 (L_0 + L_1\sqrt{D_0}) d\sqrt{d}\eta_1^2,$$

$$C_{10} \triangleq 3dg(\beta_2) \left(n-1+\frac{1+\beta_1}{1-\beta_1}\right) \frac{\sqrt{2}n}{\beta_2^{\frac{n}{2}}} L_1 C_1 \sqrt{D_1} \left(1 + \frac{1}{1-\beta_2^n}\right) n \left(nL_0 + L_1\sqrt{n}\sqrt{D_0}\right) n\sqrt{d}C_1\eta_1^3 + C_9.$$

We then analyze the first two terms in Eq. (18) here. Specifically, we have

$$\sum_{l\in\mathbb{L}_{large}^k}\frac{\eta_k\partial_l f(\boldsymbol{w}_{k,0})^2}{\sqrt{\boldsymbol{\nu}_{l,k,0}}+\varepsilon}-\sum_{l\in\mathbb{L}_{large}^k}\eta_k g(\beta_2)\left(n-1+\frac{1+\beta_1}{1-\beta_1}\right)\frac{|\partial_l f(\boldsymbol{w}_{k,0})|}{\sqrt{\boldsymbol{\nu}_{l,k,0}}+\varepsilon}\left(\max_{i\in[n]}|\partial_l f_i(\boldsymbol{w}_{k,0})|\right)$$

$$\overset{(\star)}{\geq}\sum_{l\in\mathbb{L}_{large}^k}\frac{\eta_k\partial_l f(\boldsymbol{w}_{k,0})^2}{\sqrt{\boldsymbol{\nu}_{l,k,0}}+\varepsilon}-\sum_{l\in\mathbb{L}_{large}^k}\eta_k g(\beta_2)\left(n-1+\frac{1+\beta_1}{1-\beta_1}\right)\frac{|\partial_l f(\boldsymbol{w}_{k,0})|}{\sqrt{\frac{\beta_2^n}{2n}\max_{i\in[n]}|\partial_l f_i(\boldsymbol{w}_{k,0})|}+\varepsilon}\left(\max_{i\in[n]}|\partial_l f_i(\boldsymbol{w}_{k,0})|\right)$$

$$\geq\sum_{l\in\mathbb{L}_{large}^k}\frac{\eta_k\partial_l f(\boldsymbol{w}_{k,0})^2}{2\max_{i\in[n]}|\partial_l f_i(\boldsymbol{w}_{k,0})|+\varepsilon}-\sum_{l\in\mathbb{L}_{large}^k}\eta_k g(\beta_2)\left(n-1+\frac{1+\beta_1}{1-\beta_1}\right)\frac{|\partial_l f(\boldsymbol{w}_{k,0})|}{\sqrt{\frac{\beta_2^n}{2n}\max_{i\in[n]}|\partial_l f_i(\boldsymbol{w}_{k,0})|}+\varepsilon}\left(\max_{i\in[n]}|\partial_l f_i(\boldsymbol{w}_{k,0})|\right)$$

$$\overset{(\circ)}{=}\sum_{l\in[d]}\frac{\eta_k\partial_l f(\boldsymbol{w}_{k,0})^2}{2\max_{i\in[n]}|\partial_l f_i(\boldsymbol{w}_{k,0})|+\varepsilon}-\sum_{l\in\mathbb{L}_{large}^k}\eta_k g(\beta_2)\left(n-1+\frac{1+\beta_1}{1-\beta_1}\right)\frac{|\partial_l f(\boldsymbol{w}_{k,0})|}{\sqrt{\frac{\beta_2^n}{2n}\max_{i\in[n]}|\partial_l f_i(\boldsymbol{w}_{k,0})|}+\varepsilon}\left(\max_{i\in[n]}|\partial_l f_i(\boldsymbol{w}_{k,0})|\right)$$

$$+\mathcal{O}\left(\eta_k^2+\eta_k\sum_{r=1}^{k-1}\sqrt{\beta_2}^{(r-1)n}\eta_{k-r}\sum_{j=0}^{n-1}\|\nabla f(\boldsymbol{w}_{k-r,j})\|+\eta_k\sum_{r=1}^{k-1}\sqrt{\beta_2}^{(r-1)n}\eta_{k-r}+\eta_k^2\sum_{j=0}^{n-1}\|\nabla f(\boldsymbol{w}_{k,j})\|\right)$$

$$\geq\sum_{l\in[d]}\frac{\eta_k\partial_l f(\boldsymbol{w}_{k,0})^2}{2\max_{i\in[n]}|\partial_l f_i(\boldsymbol{w}_{k,0})|+\varepsilon}-\sum_{l\in[d]}\eta_k g(\beta_2)\left(n-1+\frac{1+\beta_1}{1-\beta_1}\right)\frac{|\partial_l f(\boldsymbol{w}_{k,0})|}{\sqrt{\frac{\beta_2^n}{2n}\max_{i\in[n]}|\partial_l f_i(\boldsymbol{w}_{k,0})|}+\varepsilon}\left(\max_{i\in[n]}|\partial_l f_i(\boldsymbol{w}_{k,0})|\right)$$

$$+\mathcal{O}\left(\eta_k^2+\eta_k\sum_{r=1}^{k-1}\sqrt{\beta_2}^{(r-1)n}\eta_{k-r}\sum_{j=0}^{n-1}\|\nabla f(\boldsymbol{w}_{k-r,j})\|+\eta_k\sum_{r=1}^{k-1}\sqrt{\beta_2}^{(r-1)n}\eta_{k-r}+\eta_k^2\sum_{j=0}^{n-1}\|\nabla f(\boldsymbol{w}_{k,j})\|\right),$$

where Inequality $(\star)$ is due to Corollary 2 and Equality $(\circ)$ is due to

$$\sum_{l\in\mathbb{L}_{small}^k}\frac{\eta_k\partial_l f(\boldsymbol{w}_{k,0})^2}{2\max_{i\in[n]}|\partial_l f_i(\boldsymbol{w}_{k,0})|+\varepsilon}\leq\sum_{l\in\mathbb{L}_{small}^k}\frac{\eta_k\partial_l f(\boldsymbol{w}_{k,0})^2}{2\max_{i\in[n]}|\partial_l f_i(\boldsymbol{w}_{k,0})|+\varepsilon}\leq\frac{n}{2}\eta_k\sum_{l\in\mathbb{L}_{small}^k}\max_{i\in[n]}|\partial_l f_i(\boldsymbol{w}_{k,0})|$$

$$\leq\frac{nd\eta_k}{2}\left(C_3\eta_k+C_4\sum_{r=1}^{k-1}\sqrt{\beta_2}^{(r-1)n}\eta_{k-r}\sum_{j=0}^{n-1}\|\nabla f(\boldsymbol{w}_{k-r,j})\|+C_4 n\sum_{r=1}^{k-1}\sqrt{\beta_2}^{(r-1)n}\eta_{k-r}+\eta_k C_4\sum_{j=0}^{n-1}\|\nabla f(\boldsymbol{w}_{k,j})\|\right).$$

Parallel to Eq. (16) and summing the right-hand-side of the above inequality over $k$ from 1 to $t$, we have

$$\sum_{t=1}^{T}\frac{nd\eta_k}{2}\left(C_3\eta_k+C_4\sum_{r=1}^{k-1}\sqrt{\beta_2}^{(r-1)n}\eta_{k-r}\sum_{j=0}^{n-1}\|\nabla f(\boldsymbol{w}_{k-r,j})\|+C_4 n\sum_{r=1}^{k-1}\sqrt{\beta_2}^{(r-1)n}\eta_{k-r}+\eta_k C_4\sum_{j=0}^{n-1}\|\nabla f(\boldsymbol{w}_{k,j})\|\right)$$

$$\leq\frac{1}{2}\left(C_5\sum_{k=1}^{T}\eta_k^2\|\nabla f(\boldsymbol{w}_{k,0})\|+C_6\ln T+C_7\right).$$

Suppose now there does not exist an iteration $k\in[T]$, such that

$$\|\nabla f(\boldsymbol{w}_{k,0})\|\leq 2\sqrt{d}(2\sqrt{2}+1)\sqrt{D_0}g(\beta_2)\left(n-1+\frac{1+\beta_1}{1-\beta_1}\right)\sqrt{\frac{2n}{\beta_2^n}},$$

since otherwise, the proof has been completed. By Lemma 8, we then have

$$\sum_{l\in\mathbb{L}_{large}^k}\frac{\eta_k\partial_l f(\boldsymbol{w}_{k,0})^2}{\sqrt{\boldsymbol{\nu}_{l,k,0}}+\varepsilon}-\sum_{l\in\mathbb{L}_{large}^k}\eta_k g(\beta_2)\left(n-1+\frac{1+\beta_1}{1-\beta_1}\right)\frac{|\partial_l f(\boldsymbol{w}_{k,0})|}{\sqrt{\boldsymbol{\nu}_{l,k,0}}+\varepsilon}\left(\max_{i\in[n]}|\partial_l f_i(\boldsymbol{w}_{k,0})|\right)$$

$$\geq\eta_k\frac{1}{2(2\sqrt{2}+1)}\min\left\{\frac{\|\nabla f(\boldsymbol{w}_{k,0})\|}{\sqrt{D_1}},\frac{\|\nabla f(\boldsymbol{w}_{k,0})\|^2}{\varepsilon+\sqrt{D_0}}\right\}$$

$$+\mathcal{O}\left(\eta_k^2+\eta_k\sum_{r=1}^{k-1}\sqrt{\beta_2}^{(r-1)n}\eta_{k-r}\sum_{j=0}^{n-1}\|\nabla f(\boldsymbol{w}_{k-r,j})\|+\eta_k\sum_{r=1}^{k-1}\sqrt{\beta_2}^{(r-1)n}\eta_{k-r}+\eta_k^2\sum_{j=0}^{n-1}\|\nabla f(\boldsymbol{w}_{k,j})\|\right).$$

**Putting ① and ② together and summing over $k$,** we have

$$f(\boldsymbol{u}_{T+1})-f(\boldsymbol{u}_1)$$

$$\leq-\sum_{k=1}^{T}\eta_k\frac{1}{2(2\sqrt{2}+1)}\min\left\{\frac{\|\nabla f(\boldsymbol{w}_{k,0})\|}{\sqrt{D_1}},\frac{\|\nabla f(\boldsymbol{w}_{k,0})\|^2}{\varepsilon+\sqrt{D_0}}\right\}+\left((\frac{1}{2}+C_2)C_5+C_8\right)\sum_{k=1}^{T}\eta_k^2\|\nabla f(\boldsymbol{w}_{k,0})\|$$

$$+\left((\frac{1}{2}+C_2)C_6+C_9\right)\ln T+\left((\frac{1}{2}+C_2)C_7+C_{10}\right)+\sum_{k=1}^{T}\frac{nL_0+L_1\sqrt{n}\sqrt{D_0}}{2}3C_2^2 d\eta_k^2+\sum_{k=1}^{T}\frac{3L_1\sqrt{n}\sqrt{D_1}C_2^2 d\eta_k^2}{2}\|\nabla f(\boldsymbol{w}_{k,0})\|$$

$$\leq - \sum_{k=1}^{T} \eta_k \frac{1}{2(2\sqrt{2}+1)} \min\left\{ \frac{\|\nabla f(\boldsymbol{w}_{k,0})\|}{\sqrt{D_1}}, \frac{\|\nabla f(\boldsymbol{w}_{k,0})\|^2}{\varepsilon + \sqrt{D_0}} \right\} + \left( (\frac{1}{2}+C_2)C_5 + C_8 + \frac{3L_1\sqrt{n}\sqrt{D_1}C_2^2 d}{2} \right) \sum_{k=1}^{T} \eta_k^2 \|\nabla f(\boldsymbol{w}_{k,0})\|$$

$$+ \left( (\frac{1}{2}+C_2)C_6 + C_9 + \frac{nL_0 + L_1\sqrt{n}\sqrt{D_0}}{2} 3C_2^2 d\eta_1^2 \right) \ln T + \left( (\frac{1}{2}+C_2)C_7 + C_{10} + \frac{nL_0 + L_1\sqrt{n}\sqrt{D_0}}{2} 3C_2^2 d\eta_1^2 \right)$$

$$\leq \sum_{k=1}^{T} \eta_k \frac{1}{2(2\sqrt{2}+1)} \min\left\{ \frac{\|\nabla f(\boldsymbol{w}_{k,0})\|}{\sqrt{D_1}}, \frac{\|\nabla f(\boldsymbol{w}_{k,0})\|^2}{\varepsilon + \sqrt{D_0}} \right\} + C_{11} \sum_{k=1}^{T} \eta_k^2 \|\nabla f(\boldsymbol{w}_{k,0})\| + C_{12} \ln T + C_{13},$$

where $C_{11}$, $C_{12}$, and $C_{13}$ is defined as

$$C_{11} \triangleq (\frac{1}{2}+C_2)C_5 + C_8 + \frac{3L_1\sqrt{n}\sqrt{D_1}C_2^2 d}{2},$$

$$C_{12} \triangleq (\frac{1}{2}+C_2)C_6 + C_9 + \frac{nL_0 + L_1\sqrt{n}\sqrt{D_0}}{2} 3C_2^2 d\eta_1^2,$$

$$C_{13} \triangleq (\frac{1}{2}+C_2)C_7 + C_{10} + \frac{nL_0 + L_1\sqrt{n}\sqrt{D_0}}{2} 3C_2^2 d\eta_1^2.$$

On the other hand, as for $\forall k \in [T]$,

$$\eta_k^2 \|\nabla f(\boldsymbol{w}_{k,0})\| \leq \frac{1}{4} \frac{\sqrt{D_0}+\varepsilon}{\sqrt{D_1}} \eta_k^2 + \frac{\sqrt{D_1}}{\sqrt{D_0}+\varepsilon} \eta_k^2 \|\nabla f(\boldsymbol{w}_{k,0})\|^2,$$

we have that

$$\eta_k^2 \|\nabla f(\boldsymbol{w}_{k,0})\| \leq \frac{1}{4} \frac{\sqrt{D_0}+\varepsilon}{\sqrt{D_1}} \eta_k^2 + \eta_k^2 \min\left\{ \|\nabla f(\boldsymbol{w}_{k,0})\|, \frac{\sqrt{D_1}}{\sqrt{D_0}+\varepsilon} \|\nabla f(\boldsymbol{w}_{k,0})\|^2 \right\}$$

$$= \frac{1}{4} \frac{\sqrt{D_0}+\varepsilon}{\sqrt{D_1}} \eta_k^2 + \sqrt{D_1} \eta_k^2 \min\left\{ \frac{\|\nabla f(\boldsymbol{w}_{k,0})\|}{\sqrt{D_1}}, \frac{\|\nabla f(\boldsymbol{w}_{k,0})\|^2}{\sqrt{D_0}+\varepsilon} \right\},$$

and thus,

$$f(\boldsymbol{u}_{T+1}) - f(\boldsymbol{u}_1)$$

$$\leq - \sum_{k=1}^{T} \eta_k \frac{1}{2(2\sqrt{2}+1)} \min\left\{ \frac{\|\nabla f(\boldsymbol{w}_{k,0})\|}{\sqrt{D_1}}, \frac{\|\nabla f(\boldsymbol{w}_{k,0})\|^2}{\varepsilon + \sqrt{D_0}} \right\} + C_{11} \sum_{k=1}^{T} \eta_k^2 \|\nabla f(\boldsymbol{w}_{k,0})\| + C_{12} \ln T + C_{13}$$

$$\leq - \sum_{k=1}^{T} \eta_k \frac{1}{2(2\sqrt{2}+1)} \min\left\{ \frac{\|\nabla f(\boldsymbol{w}_{k,0})\|}{\sqrt{D_1}}, \frac{\|\nabla f(\boldsymbol{w}_{k,0})\|^2}{\varepsilon + \sqrt{D_0}} \right\} + \frac{\sqrt{D_0}+\varepsilon}{4\sqrt{D_1}} C_{11} \sum_{k=1}^{T} \eta_k^2 + C_{12} \ln T + C_{13}$$

$$+ \sqrt{D_1} C_{11} \sum_{k=1}^{T} \eta_k^2 \min\left\{ \frac{\|\nabla f(\boldsymbol{w}_{k,0})\|}{\sqrt{D_1}}, \frac{\|\nabla f(\boldsymbol{w}_{k,0})\|^2}{\varepsilon + \sqrt{D_0}} \right\}$$

$$\leq - \sum_{k=1}^{T} \eta_k \left( \frac{1}{2(2\sqrt{2}+1)} - \sqrt{D_1} C_{11} \eta_k \right) \min\left\{ \frac{\|\nabla f(\boldsymbol{w}_{k,0})\|}{\sqrt{D_1}}, \frac{\|\nabla f(\boldsymbol{w}_{k,0})\|^2}{\varepsilon + \sqrt{D_0}} \right\} + \left( C_{12} + \frac{\sqrt{D_0}+\varepsilon}{4\sqrt{D_1}} C_{11} \eta_1^2 \right) \ln T$$

$$+ \left( C_{13} + \frac{\sqrt{D_0}+\varepsilon}{4\sqrt{D_1}} C_{11} \eta_1^2 \right)$$

$$\leq - \sum_{k=1}^{T} \eta_k \frac{1}{4(2\sqrt{2}+1)} \min\left\{ \frac{\|\nabla f(\boldsymbol{w}_{k,0})\|}{\sqrt{D_1}}, \frac{\|\nabla f(\boldsymbol{w}_{k,0})\|^2}{\varepsilon + \sqrt{D_0}} \right\} + \left( C_{12} + \frac{\sqrt{D_0}+\varepsilon}{4\sqrt{D_1}} C_{11} \eta_1^2 \right) \ln T$$

$$+ \left( C_{13} + \frac{\sqrt{D_0}+\varepsilon}{4\sqrt{D_1}} C_{11} \eta_1^2 \right).$$

The proof is completed. □

**Remark 12.** *By the definitions of $C_{11}$, $C_{12}$, and $C_{13}$, one can easily observe that the hidden coefficient of $\mathcal{O}(\frac{\ln T}{\sqrt{T}})$ is in the order of $\frac{3}{2}$ for $d$ and $\frac{3}{2}$ for $n$. For completeness, we would like to emphasize that our contribution is "providing the first convergence bound of Adam without the L-smooth condition", but we agree the bound itself can be tighten. Proving tighter bound is an interesting topic and we leave it as a future work.*

**Lemma 8.** *Let Assumptions 1 and 2 hold. Let $\beta_1^2 < \beta_2$ and Eq. (4) hold. Then, either there exists a iteration $k \in [T]$, such that either*

$$\|\nabla f(\boldsymbol{w}_{k,0})\| \leq 2\sqrt{d}(2\sqrt{2}+1)\sqrt{D_0}g(\beta_2)\left(n-1+\frac{1+\beta_1}{1-\beta_1}\right)\sqrt{\frac{2n}{\beta_2^n}},$$

*or for all iteration $k \in [1, T]$, we have that*

$$\sum_{l\in[d]}\frac{\eta_k\partial_l f(\boldsymbol{w}_{k,0})^2}{2\max_{i\in[n]}|\partial_l f_i(\boldsymbol{w}_{k,0})|+\varepsilon}-\sum_{l\in[d]}\eta_k g(\beta_2)\left(n-1+\frac{1+\beta_1}{1-\beta_1}\right)\frac{|\partial_l f(\boldsymbol{w}_{k,0})|}{\sqrt{\frac{\beta_2^n}{2n}}\max_{i\in[n]}|\partial_l f_i(\boldsymbol{w}_{k,0})|+\varepsilon}\left(\max_{i\in[n]}|\partial_l f_i(\boldsymbol{w}_{k,0})|\right)$$

$$\geq \eta_k\frac{1}{2(2\sqrt{2}+1)}\min\left\{\frac{\|\nabla f(\boldsymbol{w}_{k,0})\|}{\sqrt{D_1}},\frac{\|\nabla f(\boldsymbol{w}_{k,0})\|^2}{\varepsilon+\sqrt{D_0}}\right\}.$$

*Proof.* To begin with, we have

$$\sum_{l\in[d]}\frac{\eta_k\partial_l f(\boldsymbol{w}_{k,0})^2}{2\max_{i\in[n]}|\partial_l f_i(\boldsymbol{w}_{k,0})|+\varepsilon}-\sum_{l\in[d]}\eta_k g(\beta_2)\left(n-1+\frac{1+\beta_1}{1-\beta_1}\right)\frac{|\partial_l f(\boldsymbol{w}_{k,0})|}{\sqrt{\frac{\beta_2^n}{2n}}\max_{i\in[n]}|\partial_l f_i(\boldsymbol{w}_{k,0})|+\varepsilon}\left(\max_{i\in[n]}|\partial_l f_i(\boldsymbol{w}_{k,0})|\right)$$

$$\overset{(\star)}{\geq}\sum_{l\in[d]}\frac{\eta_k\partial_l f(\boldsymbol{w}_{k,0})^2}{2\sqrt{D_1\|\nabla f(\boldsymbol{w}_{k,0})\|^2+D_0}+\varepsilon}-\sum_{l\in[d]}\eta_k g(\beta_2)\left(n-1+\frac{1+\beta_1}{1-\beta_1}\right)\frac{|\partial_l f(\boldsymbol{w}_{k,0})|}{\sqrt{\frac{\beta_2^n}{2n}}\max_{i\in[n]}|\partial_l f_i(\boldsymbol{w}_{k,0})|+\varepsilon}\left(\max_{i\in[n]}|\partial_l f_i(\boldsymbol{w}_{k,0})|\right)$$

$$=\frac{\eta_k\|\nabla f(\boldsymbol{w}_{k,0})\|^2}{2\sqrt{D_1\|\nabla f(\boldsymbol{w}_{k,0})\|^2+D_0}+\varepsilon}-\sum_{l\in[d]}\eta_k g(\beta_2)\left(n-1+\frac{1+\beta_1}{1-\beta_1}\right)\frac{|\partial_l f(\boldsymbol{w}_{k,0})|}{\sqrt{\frac{\beta_2^n}{2n}}\max_{i\in[n]}|\partial_l f_i(\boldsymbol{w}_{k,0})|+\varepsilon}\left(\max_{i\in[n]}|\partial_l f_i(\boldsymbol{w}_{k,0})|\right),$$

where Inequality $(\star)$ is due to that

$$\max_{i\in[n]}|\partial_l f_i(\boldsymbol{w}_{k,0})|=\sqrt{\max_{i\in[n]}|\partial_l f_i(\boldsymbol{w}_{k,0})|^2}$$

$$\leq\sqrt{\sum_{i\in[n]}\sum_{l'=1}^d|\partial_{l'}f_i(\boldsymbol{w}_{k,0})|^2}=\sqrt{\sum_{i\in[n]}\|\nabla f_i(\boldsymbol{w}_{k,0})\|^2}\leq\sqrt{D_1\|\nabla f(\boldsymbol{w}_{k,0})\|^2+D_0}.$$

We respectively consider the case $\varepsilon\leq\sqrt{D_0}$ and $\varepsilon>\sqrt{D_0}$.

**Case I:** $\varepsilon\leq\sqrt{D_0}$**.** In this case, we have that

$$\frac{\eta_k\|\nabla f(\boldsymbol{w}_{k,0})\|^2}{2\sqrt{D_1\|\nabla f(\boldsymbol{w}_{k,0})\|^2+D_0}+\varepsilon}-\sum_{l\in[d]}\eta_k g(\beta_2)\left(n-1+\frac{1+\beta_1}{1-\beta_1}\right)\frac{|\partial_l f(\boldsymbol{w}_{k,0})|}{\sqrt{\frac{\beta_2^n}{2n}}\max_{i\in[n]}|\partial_l f_i(\boldsymbol{w}_{k,0})|+\varepsilon}\left(\max_{i\in[n]}|\partial_l f_i(\boldsymbol{w}_{k,0})|\right)$$

$$\geq\frac{\eta_k\|\nabla f(\boldsymbol{w}_{k,0})\|^2}{2\sqrt{D_1\|\nabla f(\boldsymbol{w}_{k,0})\|^2+D_0}+\sqrt{D_0}}-\sum_{l\in[d]}\eta_k g(\beta_2)\left(n-1+\frac{1+\beta_1}{1-\beta_1}\right)\frac{|\partial_l f(\boldsymbol{w}_{k,0})|}{\sqrt{\frac{\beta_2^n}{2n}}\max_{i\in[n]}|\partial_l f_i(\boldsymbol{w}_{k,0})|}\left(\max_{i\in[n]}|\partial_l f_i(\boldsymbol{w}_{k,0})|\right)$$

$$=\frac{\eta_k\|\nabla f(\boldsymbol{w}_{k,0})\|^2}{2\sqrt{D_1\|\nabla f(\boldsymbol{w}_{k,0})\|^2+D_0}+\sqrt{D_0}}-\sum_{l\in[d]}\eta_k g(\beta_2)\left(n-1+\frac{1+\beta_1}{1-\beta_1}\right)\sqrt{\frac{2n}{\beta_2^n}}|\partial_l f(\boldsymbol{w}_{k,0})|$$

$$\geq\frac{\eta_k\|\nabla f(\boldsymbol{w}_{k,0})\|^2}{2\sqrt{D_1\|\nabla f(\boldsymbol{w}_{k,0})\|^2+D_0}+\sqrt{D_0}}-\sqrt{d}\eta_k g(\beta_2)\left(n-1+\frac{1+\beta_1}{1-\beta_1}\right)\sqrt{\frac{2n}{\beta_2^n}}\|\nabla f(\boldsymbol{w}_{k,0})\|.$$

We further discuss the case depending on whether $\|\nabla f(\boldsymbol{w}_{k,0})\|^2\leq\frac{D_0}{D_1}$ or not.

**Case I.1:** $\|\nabla f(\boldsymbol{w}_{k,0})\|^2\leq\frac{D_0}{D_1}$**.** In this case, the last line of the above equations can be further lower bounded by

$$\frac{\eta_k\|\nabla f(\boldsymbol{w}_{k,0})\|^2}{2\sqrt{D_1\|\nabla f(\boldsymbol{w}_{k,0})\|^2+D_0}+\sqrt{D_0}}-\sqrt{d}\eta_k g(\beta_2)\left(n-1+\frac{1+\beta_1}{1-\beta_1}\right)\sqrt{\frac{2n}{\beta_2^n}}\|\nabla f(\boldsymbol{w}_{k,0})\|$$

$$\geq\frac{\eta_k\|\nabla f(\boldsymbol{w}_{k,0})\|^2}{(2\sqrt{2}+1)\sqrt{D_0}}-\sqrt{d}\eta_k g(\beta_2)\left(n-1+\frac{1+\beta_1}{1-\beta_1}\right)\sqrt{\frac{2n}{\beta_2^n}}\|\nabla f(\boldsymbol{w}_{k,0})\|$$

$$=\eta_k\left(\frac{\|\nabla f(\boldsymbol{w}_{k,0})\|}{(2\sqrt{2}+1)\sqrt{D_0}}-\sqrt{d}g(\beta_2)\left(n-1+\frac{1+\beta_1}{1-\beta_1}\right)\sqrt{\frac{2n}{\beta_2^n}}\right)\|\nabla f(\boldsymbol{w}_{k,0})\|$$

**Case I.2:** $\|\nabla f(\boldsymbol{w}_{k,0})\|^2 > \frac{D_0}{D_1}$.

$$\frac{\eta_k\|\nabla f(\boldsymbol{w}_{k,0})\|^2}{2\sqrt{D_1\|\nabla f(\boldsymbol{w}_{k,0})\|^2 + D_0} + \sqrt{D_0}} - \sqrt{d}\eta_k g(\beta_2)\left(n - 1 + \frac{1+\beta_1}{1-\beta_1}\right)\sqrt{\frac{2n}{\beta_2^n}}\|\nabla f(\boldsymbol{w}_{k,0})\|$$

$$\geq \frac{\eta_k\|\nabla f(\boldsymbol{w}_{k,0})\|^2}{(2\sqrt{2}+1)\sqrt{D_1}\|\nabla f(\boldsymbol{w}_{k,0})\|} - \sqrt{d}\eta_k g(\beta_2)\left(n - 1 + \frac{1+\beta_1}{1-\beta_1}\right)\sqrt{\frac{2n}{\beta_2^n}}\|\nabla f(\boldsymbol{w}_{k,0})\|$$

$$= \eta_k\left(\frac{1}{(2\sqrt{2}+1)\sqrt{D_1}} - \sqrt{d}g(\beta_2)\left(n - 1 + \frac{1+\beta_1}{1-\beta_1}\right)\sqrt{\frac{2n}{\beta_2^n}}\right)\|\nabla f(\boldsymbol{w}_{k,0})\|$$

$$\overset{(*)}{\geq} \eta_k\frac{1}{2(2\sqrt{2}+1)\sqrt{D_1}}\|\nabla f(\boldsymbol{w}_{k,0})\|,$$

where Inequality $(*)$ is due to the constraint on $\beta_2$.

Therefore, we have either (1). there exists a iteration $k \in [T]$, such that

$$\|\nabla f(\boldsymbol{w}_{k,0})\| \leq 2\sqrt{d}(2\sqrt{2}+1)\sqrt{D_0}g(\beta_2)\left(n - 1 + \frac{1+\beta_1}{1-\beta_1}\right)\sqrt{\frac{2n}{\beta_2^n}},$$

or (2).for all $k \in [1, T]$,

$$\sum_{l\in[d]}\frac{\eta_k\partial_l f(\boldsymbol{w}_{k,0})^2}{2\max_{i\in[n]}|\partial_l f_i(\boldsymbol{w}_{k,0})| + \varepsilon} - \sum_{l\in[d]}\eta_k g(\beta_2)\left(n - 1 + \frac{1+\beta_1}{1-\beta_1}\right)\frac{|\partial_l f(\boldsymbol{w}_{k,0})|}{\sqrt{\frac{\beta_2^n}{2n}\max_{i\in[n]}|\partial_l f_i(\boldsymbol{w}_{k,0})|} + \varepsilon}\left(\max_{i\in[n]}|\partial_l f_i(\boldsymbol{w}_{k,0})|\right)$$

$$\geq \eta_k\frac{1}{2(2\sqrt{2}+1)}\min\left\{\frac{\|\nabla f(\boldsymbol{w}_{k,0})\|}{\sqrt{D_1}}, \frac{\|\nabla f(\boldsymbol{w}_{k,0})\|^2}{\sqrt{D_0}}\right\}.$$

**Case II:** $\varepsilon > \sqrt{D_0}$**.** In this case, we have that

$$\frac{\eta_k\|\nabla f(\boldsymbol{w}_{k,0})\|^2}{2\sqrt{D_1\|\nabla f(\boldsymbol{w}_{k,0})\|^2 + D_0} + \varepsilon} - \sum_{l\in[d]}\eta_k g(\beta_2)\left(n - 1 + \frac{1+\beta_1}{1-\beta_1}\right)\frac{|\partial_l f(\boldsymbol{w}_{k,0})|}{\sqrt{\frac{\beta_2^n}{2n}\max_{i\in[n]}|\partial_l f_i(\boldsymbol{w}_{k,0})|} + \varepsilon}\left(\max_{i\in[n]}|\partial_l f_i(\boldsymbol{w}_{k,0})|\right)$$

$$\geq \frac{\eta_k\|\nabla f(\boldsymbol{w}_{k,0})\|^2}{2\sqrt{D_1\|\nabla f(\boldsymbol{w}_{k,0})\|^2 + \varepsilon^2} + \varepsilon} - \sum_{l\in[d]}\eta_k g(\beta_2)\left(n - 1 + \frac{1+\beta_1}{1-\beta_1}\right)\frac{|\partial_l f(\boldsymbol{w}_{k,0})|}{\sqrt{\frac{\beta_2^n}{2n}\max_{i\in[n]}|\partial_l f_i(\boldsymbol{w}_{k,0})|} + \varepsilon}\left(\max_{i\in[n]}|\partial_l f_i(\boldsymbol{w}_{k,0})|\right).$$

Similar as **Case I**, we further divides the case regarding the value of $\|\nabla f(\boldsymbol{w}_{k,0})\|$.

**Case II.1:** $D_1\|\nabla f(\boldsymbol{w}_{k,0})\|^2 \leq \varepsilon^2$**.** In this case, we have

$$\frac{\eta_k\|\nabla f(\boldsymbol{w}_{k,0})\|^2}{2\sqrt{D_1\|\nabla f(\boldsymbol{w}_{k,0})\|^2 + \varepsilon^2} + \varepsilon} - \sum_{l\in[d]}\eta_k g(\beta_2)\left(n - 1 + \frac{1+\beta_1}{1-\beta_1}\right)\frac{|\partial_l f(\boldsymbol{w}_{k,0})|}{\sqrt{\frac{\beta_2^n}{2n}\max_{i\in[n]}|\partial_l f_i(\boldsymbol{w}_{k,0})|} + \varepsilon}\left(\max_{i\in[n]}|\partial_l f_i(\boldsymbol{w}_{k,0})|\right)$$

$$\geq \frac{\eta_k\|\nabla f(\boldsymbol{w}_{k,0})\|^2}{(2\sqrt{2}+1)\varepsilon} - \sum_{l\in[d]}\eta_k g(\beta_2)\left(n - 1 + \frac{1+\beta_1}{1-\beta_1}\right)\frac{|\partial_l f(\boldsymbol{w}_{k,0})|}{\varepsilon}\left(\max_{i\in[n]}|\partial_l f_i(\boldsymbol{w}_{k,0})|\right)$$

$$\geq \frac{\eta_k\|\nabla f(\boldsymbol{w}_{k,0})\|^2}{(2\sqrt{2}+1)\varepsilon} - \eta_k g(\beta_2)\left(n - 1 + \frac{1+\beta_1}{1-\beta_1}\right)\frac{\|\nabla f(\boldsymbol{w}_{k,0})\|}{\varepsilon}\sqrt{D_1\|\nabla f(\boldsymbol{w}_{k,0})\|^2 + D_0}$$

$$= \frac{\eta_k\|\nabla f(\boldsymbol{w}_{k,0})\|}{\varepsilon}\left(\frac{\|\nabla f(\boldsymbol{w}_{k,0})\|}{2\sqrt{2}+1} - g(\beta_2)\left(n - 1 + \frac{1+\beta_1}{1-\beta_1}\right)\sqrt{D_1\|\nabla f(\boldsymbol{w}_{k,0})\|^2 + D_0}\right).$$

**Case II.2:** $D_1\|\nabla f(\boldsymbol{w}_{k,0})\|^2 > \varepsilon^2$**.** This case is quite similar to **Case I.2**, and we have

$$
\frac{\eta_k\|\nabla f(\boldsymbol{w}_{k,0})\|^2}{2\sqrt{D_1\|\nabla f(\boldsymbol{w}_{k,0})\|^2 + \varepsilon^2} + \varepsilon} - \sum_{l\in[d]}\eta_k g(\beta_2)\left(n-1+\frac{1+\beta_1}{1-\beta_1}\right)\frac{|\partial_l f(\boldsymbol{w}_{k,0})|}{\sqrt{\frac{\beta_2^n}{2n}}\max_{i\in[n]}|\partial_l f_i(\boldsymbol{w}_{k,0})| + \varepsilon}\left(\max_{i\in[n]}|\partial_l f_i(\boldsymbol{w}_{k,0})|\right)
$$

$$
\geq \frac{\eta_k\|\nabla f(\boldsymbol{w}_{k,0})\|^2}{(2\sqrt{2}+1)\sqrt{D_1}\|\nabla f(\boldsymbol{w}_{k,0})\|} - \sum_{l\in[d]}\eta_k g(\beta_2)\left(n-1+\frac{1+\beta_1}{1-\beta_1}\right)\frac{|\partial_l f(\boldsymbol{w}_{k,0})|}{\sqrt{\frac{\beta_2^n}{2n}}\max_{i\in[n]}|\partial_l f_i(\boldsymbol{w}_{k,0})| + \varepsilon}\left(\max_{i\in[n]}|\partial_l f_i(\boldsymbol{w}_{k,0})|\right)
$$

$$
\geq \frac{\eta_k\|\nabla f(\boldsymbol{w}_{k,0})\|}{(2\sqrt{2}+1)\sqrt{D_1}} - \sqrt{d}\sqrt{\frac{2n}{\beta_2^n}}\eta_k g(\beta_2)\left(n-1+\frac{1+\beta_1}{1-\beta_1}\right)\|\nabla f(\boldsymbol{w}_{k,0})\|
$$

$$
= \eta_k\left(\frac{1}{(2\sqrt{2}+1)\sqrt{D_1}} - \sqrt{d}g(\beta_2)\left(n-1+\frac{1+\beta_1}{1-\beta_1}\right)\sqrt{\frac{2n}{\beta_2^n}}\right)\|\nabla f(\boldsymbol{w}_{k,0})\|
$$

$$
\geq \eta_k\frac{1}{2(2\sqrt{2}+1)\sqrt{D_1}}\|\nabla f(\boldsymbol{w}_{k,0})\|.
$$

Therefore, we have either (1). there exists a iteration $k \in [T]$, such that

$$
\|\nabla f(\boldsymbol{w}_{k,0})\| \leq 2\sqrt{d}(2\sqrt{2}+1)\sqrt{D_0}g(\beta_2)\left(n-1+\frac{1+\beta_1}{1-\beta_1}\right)\sqrt{\frac{2n}{\beta_2^n}},
$$

or (2). for all $k \in [1, T]$,

$$
\sum_{l\in[d]}\frac{\eta_k\partial_l f(\boldsymbol{w}_{k,0})^2}{2\max_{i\in[n]}|\partial_l f_i(\boldsymbol{w}_{k,0})| + \varepsilon} - \sum_{l\in[d]}\eta_k g(\beta_2)\left(n-1+\frac{1+\beta_1}{1-\beta_1}\right)\frac{|\partial_l f(\boldsymbol{w}_{k,0})|}{\sqrt{\frac{\beta_2^n}{2n}}\max_{i\in[n]}|\partial_l f_i(\boldsymbol{w}_{k,0})| + \varepsilon}\left(\max_{i\in[n]}|\partial_l f_i(\boldsymbol{w}_{k,0})|\right)
$$

$$
\geq \eta_k\frac{1}{2(2\sqrt{2}+1)}\min\left\{\frac{\|\nabla f(\boldsymbol{w}_{k,0})\|}{\sqrt{D_1}}, \frac{\|\nabla f(\boldsymbol{w}_{k,0})\|^2}{\varepsilon}\right\}.
$$

**As a conclusion of** **Case I** **and** **Case II**, we have that either there exists a iteration $k \in [T]$, such that

$$
\|\nabla f(\boldsymbol{w}_{k,0})\| \leq 2\sqrt{d}(2\sqrt{2}+1)\sqrt{D_0}g(\beta_2)\left(n-1+\frac{1+\beta_1}{1-\beta_1}\right)\sqrt{\frac{2n}{\beta_2^n}},
$$

or for all iteration $k \in [1, T]$, we have that

$$
\sum_{l\in[d]}\frac{\eta_k\partial_l f(\boldsymbol{w}_{k,0})^2}{2\max_{i\in[n]}|\partial_l f_i(\boldsymbol{w}_{k,0})| + \varepsilon} - \sum_{l\in[d]}\eta_k g(\beta_2)\left(n-1+\frac{1+\beta_1}{1-\beta_1}\right)\frac{|\partial_l f(\boldsymbol{w}_{k,0})|}{\sqrt{\frac{\beta_2^n}{2n}}\max_{i\in[n]}|\partial_l f_i(\boldsymbol{w}_{k,0})| + \varepsilon}\left(\max_{i\in[n]}|\partial_l f_i(\boldsymbol{w}_{k,0})|\right)
$$

$$
\geq \eta_k\frac{1}{2(2\sqrt{2}+1)}\min\left\{\frac{\|\nabla f(\boldsymbol{w}_{k,0})\|}{\sqrt{D_1}}, \frac{\|\nabla f(\boldsymbol{w}_{k,0})\|^2}{\varepsilon + \sqrt{D_0}}\right\}.
$$

The proof is completed. $\qquad\square$

# E EXPERIMENT DETAILS

This section collects experiments and their corresponding settings, and is arranged as follows: to begin with, we show that Adam works well under the different reshuffling order; we then provide the experiment settings of Figure 1.

## E.1 ADAM WORKS WELL UNDER DIFFERENT RESHUFFLING ORDER

We run Adam on ResNet 110 for CIFAR 10 across different random seeds and plot the 10-run mean and variance in Figure 4. One can observe that the performance of Adam is robust with respect to random seed, and support Theorem 1 in terms of trajectory-wise convergence. The experiment is based on this repo, where we adopt the default hyperparameters settings.

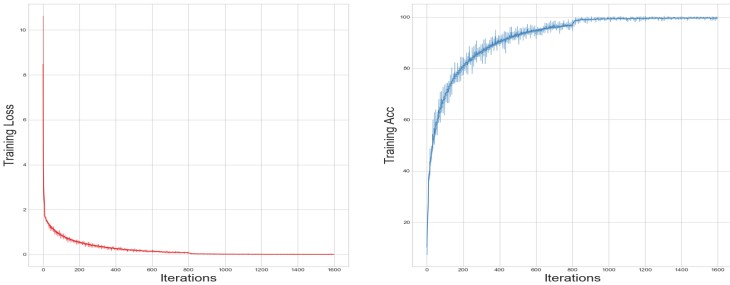

Figure 4: Performance of Adam with different shuffling orders. We respectively plot the training loss and the training accuracy of Adam together with their variances over 10 runs with different random shuffling order. The result indicate the performance of Adam is robust w.r.t. the shuffling order.

## E.2 LOCAL SMOOTHNESS VS. GRADIENT NORM

In this section, we provide the models and hyperparameter settings of Figures 1. We will also illustrate how we evaluate the local smoothness.

**Models and hyper-parameter settings in Figures 1.** In Figure 1, we use exactly the same setting as Vaswani et al. (2017) on WMT 2014 dataset, based on this repo.

**How we evaluate the local smoothness.** We use the same method as Zhang et al. (2019a). Specifically, with a finite-difference step $\alpha$, we calculate the smoothness at $\boldsymbol{w}_k$ as

$$\text{local smoothness} = \max_{\gamma \in \{\alpha, 2\alpha, \cdots, 1\}} \frac{\|\nabla f(\boldsymbol{w}_k + \gamma(\boldsymbol{w}_{k+1} - \boldsymbol{w}_k)) - \nabla f(\boldsymbol{w}_k)\|}{\gamma \|\boldsymbol{w}_{k+1} - \boldsymbol{w}_k\|}.$$

