# OpenReview forum: "Provable Adaptivity in Adam"
_ICLR.cc/2023/Conference — Submitted to ICLR 2023_

### Official Review · Reviewer_z7aZ · 2022-10-23

**Confidence:** 3
**Clarity, Quality, Novelty And Reproducibility:** I have several concerns about this pa…
**Correctness:** 2
**Technical Novelty And Significance:** 3
**Empirical Novelty And Significance:** 3
**Recommendation:** 5

**Strength And Weaknesses:**

I have the following problems/concerns, and I hope the authors could help me understand these points:


a) For Theorem 1, I have the following problems/concerns:

1.	Randomness of the algorithm: In page 4, the authors say that the algorithm is actually a double-loop algorithm with stochastic shuffling. However, Theorem 1 holds without expectation or high probability, which is odd. Moreover, in which part of the proof the shuffling is used? The authors say “random shuffling is not a key factor in our analysis. The proof also works for other settings.”Can the authors be more specific about what are the “other settings”? Finally, isn't shuffling also a modification of Adam, which makes the algorithm not a pure Adam algorithm?

2.	Dependance on n and d: From the Appendix, it seems that the upper bound is linear wrt n and d. Can the authors be clearer about this?

3.	The inequality in (4): It is not clear to me why we can get this inequality. Where is rootd and n? Moreover, delta is upper bounded by 1/sqrt{D_1}. Thus, when D_1=0 in Assumption 2 (which only means the sum of the norm of the gradients is upper bounded), delta will explode?


4.     Theorem 1 does not assume beta_2<1, which seems incorrect.


b) **I have the following major concern** about writing: After a comparison, I find that many paragraphs here are basically copied/paraphrased from a previous paper (Zhang et al., 2022). For example, one can compare the two paragraphs of this paper under Assumption 2 and those under Assumption 2.2 of Zhang et al, (2022). Although the two papers are strongly related, I do not think this way of writing is appropriate.

c. Other problems:


1. Proof of Lemma 1: second equality: I do not see how to obtain this equality. Why is v_l (second order momentum of the l-th dimension) related to the max_{[n]} of abs(gradient)?


2. Relation between Assumptions 1 and 2: when n=1, then Assumption 2 implies \emph{bounded gradient} when D_1\in(0,1] (it can also extend to n with D_1\leq 1/n). So Assumption 1 only makes sense when D_1 is large.

3. In page 17, it seems that all C_i are non-decreasing wrt n_1. So in this case why don't one just set eta_1=0? Is there a term related to 1/eta_1?

4. The experiments are not very sufficient. As mentioned before, the authors did make some modifications to Adam (the shuffling and double-loop), which makes it not exactly the original Adam.





**Summary Of The Paper:**

This paper studies well-known Adam algorithm. Previous work shows that Adam can converge to a neighborhood of stationary point if the gradients are bounded (functions are smooth). In this paper, the authors show that, Adam can guarantees a even better results under a milder condition, i.e., L_0,L_1 smoothness, under which the gradients can be unbounded. Moreover, SGD may diverge under this condition.




**Summary Of The Review:**

I do like this paper, but I have several concerns about this paper, which are listed above.

---

> ### Author Response · Authors · 2022-11-15
> **(Part 1) Thank you for your feedbacks and constructive comments**
>
> We thank the reviewer for the constructive feedback. We have revised the paper accordingly (marked as blue). The concerns are dually addressed below.
>
> **Q1**：Theorem 1 holds without expectation or high probability, which is odd.
>
> **A1**： Our convergence result holds for every trajectory.  Such a result is standard for optimizers with shuffling (see (Gurbuzbalaban et al. 2015; Shi et al. 2020) as examples), and it is not odd.  Instead, it is strictly stronger than  the results of convergence "with high probability" or "in expectation."
>
>
>
> **Q2**： Moreover, in which part of the proof the shuffling is used?  Can the authors be more specific about what are the “other settings”?
>
> **A2**： In analysis, the property of shuffling is used in Lemma 3, where we used the property that “each f_i occurs once and only once in each epoch". This property holds for shuffling.
>
> Moreover, we would like to point out that this property is not limited to the stochastic shuffling setting. It also holds for many other variants of shuffling. For instance, it also holds for cyclic ordering (123 123), shuffle-once-then-cyclic (213 213), and mountain-climbing (123 321). These variants are also used in practical applications such as NLP tasks.
>
> **Q3**： Finally, isn't shuffling also a modification of Adam, which makes the algorithm not a pure Adam algorithm?”
>
> **A3**：
> We believe shuffling is not a modification.  First, it does not change the update rule of Adam (unlike other variants such as AdaBound and AMSgrad). Second, shuffling Adam is the default setting in practice: in both Pytorch and Tensorflow, the pure Adam is implemented in the random shuffling fashion and it is used for various tasks such as computer vision, NLP, and generative models, etc.. We choose to analyze the shuffling version since it is closer to practice.  We have added a more relevant discussion to show our motivation.
>
>
>
>
> **Q4**: It seems that the upper bound is linear wrt n and d. Can the authors be clearer about this?
>
> **A4**: The constant terms in the upper bound is in the order of $\frac{3}{2}$ for $d$ and  $\frac{3}{2}$ for $n$. This can be seen in the last inequality above Lemma 6 in Appendix D.2 (with the definition of constants listed in Appendix D.1.1). We will add more clearer discussion in Appendix D.2.
>
> For completeness, we would like to emphasize that our contribution is   “providing the first convergence bound of Adam with $(L_0,L_1)$-smooth condition”, but we agree the bound itself may not be the tightest. We would like to also mention that in existing literature, the upper bounds for Adam has similar dependence over $n$ and $d$ even under $L$-smooth condition (c.f. (Shi et al. 2020; Zhang et al. 2022)).  Proving tighter bounds is an interesting but challenging topic and we leave it as future work.
>
> **Q5**： The inequality in (4): It is not clear to me why we can get this inequality. Where is rootd and n?
>
> **A5**: Thanks for asking. In a word, the $\sqrt{d}$ comes from the coodinate-wise calculation of the error term, and $n$ comes from the gap between $f_i$ and $f$. More formally, the intuition of (4) is that when $\beta_2$ is close to $1$, $\nu_{k,i}=\beta_2 \nu_{k,i-1}+(1-\beta_2)\nabla f_{\tau_{k,i}}(w_{k,i})^2$ is close to $\nu_{k,i-1}$. Therefore, when bounding the change of $f(u_{k,0})$, except the descent term $-\eta_k\Vert \nabla f(w_{k,0})/\sqrt[4]{\nu_{k,0}}+\varepsilon\Vert^2$, there is an error term caused by the approximation of $\nu_{k,i}$ to $\nu_{k,0}$. The negative term is in order of $-\delta(\beta_2)\times\eta_k\sum_i\sum_l\vert \nabla_l f_i(w_{k,0})/\sqrt[4]{\nu_{l,k,0}}+\varepsilon\vert^2$ as it is derived by calculating the coordinate-wise (introducing the sum of l) and step-wise error (introducing the sum of i). Transferring the error term to the form of the descent term requires the use of Cauchy's inequality and introduces n and $\sqrt{d}$.

---

> ### Author Response · Authors · 2022-11-15
> **(Part 2) Thank you for your feedbacks and constructive comments**
>
> **Q6**： Moreover, delta is upper bounded by $1/\sqrt{D_1}$. Thus, when $D_1=0$ in Assumption 2 (which only means the sum of the norm of the gradients is upper bounded), $\delta$ will explode?
>
> **A6**： We respectfully point out that :  $D_1=0$  will cause the explosion of “the upper bound of $\delta$” instead of “the explosion of $\delta$”. In this case, our convergence result actually allows more choices of beta2. We explain as follows.
>
> To ensure convergence, we require
>
> **Condition 1:** $\delta (\beta_2) < 1/\sqrt{D_1}$.
>
> We need to properly choose $\beta_2$ such that Condition 1 holds. To do so, we consider the following two cases:
>
> **Case 1:** When $D_1 > 0$,  the r.h.s is a finite constant. Since $\delta (\beta_2)$ approaches 0 when $\beta_2$ approaches 1, we need to choose large enough  $\beta_2$ to ensure Condition 1 holds. This is the statement in Theorem 1.
>
> **Case 2:**  When $D_1 = 0$,   the r.h.s, becomes infinity while $\delta (\beta_2)$ is always bounded for any $\beta_2$ in [0,1). That is to say, we can choose any $\beta_2$ in [0,1) to ensure convergence. This result makes sense because: when $D_1 = 0$, Assumption 2 reduces to the bounded gradient condition. Further, when the gradient is bounded, Assumption 1 reduces to $L$-smooth condition (see the discussion under Assumption 1). That is to say, we now deal with the case with L-smooth and bounded gradient. In this simplified setting, Adam is shown to converge with any beta2 in [0,1) in the existing literature such as [Defossez et al. 2020]. So our result coincides with existing results in this setting.  In other words, our convergence result is more general and the existing result can be viewed as a special case of our Theorem 1 under a simplified setting ($L$-smooth and bounded gradient)
>
>
> **Q7**: I find that many paragraphs here are basically copied/paraphrased from a previous paper”
>
> A7: Thanks for pointing that out. We have rewritten the discussion about Assumption 2 carefully.
>
>
> **Q8**: Other problems.
>
> **Q8.1**: Proof of Lemma 1: second equality: I do not see how to obtain this equality. Why is v_l (second order momentum of the $l$-th dimension) related to the max_{[n]} of abs(gradient)?
>
> **A8.1**: In the this equality, we use $\nu_{l,1,-1}=\max_j\{\partial_l f_j(w_0)^2\}$ which is the initialization of $\nu_{l,1,-1}$. Such initialization is used for simplicity without loss of generality, as the information of the initialization in the exponentially decayed average of Adam (both in $m_{k,i}$ and $\nu_{k,i}$) decays rapidly with k increasing. We have already clarified this at the beginning of Appendix D in the original version of this paper, and we invite the reviewer to check there for details.
>
> **Q8.2**: Assumption 2 only makes sense when $D_1$ is large.
>
> **A8.2**: We do agree that when $D_1$ is small, Assumption 2 degenerates to the bounded gradient assumption. However, our Theorem applies to every $D_0,D_1\ge 0$, which contains a broad variety of problems.
>
> **Q8.3**: In page 17, it seems that all C_i are non-decreasing wrt \eta_1. So in this case why doesn't one just set eta_1=0? Is there a term related to 1/eta_1?
>
> **A8.3**: Thanks for asking and there is indeed a term related to $1/\eta_1$. When proving Theorem 1, we have the following intermediate results
> $$\frac{1}{T}\sum_{k=1}^T
>     \min\{ \frac{ \Vert \nabla f(w_{k,0})\Vert}{\sqrt{D_1}}, \frac{ \Vert \nabla f(w_{k,0})\Vert^2}{\varepsilon+\sqrt{D_0}}
>    \}\le O(\frac{f(u_{1})-f(u_{T+1})}{\eta_1\sqrt{T}})+O(\frac{\log T}{\sqrt{T}})+O(\min[ \delta(\beta_2), \delta(\beta_2)^2 ]).$$
>
> Therefore, when $\eta_1=0$, the term  $O(\frac{f(u_{1})-f(u_{T+1})}{\eta_1\sqrt{T}})$ is infinite and the bound does not convey anything.
>
> **Q8.4**: The experiments are not very sufficient. As mentioned before, the authors did make some modifications to Adam (the shuffling and double-loop), which makes it not exactly the original Adam.
>
> **A8.4**: Please refer to **A3** for details. We use the PyTorch implementation of Adam for experiments, which exactly agrees with our definition of Adam.
>
> **References**:
>
> Gurbuzbalaban et al, Convergence Rate of Incremental Gradient and Incremental Newton Methods, 2015
>
> Zhang et al., Adam can converge without any modification on update rules, 2022
>
> Shi et al.,  RMSProp converges with proper hyper- parameter, 2021
>
> Deffosez et al., A Simple Convergence Proof of Adam and Adagrad, 2019

---

> ### Author Response · Authors · 2022-12-09
> **Any concern remains?**
>
> We would like to thank the reviewer for your constructive comments, and we appreciate the time you put into handling our paper. Given our detailed reply, please let us know if your concerns remain. Are there any additional questions we can answer to facilitate a fair reevaluation of this work?
>
> Thanks,
>
> Paper 388 authors

---

### Official Review · Reviewer_8ES2 · 2022-10-24

**Confidence:** 3
**Correctness:** 3
**Technical Novelty And Significance:** 2
**Empirical Novelty And Significance:** Not applicable
**Recommendation:** 3

**Clarity, Quality, Novelty And Reproducibility:**

For writing, most parts of this work are well-written and clear. But some key problems, such as the above ones,  are not well explained.

For novelty, since the key assumption comes from others, and it seems standard optimization can be directly adopted here, the novelty of this work is not high. The most severe issue is that it does not provide new insights since it is well-known that the second-order moment could help the convergence of Adam.


**Details Of Ethics Concerns:**

N.A.

**Strength And Weaknesses:**

Strength:
There are main two contributions in this work.
1)It proves the convergence of Adam under the (L0,L1)-smoothness condition.

2)Under the same assumption, it uses examples to show the advantage of Adam over GD and SGD.

Weaknesses:
1)The main assumption is borrowed from other works but is actually rarely used in the optimization field. Moreover, the benefits of this assumption is not well investigated. For example, a) why it is more reasonable than the previous one? B) why it can add  gradient norm L_1||\nabla f(w_1)|| in Eqn (3) or why we do not add other term?  It should be mentioned that a milder condition does not mean it is better, since it may not reflect the truth. For me, problem B) is especially important in this work, since the authors do not well explain and investigate it.

2)Results in Theorem 1 show that Adam actually does not converge, since this is a constant term O(D_0^{0.5}\delta) in Eqn. (5). This is not intuitive, the authors claim it is because the learning rate may not diminish. But many previous works, e.g. [ref 1],  can prove Adam-type algorithms can converge even using a constant learning rate.  Of course, they use the standard smooth condition. But (L0,L1)-smoothness condition should not cause this kind of convergence, since for nonconvex problem, in most cases, we only need the learning rate to be small but does not care whether it diminishes to zero.

[ref 1] Dongruo Zhou, Jinghui Chen, et al. On the Convergence of Adaptive Gradient Methods for Nonconvex Optimization

3)It is not clear what are the challenges when the authors analyze Adam under the (L0,L1)-smoothness condition. It seems one can directly apply standard analysis on the (L0,L1)-smoothness condition.  So it is better to explain the challenges, especially the difference between this one and Zhang et al.

4)Under the same assumption, the authors use examples to show the advantage of Adam over GD and SGD. This is good. But one issue is that is the example reasonable or does it share similar properties with practical problems, especially for networks. This is important since both SGD and ADAM are widely used in the deep learning field.

5)In the work, when comparing SGD and ADAM, the authors explain the advantage of adam comes from the cases when the local smoothness varies drastically across the domain. It is not very clear for me why Adam could better handle this case. Maybe one intuitive example could help.

6)The most important problem is that this work does not provide new insights, since it is well known that the second order moment could help the convergence of Adam. This work does not provide any insights beyond this point and also does not give any practical solution to further improve.

**Summary Of The Paper:**

In this work, the authors mainly prove the convergence of Adam under the (L0,L1)-smoothness condition which is borrowed from existing work. Moreover, they use examples to show that GD and SGD can converge much slower than Adam under this assumptions. The reason behind for this advantage of Adam contributes to its usage on the local smoothness which is called “adaptivity”.

**Summary Of The Review:**

Although this work provides some good theoretical analysis under the (L_0, L1) smoothness assumptions, it does not provide good new insights and practical improvements.  Moreover, some key points are not well explained.

---

> ### Author Response · Authors · 2022-11-15
> **(Part 1) Thank you for your feedbacks and constructive comments**
>
> Thanks for your valuable comments and suggestions. We address your concern as follows.
>
> **Q1**: The benefits of this assumption are not well investigated. For example, a) why it is more reasonable than the previous one? b) why it can add a gradient norm or why we do not add other terms?
>
>
> **A1**: Thanks for asking.
>
> **Regarding "Why it is more reasonable"**: We answer this question both from the perspective why $L$-smooth condition is less reasonable and why $(L_0,L_1)$-smooth condition is more reasonable.
>
> **1. Why $L$-smooth condition is less reasonable**: It is empirically observed that the smoothness varies sharply in many deep learning tasks (Zhang et al., 2019; 2020; Crawshaw et al., 2022).
>
> **2. Why $(L_0,L_1)$-smooth condition is more reasonable**:  This is because Assumption 1 is observed to hold in deep learning tasks such as LSTM and Transformers. More  empirical evidence  can be in Figure 1 as well as the figures in (Zhang et al., 2019; 2020; Crawshaw et al., 2022)
>
> We have emphasized more in the revised version that $(L_0,L_1)$-smooth condition is closer to practice,  which better reveals the truth.
>
>
> **Regarding "Why we do not add other terms?":**
>
> According to the experiments such as Figure 1 and those in (Zhang et al. 2019), adding a linear term on the gradient norm is already sufficient to capture the real situation. This is why we only consider adding a linear term.
>
> Adding other terms is also an interesting research direction if the new terms can better reflect the practical situation.
>
>
>
> **Q2**: Converging to a bounded region is not intuitive. Zhou et al. (2018) prove that Adam-type algorithms can converge even using a constant learning rate under $L$-smooth condition. $(L0,L1)$-smooth condition should not cause this kind of convergence.
>
>
>
> **A2**: We agree that “$(L_0,L_1)$-smoothness condition should not cause this kind of convergence (to bounded region)”. Converging to a bounded region also happens for Adam under regular $L$-smooth condition. This is theoretically shown by (Zhang et al. 2022; Shi et al. 2021, Zaheer et al., 2018), which all consider L-smooth condition. Further, convergence to the bounded region (instead of stationary points)  is also empirically verified in Figure 2. In summary, convergence to the bounded region is not due to the new smoothness condition. Rather, it is an intrinsic property of Adam.
>
>
> As for  (Zhou et al. 2018), we would like to point out that they focus on AMSGrad instead of Adam. This difference is crucial: AMSGrad forces 1/vk to decrease along the trajectory, so the effective stepsize is monotonically decreasing even with a constant stepsize. Therefore, AMSGrad can converge to stationary points as shown by (Zhou et al. 2018). However, as argued in Remark 2 of our revised draft,  this is not true for Adam: when the gradient norm decreases, $\frac{1}{\sqrt{\nu_{k,i}}+\varepsilon 1_d}$ increases, so the effective stepsize  $\frac{\eta_k}{\sqrt{\nu_{k,i}}+\varepsilon 1_d}$  might not decrease.
>
>
>
> **Q3**: What are the challenges when the authors analyze Adam under $(L0,L1)$-smoothness condition. It is better to explain the challenges, especially the difference between this one and Zhang et al.
>
> **A3**: We summarize the proof idea in Section 4.2. Specifically, the challenge brought by $(L_0,L_1)$-smooth condition is discussed in Remark 6. We briefly explain here: Under L-smooth condition as in (Zhang et al. 2022), the difference between $\nu_{k,i}$ can be easily controlled by the constant L and the diminishing stepsize. While under $(L_0,L_1)$-smooth condition, the changes between $\nu_{k,i}$ are quite complicated with historical gradient information involved. Furthermore, the error brought by the change of $\nu_{k,i}$ is itself bounded under $L$-smooth condition, while it is unbounded under $(L_0,L_1)$-smooth condition and may even prevent the objective function from decreasing.  We address these difficulties and explain how to solve them in [Stage I, Section 4.2].
>
>
>
> **Q4**: Regarding the counter-example of SGD, “is that is the example reasonable or does it share similar properties with practical problems, especially for networks”.
>
> **A4**: Yes, we believe this example is reasonable.  Firstly, this counter-example considers minimizing the cross-entropy function composed with a linear function (see Appendix C for the description).  This example shares similar properties with practical **classification problems** since cross-entropy is the default choice of loss function for classification.   For the simplicity of analysis, we use linear models rather than practical neural nets in this counter-example. Unfortunately, SGD performs badly even on this simple linear classification problem.
>
> We have added more relevant discussions to the script.

---

> ### Author Response · Authors · 2022-11-15
> **(Part 2) Thank you for your feedbacks and constructive comments**
>
> **Q5**: In the work, when comparing SGD and ADAM, the authors explain the advantage of adam comes from the cases when the local smoothness varies drastically across the domain. It is not very clear for me why Adam could better handle this case. Maybe one intuitive example could help.
>
>
> **A5**: We provide intuition as follows:
>
> For SGD, its stepsize is pre-determined before the algorithm is deployed, often chosen to be a constant smaller than $2/L$ ($L$ is the global smoothness) or chosen to be diminishing. When the local smoothness coefficient L  varies drastically in the domain,  it is difficult to decide a  pre-decide stepsize that guarantees to work along SGD trajectories. To better understand, we illustrate the following two cases for SGD:
>
> **Case 1:** When entering a sharp local region, the pre-determined steps might be too large and cause divergence of SGD.
>
> **Case 2:** When entering a flat region, the pre-determined steps might be too small to make progress. It will cause slow convergence of SGD. The situation gets worse when using diminishing steps.
>
> For Adam, the effective stepsize involves $\frac{1}{\sqrt{\nu_{k,i}}+\varepsilon1_d}$, which is adaptive changing along the trajectory. Adam can better handle the above two cases:
>
> **Case 1:** When entering the sharp local region, the gradient is usually large and $\frac{1}{\sqrt{\nu_{k,i}}+\varepsilon1_d}$ will gradually decrease in this region.  Finally, it will reach a small enough step size to slip into the sharp region.
>
> **Case 2:** When entering the flat region, the gradient is consistently small and thus $\frac{1}{\sqrt{\nu_{k,i}}+\varepsilon1_d}$ will gradually increase, leading to larger stepsize and thus Adam will converge faster.
>
>
> We will add more intuitive discussion to the script.
>
>
> **Q6**： The most important problem is that this work does not provide new insights, since it is well-known that the second-order moment could help the convergence of Adam.
>
> **A6**: We agree that the benefit of second-order-moment of Adam is **empirically observed** in many DL tasks. However, we respectfully point out that the “benefit” is not “known” in theory. In fact, the existing theoretical understanding of second-order-moment is mostly negative. For instance, Reddi et al. (2018) pointed out that small $\beta_2$ (a hyperparameter to control second-order-moment) might leads to divergence. Some recent works then argue that large $\beta_2$ can still lead to convergence, but the rate is no better than SGD. Up to this work, the benefit of second-order-moment is unclear. We step forward the first step on this topic.
>
>
> **Q7**: This work does not provide any insights beyond this point.
>
> **A7**: We believe our result provides new insights that second-order-moment is helpful when the local smoothness varies drastically across the domain. In particular, we provide the following insights:
>
> 1) When $\beta_2 =1$, there is no second-order-moment, and Adam is reduced to SGD, which is bad when the global L-smoothness condition fails (Section 4.3).
>
> 2) When $\beta_2 <1$,   second-order-moment is introduced,  Adam converges well under the same setting above, justifying the advantage of second-order-moment.
>
>
>
> **Q8**: This work does not give any practical solution to further improve.
>
> **A8**: We agree that our analysis does not directly lead to a new algorithm design.  However, we still think our result is important for the following reasons:
>
> First, Adam receives great popularity among practitioners (with more than 100k citations). It is important to theoretically understand this current algorithm.
>
> Second, we provide new suggestions for practitioners: When running experiments on tasks such as Transformers and LSTM training, we suggest using Adam instead of SGD. Though this is a folk result based on engineering experience, it is firstly theoretically justified. Third, we provide suggestions for hyperparameter tuning (based on the convergence conditions in theorem 1): when running Adam, we suggest tune-up $\beta_2$ and trying different $\beta_1<\sqrt{\beta_2}$. This suggestion would save much effort of grid-searching the $(\beta_1,\beta_2)$ combination.
>
> We have modified the script accordingly and included the relevant discussions.

---

> ### Author Response · Authors · 2022-11-15
> **(Part 3) References**
>
> Reddi et al., On the Convergence of Adam and Beyond, 2018
>
> Dongruo Zhou, Jinghui Chen, et al. On the Convergence of Adaptive Gradient Methods for Nonconvex Optimization, 2018
>
> Zhang et al., Why gradient clipping accelerates training: A theoretical justification for adaptivity, 2019
>
> Zhang et al., Improved Analysis of Clipping Algorithms for Non-convex Optimization, 2020

---

> ### Author Response · Authors · 2022-12-09
> **Any concern remains?**
>
> We would like to thank the reviewer for your constructive comments, and we appreciate the time you put into handling our paper. Given our detailed reply, please let us know if your concerns remain. Are there any additional questions we can answer to facilitate a fair reevaluation of this work?
>
> Thanks,
>
> Paper 388 authors

---

### Official Review · Reviewer_8djk · 2022-10-25

**Confidence:** 4
**Correctness:** 3
**Technical Novelty And Significance:** 3
**Empirical Novelty And Significance:** 2
**Recommendation:** 5

**Clarity, Quality, Novelty And Reproducibility:**

The three paragraphs after Assumption 2 are very similar to the paragraphs after Assumption 2.2 in [1]. The authors should rephrase these parts.

**Strength And Weaknesses:**

Strengths:

S1. The work establishes the convergence property of Adam under the ($L_0, L_1$)-smooth condition that is weaker than the $L$-smooth condition.
S2.  It provides counter-examples where GD/SGD can diverge under the ($L_0, L_1$)-smooth condition, which shows the advantage of Adam over GD/SGD.

Weakness:
1. The main issue lies in the experimental part although the main point of the paper is its theoretical analysis. At the beginning of the paper, it mentions the examples of deep neural networks that satisfy ($L_0, L_1$)-smoothness requirements. But, the experiments only consider a quadratic function that is also $L$-smooth. It should add some more complex experiments that match the motivations of the paper.

2. The paper should give a detailed comparison with recent related works including [1,2,3,4,5]. In [1], similar convergence results are established under the same choice of ($\beta_1,\beta_2$). It should clarify the technical difficulty under this new smoothness constraint. In [2], the weakness of SGD under the ($L_0, L_1$)-smooth condition has been studied. Though this work considers the diminishing stepsize, the significance should be discussed. The potential function in Section~4.2 is a widely-used trick in the analysis of the momentum-based methods, e.g., see [3,4,5]. The authors could provide some intuitive ideas behind the construction of the potential function and analyze its difference from other momentum-based methods.

3. More theoretical comparisons between Adam and GD/SGD can be discussed.:1) both GD/SGD and Adam use the constant learning rate; 2) If the hyperparameters of Adam are chosen improperly, the convergence of Adam can be arbitrarily slow; 3) The convergence rate of Adam in Theorem~1 depends on the dimension $d$, while GD/SGD does not depend on it.

4. As the paper contains many lemmas and theorems. It can be better organized for readers, especially for the appendix part. More details can be added for some inequalities e.g. the first inequality in the proof of Lemma 2.
In the second paragraph on Page 1, the statement ``larger $\beta_1$ and $\beta_2$ will bring more historical information into the update'' is rather vague. The authors can add explanations for the historical information.

5. It could be good if the author can add some discussions on below questions:

(1) The theorem suggests that $\beta_2 = 1$ is superior for Adam since $\delta(\beta_2) = 0$ in this case. Does it mean $\beta_2$ is redundant for Adam? If not, can you provide more evidence of the effect of $\beta_2$ in theory or with experiments?

(2) In practice, it is common to use a constant learning rate in a stage and it is known that GD converges with constant stepsize. So what is the convergence result of Adam when a constant learning rate is used?

(3) Since gradient clipping is proven to be beneficial for GD/SGD [2], what is the advantage of Adam over the GD/SGD with gradient clipping or normalized gradient?

(4) Since pretrained model is common, where the initialization is good, what is the advantage of Adam over SGD in this case? Note that in the last equality of Page 7, it may assume that $\|\nabla f(w_{k,0})\|_2$ is large, so the cubic term of the gradient is dominant.

But, in practice, when the initialization is good, i.e. $\|\nabla f(w_{k,0})\|_2$ is small, more discussion can be added.

(5) In Page~7, the paragraph above Inequality (8): ``further notice that $u_{k,i}$ is close to $w_{k,i}$''. Can you give more explanation of this insight?


[1] Yushun Zhang, et al. Adam can converge without any modification on update rules. Advances in Neural Information Processing Systems, 2022.

[2] Jingzhao Zhang, et al. Why gradient clipping accelerates training: A theoretical justification for adaptivity. In International Conference on Learning Representations, 2019.

[3] Euhanna Ghadimi, et al. Global convergence of the Heavy-ball method for convex optimization.  European Control Conference, 2015.
[4] Yanli Liu et al. An Improved Analysis of Stochastic Gradient Descent with Momentum. Advances in Neural Information Processing Systems, 2020.

[5] Xiangyi Chen, et al. On the convergence of a class of Adam-type algorithms for non-convex optimization. International Conference on Learning Representations, 2019.

[6] Naichen Shi, Dawei Li, Mingyi Hong, and Ruoyu Sun. RMSProp converges with proper hyper-
parameter. In International Conference on Learning Representations, 2021.


**Summary Of The Paper:**

Adam is a popular optimizer for training neural networks in deep learning. This paper analyzes the convergence of Adam under the so-called ($L_0, L_1$)-smooth condition, and shows that Adam has a convergence guarantee without the bounded gradient assumption. They also provide counter-examples where GD/SGD can diverge under the ($L_0, L_1$)-smooth condition. A toy example is used to test the main theory.

**Summary Of The Review:**

The paper works on an important problem related to the convergence of Adam and shows the potential advantages over GD/SGD. I hope the authors add more experiments and more analysis in the revision for better reading and deeper understanding.

---

> ### Author Response · Authors · 2022-11-15
> **(Part 1) Thank you for your feedbacks and constructive comments**
>
> We thank the reviewer for the constructive feedback. We have revised the paper accordingly (marked as blue). The concerns are dually addressed below
>
> **Q1**: The experiments only consider a quadratic function that is also $L$-smooth. It should add some more complex experiments that match the motivations of the paper.
>
> **A1**：We would like to clarify that **we do have complex experiments to support the motivation of the paper**,  which are shown in Figure 1. Specifically, Figure 1. (a) shows that Adam converges faster than SGD for the transformer, and Figure 1. (b) shows that $(L_0,L_1)$-smooth condition holds in the same setting. Combined together, Figure 1 (a) and (b) well support our motivation: for practical deep learning tasks,  $(L_0,L_1)$-smooth condition is a more proper assumption than L-smooth condition, and Adam performs better than SGD under $(L_0,L_1)$-smooth condition.
>
> On the other hand, the quadratic function in Figure 2 is to verify one claim in Theorem 1 that Adam may not converge to a stationary point with $D_0\ne0$. This point is not related to the smoothness assumption but rather an intrinsic property of Adam. As $L$-smooth condition is stronger than $(L_0,L_1)$-smooth condition, choosing a quadratic function won’t undermine our verification but eases the illustration
>
> **Q2**: Comparison to related works [1,2,3,4,5].
>
> **A2**:
> **As for [1]**: There are two major differences between their result and ours.
>
>  First, we derive the result under (L_0,L_1)-smooth condition instead of $L$-smooth condition in [1]. The technical challenge is discussed in [Stage I, Section 4.2] in details and we briefly summarize it here. An important step to prove Adam's convergence is to bound the change of $\nu_{k,i}$. Under L-smooth condition as in [Zhang et al. 2022], the difference between $\nu_{k,i}$ can be easily controlled by the constant L and the diminishing stepsize. While under $(L_0,L_1)$-smooth condition, the changes between $\nu_{k,i}$ are quite complicated with historical gradient information involved. Furthermore, the error brought by the change of $\nu_{k,i}$ is itself bounded under $L$-smooth condition, while it is unbounded under $(L_0,L_1)$-smooth condition and may even prevent the objective function from decreasing.  We address these difficulties and explain how to solve them in [Stage I, Section 4.2].
>
> Second, we use the potential function f(u_k), which allows us to derive a trajectory-wise convergence result instead of the in-expectation convergence results in [1] (a detailed discussion can be seen in Remark 8 of this paper).
>
> **As for [2]**:
> We acknowledge that [2] has established the weakness of GD under $(L_0,L_1)$-smooth condition with a constant learning rate. However, our counterexample using diminishing learning rate is stronger. This is because using a diminishing learning rate can make diverging algorithms converge. As an example, SGD does not converge to a stationary point with bounded variance noise using a constant learning rate but converges using a diminishing learning rate.
>
> Our significance over [2] further lies in the convergence result of Adam. [2] studied the convergence of clipped SGD while we study Adam. There are huge differences between these two algorithms:  First, Adam uses the gradient information of historical iterations to update the learning rate, while clipped SGD only uses the current gradient to rectify the learning rate. Second, Adam uses coordinate-wise learning rates, while clipped SGD uses a unified learning rate.  As such, it requires developing new and rather different techniques in order to analyze Adam, which justifies the theoretical significance of our result.
>
> **As for [3,4,5]**: Thanks again for pointing out that the "new potential function" in Section 4.2 is confusing. What we mean is that such a potential function is applied to Adam for the first time with convergence derived. We have removed the "new potential function", and have discussed these paper [3,4,5] in Section 4.2 and Appendix B as follows: We notice that similar potential functions have already been applied in the analysis of other momentum-based optimizers, e.g., momentum (S)GD in [3] and [4] and Adam-type optimizers (except Adam) in [5]. However, extending the proof to Adam is highly nontrivial and [5] fails to provide a convergence result for Adam. The key difficulty is to bound the change of $\nu_{k,i}$ as discussed above.

---

> ### Author Response · Authors · 2022-11-15
> **(Part 2) Thank you for your feedbacks and constructive comments**
>
> **Q3**: More theoretical comparisons between Adam and GD/SGD can be discussed.:1) both GD/SGD and Adam use the constant learning rate; 2) If the hyperparameters of Adam are chosen improperly, the convergence of Adam can be arbitrarily slow; 3) The convergence rate of Adam in Theorem 1 depends on the dimension, while GD/SGD does not depend on it.
>
> **A3**: **As for 1).**: We have added a Remark 4 stating the convergence of Adam with a constant learning rate. Roughly speaking, Adam will converge faster to the neighborhood (with the rate $1/\sqrt{t}\rightarrow 1/t$), but the size of the neighborhood is enlarged by an additional term $\mathcal{O}(\eta)$ due to the step-size under $(L_0, L_1)$-smooth assumption. In contrast, SGD with a constant learning rate can still converge arbitrarily slow under the $(L_0, L_1)$-smooth assumption.
>
> **As for 2).**:  When beta2 is chosen improperly(i.e., not close to 1), Adam can diverge even under L-smooth condition. This is shown by [Zhang et al. 2022]. This indicates the convergence condition that \beta_2 is close to 1 is both sufficient and necessary. Therefore, when demonstrating the advantage of Adam over SGD, the hyperparameters (\beta_1,\beta_2) of Adam should be "properly selected".
>
> **As for 3).**:
> Correct us if wrong: the reviewer seems to suggest that "SGD converges with rate independent on d". We respectfully point out that this comment is only true under $L$-smooth condition.   Under $(L_0,L_1)$-smooth condition, we have shown that SGD might diverge.  To justify the advantage of Adam, we only need to show Adam converges in the same setting (as shown in Theorem 1). As such, Although with a dependence on $d$, Theorem 1 already suffices to lead us to the desired conclusion.
>
> We believe that removing the dependence on d is possible and worthy of study. However, it is technically difficult and deserves independent work. We leave this part as future directions and will include the discussion under Theorem 1.
>
> **Q4**: As the paper contains many lemmas and theorems. It can be better organized for readers, especially for the appendix part, e.g. the first inequality in the proof of Lemma 2.
>
> **A4**: Thanks for pointing it out and we have revised the paper accordingly.
>
> **Q5**: What does the historical information mean?
>
> **A5**: "Historical information" means the "gradient information of historical iterations". We have revised the paper accordingly.
>
> **Q6**:The theorem suggests that $\beta_2=1$  is superior for Adam since $\delta(\beta_2)=0$  in this case.
>
> **A6**: Thanks for pointing it out. Theorem 1 only applies to the case $\beta_2<1$. We mention it when we describe the Adam algorithm in Section 3. We have added this condition Theorem 1 in the revised draft.
>
> **Q7**: In practice, it is common to use a constant learning rate in a stage and it is known that GD converges with a constant step size. So what is the convergence result of Adam when a constant learning rate is used?
>
> **A7**: Please refer to **1)** in **A3**.
>
> **Q8**: What is the advantage of Adam over the GD/SGD with gradient clipping or normalized gradient?
>
> **A8**: We are still not clear on what the advantage is. One potential advantage may be that Adam can handle more complex noise. This is because: our convergence of Adam is under affine noise variance assumption (Assumption 2), while the previous results on gradient clipping ([2] and [8]) require a **strictly stronger assumption** called "almost surely bounded noise".  It seems unclear how to relax this bounded condition for gradient clipping.

---

> ### Author Response · Authors · 2022-11-15
> **(Part 3) Thank you for your feedbacks and constructive comments**
>
> **Q9**: In finetuning tasks, where the gradient norm is usually small, more discussion can be added.
>
> **A9**: We have added the following discussion in Appendix B: For the case where the gradient along the trajectory is small, $(L_0,L_1)$-smooth condition will degenerate to $L$-smooth condition, and thus SGD works well. This may explain the phenomenon that SGD is also adopted in some finetuning tasks, as pretraining can be viewed as selecting a good initialization (and we can expect that the gradient is small along the trajectory). While the above discussion is rather intuitive, it is an interesting future work to formally prove it.
>
> **Q10**: Why $u_{k,i}$ is close to $w_{k,i}$?
>
> **A10**: In a word, it is due to the bounded update of Adam and small learning rate. Specifically, by the definition of u_{k,i}, we have $u_{k,i}=w_{k,i}+\frac{\beta_1}{1-\beta_1} (w_{k,i}-w_{k,i-1})$. Therefore, we only need to show that the update $w_{k,i}-w_{k,i-1}$ is small. We then check the form of the update $w_{k,i}-w_{k,i-1}=-\eta_k\frac{1}{\sqrt{\nu_{k,i}}+\varepsilon 1_d }\odot m_{k,i}$. By Cauchy's inequality, $\Vert \frac{1}{\sqrt{\nu_{k,i}}+\varepsilon 1_d }\odot m_{k,i}\Vert $ is bounded a constant which only depends on $(\beta_1,\beta_2)$ (please refer to Lemma 1 for details), and $\eta_k$ is small since it is diminishing. We then derive $w_{k,i}-w_{k,i-1}$ is small by combining the above facts together.
>
> **References**:
>
> [1] Yushun Zhang, et al. Adam can converge without any modification on update rules. Advances in Neural Information Processing Systems, 2022.
>
> [2] Jingzhao Zhang, et al. Why gradient clipping accelerates training: A theoretical justification for adaptivity. In International Conference on Learning Representations, 2019.
>
> [3] Euhanna Ghadimi, et al. Global convergence of the Heavy-ball method for convex optimization. European Control Conference, 2015.
>
> [4] Yanli Liu et al. An Improved Analysis of Stochastic Gradient Descent with Momentum. Advances in Neural Information Processing Systems, 2020.
>
> [5] Xiangyi Chen, et al. On the convergence of a class of Adam-type algorithms for non-convex optimization. International Conference on Learning Representations, 2019.
>
> [6] Naichen Shi, Dawei Li, Mingyi Hong, and Ruoyu Sun. RMSProp converges with proper hyper- parameter. In International Conference on Learning Representations, 2021.
>
> [7]. Deffosez et al., A Simple Convergence Proof of Adam and Adagrad, TMLR 2022
>
> [8]. Zhang et al., Improved Analysis of Clipping Algorithms for Non-convex Optimization, 2020

---

> ### Author Response · Authors · 2022-12-09
> **Any concern remains?**
>
> We would like to thank the reviewer for your constructive comments, and we appreciate the time you put into handling our paper. Given our detailed reply, please let us know if your concerns remain. Are there any additional questions we can answer to facilitate a fair reevaluation of this work?
>
> Thanks,
> Paper 388 authors

---

### Official Review · Reviewer_Ydg4 · 2022-10-31

**Confidence:** 3
**Correctness:** 3
**Technical Novelty And Significance:** 3
**Empirical Novelty And Significance:** 3
**Recommendation:** 6

**Clarity, Quality, Novelty And Reproducibility:**

This paper is well-written and easy to follow. I really appreciate the effort made by the authors to clearly explains the motivation, setting, and proof ideas. I didn't check the proofs in the appendix due to the limited time fo reviewing, but the sketch in the main paper looks very reasonable.

**Strength And Weaknesses:**

##Pros:

1. This paper tackles an important problem in optimization theory, regarding the superior convergence of ADAM over SGD. Despite the empirical success of ADAM, no theoretical separation on convergence rate has been made before. This paper first made such separation rigorously, in the $(L_0,L_1)$-smooth setting, which is arguably more realistic than the standard $L$-smooth setting for deep learning loss.

2. This paper is well-written and easy to follow. I really appreciate the effort made by the authors to clearly explains the motivation, setting and proof ideas.

3. The paper also points out a necessary condition for ADAM to converge, that is, $\beta_2$ has to be sufficiently close to $1$. This is reflected through both  their upper bounds for gradient norm and their simulation.

4. Though the indivual insights from the proofs are not compeletely novel, like the construction of the new poenntial function highlighted in Insight 1 is well-known (e.g., Ghadimi et al. 2015, equation (7)), combnining those technques and applying them correctly on a new setting is still non-trivial and needs a lot of effort.

##Cons:

1. Momentum seems not useful at all in the analysis. Indeed the existence of momentum seems only to make the analysis more complicated. To achieve the separation against SGD, it seems RMSProp (setting $\beta_1=0$) is already enough. I note the authors do mention understanding the effect of momentum in future work, but it would be more useful to spell out or give a related discussion in the main paper as it can help the readers to understand.

2. In the abstract, the authors claim that "Adam can adapt to local smoothness, justifying Adam's adaptivity". It is not clear what this sentence actually means in the algorithm level, e.g., does it mean the effective lr always maintains a certain relationship to the hessian? It seems the adaptivity of the moment estimation in ADAM holds by definition. This paper is really about the benefit of adaptivity. Same issue exists for the title, that is, instead of provable adaptivity, it should be the provable benefit of adaptivity.

2. It is not clear to me whether Figure 1(b) satisfies the claimed relationship of 'local smoothness = $e \times$ gradient norm'. It seems to me the slope is larger than 1, e.g., the right-top point is close to (1.0,2.5). I suggest the authors run a linear regression to decide the slope, which would be more convincing.

##Detailed comments:

1. In modern networks like ResNet and Transformer, normalization layers are ubiquitous. But unfortunately, the normalization layers doesn't quite satisfy the $(L_0,L_1)$-smoothness, because if one keeps the parameter before some normalization layer while keeping its direction, the gradient is proportional to the inverse of parameter norm and the hessian is proportional to the inverse of the squared parameter norm, which means that the local smoothness of any neural networks grows provably at the order of squared gradient norm. Thus a related question is, what is the limit of the current approach? Can it be used to deal with an even broader class of loss functions, which allow the smoothness to grow as a higher degree polynomial of gradient norm?

2. typo, in Assumption 1, "there exists positive constants" should be there exist.


3. typo, on page 5, "Assumption 2 is quite general. When $D_1 = 1/n$". I think it should $D_1=0$.

#References

- Ghadimi, Euhanna, Hamid Reza Feyzmahdavian, and Mikael Johansson. "Global convergence of the heavy-ball method for convex optimization." In 2015 European control conference (ECC), pp. 310-315. IEEE, 2015.


**Summary Of The Paper:**

ADAM has been observed to have better convergence properties than SGD in property, but the existing theoretical analysis cannot distinguish them, under the standard L-smooth setting. This paper considers a strictly broader class of loss functions, that is, the functions satisfy $(L_0, L_1)$-smoothness (Zhang et al., 2019), and rigorously show that ADAM can find an approximate first-order stationary point for any $(L_0, L_1)$-smooth loss function, while there is some $(L_0, L_1)$-smooth function that SGD will diverge.


**Summary Of The Review:**

This paper studies an open problem that lies at the core of the interest of the ICLR community, that why ADAM can have better faster convergence than SGD. The authors give a theoretical rigorous treatment for convergence of ADAM to approximate the first stationary point under the recent proposed $(L_0,L_1)$-setting, where SGD is not guaranteed to converge, even with $1/\sqrt{t}$ learning rates. This is a good paper and thus I suggest acceptance.


=======

Update after rebuttal: After discussion with AC, I realize that the current bound in Theorem 1 also hides factors like $poly(1/(1-\beta_2))$ in $O(\cdot)$. This dependency should be mentioned explicitly and discussed in detail because it affects the optimal choice of $\beta_2$. Therefore I decreased my score to 6.

---

> ### Author Response · Authors · 2022-11-15
> **Thank you for your positive feedback and constructive comments**
>
> We thank the reviewer for positive feedback and constructive comments. The concerns are addressed below. The paper has been revised accordingly and marked blue in the updated manuscript.
>
> **Q1**: Momentum seems not useful at all in the analysis. To achieve the separation against SGD, it seems RMSProp is already enough. It is suggested that the authors spell out or give a related discussion about the effect of momentum in the main paper.
>
> **A1**: Indeed in the current analysis, the momentum term is an obstacle to overcome rather than a beneficial term. We added a result (Remark 5) for RMSProp as a direct conclusion of Theorem 1 in the revised draft.
>
> We agree it is interesting to investigate the advantage of using Adam versus RMSprop (i.e., showing the benefit of momentum). However, this is highly non-trivial as the effect of momentum is not clear even for momentum SGD in non-convex optimization. We believe that this question is beyond the scope of this paper and leave it as a future work.
>
> **Q2**: In the abstract and the title, the "(provable) adaptivity" should be "(provable) benefit of adaptivity".
>
> **A2**: Thanks for the suggestion and we agree. The paper has been revised accordingly.
>
> **Q3**: The slope in Figure 1 (b) seems larger than 1.
>
> **A3**: Thanks for pointing it out. In order to verify $(L_0,L_1)$- smooth condition, we only need to show that $(\log gradient, \log smoothness)$ points lie under (instead of precisely on) a line with slope 1. We draw a line $\log smoothness=1\times\log gradient+1.4$ in the picture and find that the points all indeed lie under this line, so $(L_0,L_1)$- smooth condition indeed holds. We have added this line to Figure 1 (b) and modified the caption of Figure 1 accordingly in the revised paper.
>
> **Q4**: What is the limit of the current approach? Can it be used to deal with an even broader class of loss functions, which allow the smoothness to grow as a higher degree polynomial of gradient norm?
>
> **A4**: If the smoothness grows as a higher degree polynomial of gradient norm, it is unclear how to prove the convergence of Adam through the current approach. As shown by [Stage I, Section 4.2], the proof of convergence uses the fact that the second-order term matches the first-order term in terms of the degree of $\Vert \nabla f(w_{k,0})\Vert$ while the second order term has a smaller coefficiency. If the smoothness grows as a higher degree polynomial of gradient norm, the degree of $\Vert \nabla f(w_{k,0})\Vert$ in the second order term is larger than the first order term, and the first order term no longer dominates.
> However, we do find that studying whether and how Adam can converge under such a looser smoothness condition is interesting, and we put it as future work.
>
> **Q5**: Typos.
>
> **A5**: Thanks for pointing out the typos and we have revised the paper accordingly with the blue mark.

---

> > ### Comment · Reviewer_Ydg4 · 2022-11-16
> > **A follow-up question**
> >
> > Thanks for the clarification of the intention of Figure 1(b). Given the authors' explanation to Q3, I think it is fair to say that Figure 1(b) admits a constant L-smoothness. Thus my question is what is the advantage of the analysis in this paper compared to the standard analysis of ADAM in the L-smoothness setting? On one hand, in a single run of ADAM, the $L$-smoothness is always bounded by some positive real number. On the other hand, if we take the supremum of all possible weights, then neither $(L_0,L_1)$-smoothness assumption nor $L$-smoothness assumption are correct for models with normalization. Do you get better constants by decomposing $L$-smoothness into $(L_0, L_1)$-smoothness in the setting of Figure 1(b)? This needs some justification otherwise it might limit the current paper's contribution.

---

> > > ### Author Response · Authors · 2022-11-16
> > > **Interesting question**
> > >
> > > Thanks for this interesting question.
> > >
> > > First, while in a single run of Adam, $(L_0,L_1)$-smooth condition and $L$-smooth condition both apply,  $(L_0,L_1)$-smooth condition allows us to derive a tighter bound. Suppose $(L_0,L_1)$-smooth condition holds, and the gradient norm is bounded by $M$ in the single run. Then we have $L=L_0+L_1M$-smooth condition also hold. Therefore, the convergence rate derived under $L$-smooth condition has a dependence on $M$. Specifically, Theorem 1 shows that the rate has the form $O(L_0^2+L_1^2)$ under $(L_0,L_1)$-smooth condition, which is independent of $M$, and thus has the form $O((L_0+L_1M)^2)$ under $L$-smooth condition. When $M$ is large, the rate under $L$-smooth condition can be significantly looser than that under $(L_0,L_1)$-smooth condition.
> > >
> > > Second, $L_1$ is invariant with the objective scaling, which perfectly matches the practical scale-invariant property of Adam.

---

### Decision · Program_Chairs · 2023-01-20

**Decision:**

Reject

**Justification For Why Not Higher Score:**

Lack of transparency, lack of mathematical rigor in a theory paper, weak results.

**Justification For Why Not Lower Score:**

N/A

**Metareview: Summary, Strengths And Weaknesses:**

The paper proposed an analysis of the convergence of the average gradients to a small number (different than 0) of the gradients of Adam when used on functions satisfying the (L_0, L_1) condition. This result is established under the critical assumption that we minimize a finite-sum objective function using sampling without replacement.

While the result is potentially new, there are a number of problems with this submission.

First of all, the critical assumption on the sampling without replacement is hidden in the submission. Indeed, there is no mention of it till Section 3 and even there it is not clearly stated as such. This lack of transparency essentially oversells the results of the paper and it was the source of confusion for the reviewers and for myself. Other major problems are present in the main theorem. First of all, the theorem does not show that the minimal gradient converges to 0, but only to a small value depending on beta_2. In fact, the theorem cryptically states that the theorem holds only for beta_2<1, while the results seems to get uniformly better when beta_2 approaches 1. Questioned on this point by a reviewer, the authors did not clarify it. Only looking at the proof, we realized that there is a number of constants hidden in the big-O notation that explode when beta_2=1. This effectively means that beta_2 is not free as the authors claim, but it has to be set as a function of the number of iterations to obtain a convergence of the minimal gradient to 0.
Finally, the dependency on the number of functions in the finite-sum objective function is not clearly discussed and it is not apparent when the convergence rate is not vacuous: if we need a number of epochs equal to the number of training samples, it should be clear that this result is useless.

Overall, the paper falls below the bar to be accepted at ICLR.

**Summary Of Ac-Reviewer Meeting:**

I discussed this paper through mails with Reviewer Ydg4. We agreed that two of the points I raised were actually fine, but he agreed with me that he didn't realize about the issue of beta_2.